

# Five-dimensional SCFTs and gauge theory phases: an M-theory/type IIA perspective

Cyril Closset[1], Michele Del Zotto[2] and Vivek Saxena[3]

**1** Mathematical Institute, University of Oxford
Woodstock Road, Oxford, OX2 6GG, United Kingdom
**2** Department of Mathematical Sciences and Centre for Particle Theory
Durham University, Durham DH1 3LE, UK
**3** C.N. Yang Institute for Theoretical Physics, Stony Brook University
Stony Brook, NY 11794-3840, USA

## Abstract

**We revisit the correspondence between Calabi-Yau (CY) threefold isolated singularities X and five-dimensional superconformal field theories (SCFTs), which arise at low energy in M-theory on the space-time transverse to X. Focussing on the case of toric CY singularities, we analyze the "gauge-theory phases" of the SCFT by exploiting fiberwise M-theory/type IIA duality. In this setup, the low-energy gauge group simply arises on stacks of coincident D6-branes wrapping 2-cycles in some ALE space of type $A_{M-1}$ fibered over a real line, and the map between the Kähler parameters of X and the Coulomb branch parameters of the field theory (masses and VEVs) can be read off systematically. Different type IIA "reductions" give rise to different gauge theory phases, whose existence depends on the particular (partial) resolutions of the isolated singularity X. We also comment on the case of non-isolated toric singularities. Incidentally, we propose a slightly modified expression for the Coulomb-branch prepotential of 5d $\mathcal{N} = 1$ gauge theories.**

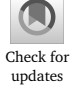
# 1  Introduction

Five-dimensional gauge theories are infrared-free and become strongly coupled at scales of the order of the inverse gauge coupling, $1/g^2$. Being non-renormalizable, they are not, by themselves, well-defined quantum field theories. In the mid-1990's, it was realized that many supersymmetric 5d gauge theories could be viewed as the low-energy description of certain deformations of superconformal field theories (SCFTs) in five and six dimensions [1–5].[1] A key feature of these RG flows is that a given SCFT can admit several deformations of this kind, leading to apparently inequivalent gauge theory phases in the infrared (IR) [5], a phenomenon sometimes called an "ultraviolet (UV) duality" [8–13].

Since renormalization group (RG) flows are unidirectional, one should really start with the 5d SCFT and consider its deformations to whatever IR phases are possible. The five-dimensional SCFT itself is necessarily strongly coupled [14], therefore a direct field-theory analysis would be very challenging to perform. Instead, suitable string-theory embeddings of the SCFT allow us to study its deformation to massive phases rather systematically. The purpose of this paper is to provide a detailed exploration of this aspect of the physics of these systems, addressing the infrared phases of selected 5d SCFTs and their deformations. Along the way, we obtain several other new results, a short account of which can be found below. In particular, we obtain a new expression for the well-known 5d Coulomb branch prepotential that correctly accounts for the 5d parity anomaly.

The ideal laboratory for our study is provided by five-dimensional field theories obtained by geometric engineering in M-theory [2–4] — see also [15–17] for more on geometric engineering and [18–26] for more recent studies. The conjectured correspondence between local Calabi-Yau (CY) threefold isolated singularities **X** and 5d SCFTs

$$\text{M-theory on } \mathbb{R}^{1,4} \times \mathbf{X} \qquad \longleftrightarrow \qquad \mathcal{T}_{\mathbf{X}} \equiv \text{SCFT}(\mathbf{X}) \,, \qquad (1.1)$$

provides geometric tools to address the structure of the Coulomb phase of $\mathcal{T}_{\mathbf{X}}$ and of its deformations. Another well-known construction of 5d SCFTs is in terms of $(p,q)$-brane webs in type IIB [5, 9, 12, 27–29]. Whenever **X** is a toric singularity, the $(p,q)$-fivebrane web and the M-theory setup are dual to each other [30]. Further evidence for the existence of these 5d fixed points, in some appropriate large $N$ limit, is provided by the AdS/CFT duality [31–41].

In the present paper, we introduce a complementary description to the M-theory and $(p,q)$-webs, focussing on the case when **X** is toric. [2] The M-theory description is purely geometric,

---

[1]For instance, five-dimensional maximal SYM is believed to describe the 6d $\mathcal{N} = (2,0)$ theory compactified on a circle with radius $R = g^2$—in the $SU(N)$ case, this is directly implied by the M-theory/type IIA duality [6, 7], since a stack of $N$ M5-branes wrapping a circle is dual to a stack of $N$ D4-branes in flat space.

[2]Many of the recent results obtained in the context of $(p,q)$-brane webs in IIB with orientifolds [42–46] have non-toric M-theory duals. We leave the natural generalization of our methods to these setups (consisting of including orientifold planes and D8-branes in our analysis) for future work.

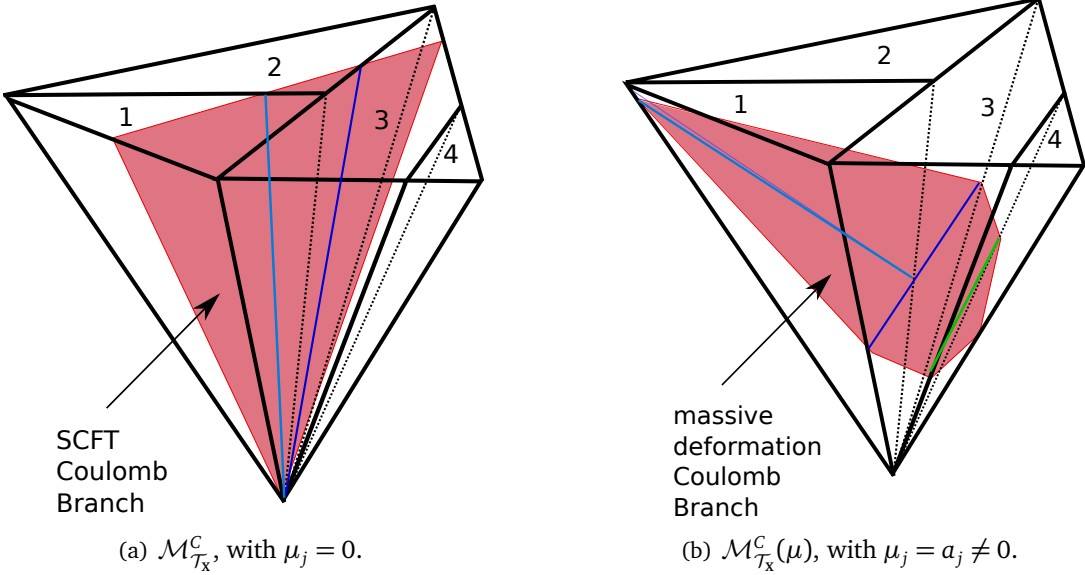

(a) $\mathcal{M}^{C}_{\mathcal{T}_{\mathbf{X}}}$, with $\mu_j = 0$.

(b) $\mathcal{M}^{C}_{\mathcal{T}_{\mathbf{X}}}(\mu)$, with $\mu_j = a_j \neq 0$.

Figure 1: Structure of the extended Coulomb parameter space of a 5d SCFT. On the left-hand-side, we represent the codimension-$f$ hypersurface $\mu_j = 0$ (in violet), which corresponds to the Coulomb branch, $\mathcal{M}^{C}_{\mathcal{T}_{\mathbf{X}}}$, of the SCFT $\mathcal{T}_{\mathbf{X}}$ itself. Any other slicing of the parameter space with $\mu_j = a_j \neq 0$, as depicted on the right-hand-side, corresponds to the Coulomb branch of a theory obtained from the 5d SCFT by a mass deformation.

while the IIB picture is purely in terms of branes in flat space. Our setup gives an intermediate dual configuration in type IIA string theory, in terms of D6-branes and geometry. The arbitrary-looking separation between "branes" and "geometry" in the type IIA setup will have a very neat interpretation in terms of the "gauge theory phases" of the SCFT $\mathcal{T}_{\mathbf{X}}$, as we will explain momentarily.

In the rest of this introduction, we first review some general features of $\mathcal{T}_{\mathbf{X}}$. Then, we summarize our type IIA approach, and state our main results. Finally, we discuss some more subtle point regarding the so-called parity anomaly in five-dimensional field theories.

## 1.1 Extended parameter space of a 5d SCFT

n this paper, we are interested in studying the Coulomb phase of 5d SCFTs. From the geometric engineering (1.1), the picture that emerges is as follows. The space of all possible Coulomb-branch vacuum expectation values (VEVs) and massive deformations of the field theory $\mathcal{T}_{\mathbf{X}}$ is identified with the extended Kähler cone of the singularity [47], which we refer to as the extended (Coulomb) parameter space of the 5d SCFT—see figure 1 for a schematic view. [3] This space is obtained as the union of the Kähler cones of all possible crepant resolutions of the singularity $\mathbf{X}$. The latter are smooth local CY threefolds, that we denote $\widehat{\mathbf{X}}_\ell$. Each such resolution gives a chamber on the extended Coulomb parameter space, corresponding to a given phase that we denote by $\mathscr{C}_\ell$. In Figure 1, we depict a case with four possible phases $\ell = 1, 2, 3, 4$. The VEVs of Coulomb branch operators in the 5d SCFT correspond to

---

[3] The 5d SCFT has a larger parameter space which includes the VEVs of Higgs branch operators. The latter are identified via the correspondence (1.1) with the space of complex structure deformations of the singularity $\mathbf{X}$ [2, 4, 5]. The 5d SCFT Higgs branch has been studied recently *e.g.* in [48–50]. In this work, we focus on the Coulomb branch.

the extended Kähler moduli dual to compact divisors, while the BPS massive deformations of the 5d SCFT correspond to the extended Kähler parameters that are dual to non-compact divisors. [4] We denote the former by $\nu_a$, where $a = 1, ..., r$ and $r$ is the rank of the corresponding SCFT, and the latter by $\mu_j$ where $j = 1, ..., f$ and $f$ is the rank of the global symmetry group of the SCFT. These real variables $(\nu, \mu) \in \mathbb{R}^{r+f}$ can be thought of as coordinates for an ambient space in which the extended Kähler cone is embedded, with the $\nu_a$'s being the Coulomb branch scalars. At any fixed value of the mass parameters $\mu_j$, the Coulomb branch metric (and the rest of the low-energy effective action) is captured by a prepotential that is computed geometrically as a function of triple-intersection numbers of a crepant resolution of $\mathbf{X}$, corresponding to the values of the extended Kähler moduli $\nu_a$ and $\mu_j$:

$$\mathcal{F} = \mathcal{F}(\nu_a, \mu_j) = \text{vol}(\widehat{\mathbf{X}}_\ell) \qquad \text{for} \qquad (\nu_a, \mu_j) \in \mathscr{C}_\ell , \tag{1.2}$$

for some properly regularized Kähler volume of the non-compact threefold. This prepotential is non-smooth at codimension-one walls in the interior of the extended Kähler cone, corresponding to flop transitions among birationally-equivalent resolutions $\widehat{\mathbf{X}}_\ell$ and $\widehat{\mathbf{X}}_{\ell'}$—these are codimension-one walls along which $\mathscr{C}_\ell$ and $\mathscr{C}_{\ell'}$ intersect. [5] The external boundaries of the extended Kähler cone are characterized by divisors shrinking either to a point or a curve. The former case corresponds to tensionless string, the latter to a gauge enhancement. The Coulomb branch of the SCFT is canonically identified as the codimension-$f$ hypersurface:

$$\mathcal{M}^C_{\mathcal{T}_\mathbf{X}} \equiv \{\mu_j = 0\} \cap \widehat{\mathcal{K}}(\mathbf{X}) . \tag{1.3}$$

This identification entails that we can compute the SCFT Coulomb branch prepotential as:

$$\mathcal{F}_{\text{SCFT}} = \mathcal{F}_{\text{SCFT}}(\nu) \equiv \mathcal{F}(\nu, \mu)\Big|_{\mu_j = 0, \forall j} . \tag{1.4}$$

This space has been referred to, in the recent literature, as the "physical Coulomb branch" [22]. We stress here that, since the gauge coupling is a specific deformation of the SCFT, this cannot be a gauge theory phase; but, as we shall see below, it is compatible with one such phase whenever the corresponding chamber admits a gauge theory interpretation and survives the $\mu_j \to 0$ limit. The structure of phases along the SCFT Coulomb branch is given by the interesections of $\mathcal{M}^C_{\mathcal{T}_\mathbf{X}}$ with the chambers $\mathscr{C}_\ell$. For instance in figure 1 the SCFT Coulomb branch is not compatible with the chamber $\mathscr{C}_4$; yet, it is possible to reach a theory in phase $\mathscr{C}_4$ by deforming the SCFT (see figure 1(b)).

In this sense, the geometry of the resolutions of $\mathbf{X}$ gives the analogue of the Seiberg-Witten solution of 4d $\mathcal{N} = 2$ theories for this class of 5d $\mathcal{N} = 1$ SCFTs. Exploiting geometry, one can compute the prepotential for any value of the parameters and understand the structure of the fibration of the Coulomb moduli over the space of deformation parameters of the SCFT.

## 1.2 Type IIA perspective and 5d $\mathcal{N} = 1$ supersymmetric gauge theories

In the following, we revisit the problem of understanding which deformations of $\mathcal{T}_\mathbf{X}$ have a low-energy description as gauge theories. Our main tool is the M-theory/IIA duality. In this section, as in most of the paper, we assume that $\mathbf{X}$ is toric. As we have reviewed above, any relevant deformation of $\mathcal{T}_\mathbf{X}$ corresponds to performing a crepant resolution of the singularity:

$$\pi_\ell : \widehat{\mathbf{X}}_\ell \to \mathbf{X} , \tag{1.5}$$

---

[4]In this sentence, we distinguished between the "Kähler moduli" $\nu$, which are dynamical fields in the low-energy M-theory setup, and the "Kähler parameters" $\mu$, which are non-dynamical. In the following, by abuse of notation, we will use the two terms interchangeably.

[5]For instance, in Figure 1, the chambers $\mathscr{C}_3$ and $\mathscr{C}_4$ are connected by a flop transition, while $\mathscr{C}_1$ and $\mathscr{C}_4$ are not.

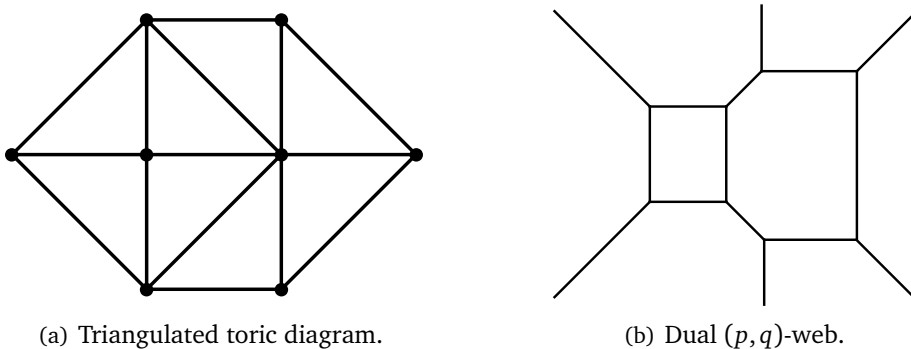

(a) Triangulated toric diagram.     (b) Dual $(p,q)$-web.

Figure 2: Triangulated toric diagram and dual $(p,q)$ web for a particular resolution of the "beetle" singularity. This resolution admits both an $SU(3)$ $N_f = 2$ and an $SU(2) \times SU(2)$ quiver gauge-theory description.

with $\widehat{\mathbf{X}}_\ell$ a smooth (or, at least, less singular) local CY$_3$. We then choose an abelian sugroup $U(1)_M$ which is part of the $T^3$ toric action on the threefold $\widehat{\mathbf{X}}_\ell$, and we consider the "reduction" to type IIA string theory along that $U(1)_M$, viewed as the "M-theory circle" [51–54]. On general ground, we have a duality:

$$\text{M-theory on } \mathbb{R}^{1,4} \times \widehat{\mathbf{X}}_\ell \qquad \longleftrightarrow \qquad \text{Type IIA string theory on } \mathbb{R}^{1,4} \times \mathcal{M}_5 , \qquad (1.6)$$

where the transverse five-manifold is the quotient of the local CY$_3$ by $U(1)_M$:

$$\mathcal{M}_5 \cong \widehat{\mathbf{X}}_\ell / U(1)_M . \qquad (1.7)$$

The resulting five-dimensional type IIA background is a fibration of the ALE space corresponding to the resolution of the singularity $\mathbb{C}^2/\mathbb{Z}_K$ over a real line. Under certain conditions, which we will spell out explicitly, the type-IIA description is under perturbative control and contains D6-branes. [6] In particular, there can be $N$ D6-branes wrapped over a $\mathbb{P}^1$ inside the ALE space, which leads to a non-abelian 5d $\mathcal{N} = 1$ supersymmetric $SU(N)$ gauge theory with an inverse gauge coupling $\frac{1}{g^2} = \text{vol}(\mathbb{P}^1)$. More generally, as we will see, deformations of toric singularities can only give rise to linear quivers with special unitary gauge groups. [7] Whenever the threefold has chambers that admit such a perturbative map to IIA, we identify a gauge theory chamber. If the gauge theory chamber is compatible with the SCFT Coulomb phase, meaning that it survives in the limit $\mu_j \to 0$ on the Kähler parameters, then we can consider the gauge-theory prepotential as valid even in the strong-coupling limit $\mu_j \to 0$, because by construction it is going to match with the geometry prepotential in the chamber considered. This gives a surprising result: it is possible to trust a gauge theory computation in the limit of infinite coupling, precisely when the effective field theory approximation breaks down. We stress that this is not going to be valid for all gauge theory phases, but only for those that are compatible with the SCFT Coulomb branch. For instance, in Figure 1, if chamber $\mathcal{C}_4$ is a gauge theory phase, it is not going to be connected to the origin of the moduli space.

For *isolated* toric singularities, we find a perfect matching between the allowed (*i.e.* "perturbative") IIA reductions and the Coulomb-branch chambers of the gauge theories in question. Our main example will be what we call the "beetle SCFT," a famous rank two SCFT which has

---

[6]Here, we mean "perturbative in the open-string sector." We do not keep track of the details of the metric of $\mathcal{M}_5$ beyond the relevant Kähler moduli inherited from $\widehat{\mathbf{X}}$; in particular, the curvatures need not be small.

[7]To obtain more general gauge groups and/or representations, we should leave the toric realm. We leave that generalization for future work.

gauge theory phases corresponding to $SU(3)$ $N_f = 2$ and $SU(2) \times SU(2)$ with a bifundamental hyper. The name "beetle" is suggested by the toric diagram and dual $(p, q)$-web of the CY threefold, as shown in Figure 2. The beetle geometry has 24 resolutions, out of which there are 16 resolutions that admit a IIA description with an $SU(3)$ $N_f = 2$ gauge-theory interpretation, and 6 resolutions that admit a IIA descriptions with an $SU(2) \times SU(2)$ quiver interpretation. (Moreover, we identify 6 chambers that do not admit any gauge theory interpretation at all.) These numbers match exactly with the number of inequivalent Coulomb chambers one finds in the respective gauge theories. Only 4 of these gauge-theory chambers overlap, meaning that they corresponds to resolutions that have both an $SU(2) \times SU(2)$ interpretation and an $SU(3)$ interpretation. Perhaps not surprisingly, these resolutions are precisely the ones that survive the map to the SCFT Coulomb branch, while the other chambers are not compatible with the $\mu_j \to 0$ limit.

For *non-isolated toric singularities*, this beautiful feature is lost. The number of perturbative type-IIA reductions does not match the number of Coulomb-branch phases of the corresponding gauge theories. We interpret this as an indication that non-isolated toric singularities do not fully characterize a 5d SCFT. In the literature, however, such singularities are often used to described 5d fixed points, since they are particularly convenient to deal with—see *e.g.* [9,29,55]. An important class of such examples is provided by the 5d $T_N$ SCFTs, which can seemingly be described as an orbifold toric singularity $\mathbb{C}^3/(\mathbb{Z}_N \times \mathbb{Z}_N)$ in M-theory [28]. We conjecture that, in any such case, there exists some non-toric *isolated* CY singularity that describe the 5d fixed point and whose extended Kähler cone encodes the fully extended parameter space of the SCFT, unlike the toric model. A well-known example is given by the $T_3$ theory, which is the 5d fixed point with $E_6$ exceptional symmetry first found by Seiberg [1]. In that case, the isolated (non-toric) singularity is the complex cone over the del Pezzo surface $dP_6$, to be compared with the $\mathbb{C}^3/(\mathbb{Z}_3 \times Z_3)$ toric singularity.

## 1.3 The 5d parity anomaly and the gauge-theory prepotential

An interesting feature of 5d field theories, as in any odd number of space-time dimension, is the possible presence of parity-odd terms in the effective action. [8] In particular, the five-dimensional Chern-Simons term for a 5d (background) gauge field breaks parity explicitly. Moreover, massless Dirac fermions coupled to gauge fields, such as the ones that appear on the walls of Coulomb branch chambers, suffer from the so-called *parity anomaly*, which is really a mixed gauge-parity anomaly [59]. In this work, we will use an (implicit) regularization that preserve gauge invariance for both dynamical and background gauge fields (coupling to conserved currents for gauged and global symmetries, respectively). This leads us to a slightly different form of the one-loop-exact prepotential of 5d $\mathcal{N} = 1$ supersymmetric gauge theories, compared to the well-known Intriligator-Morrison-Seiberg result [4] generally quoted in the literature. This is explained in detail in section 2 and appendix A, which can be read independently from the rest of this work.

This paper is organized as follows. In section 2, we revisit the basic properties of five dimensional gauge theories; in particular we give a new formula for the 5d prepotential from gauge theory. In section 3, we review the M-theory engineering of 5d SCFTs, and we formulate our IIA perspective on gauge theory phases. In sections 4 and 5, we revisit some known examples from our perspective. In section 6, we comment on the so-called "UV dualities" and we give a detailed study of the phases of the beetle SCFT. Section 7 is devoted to the analysis of non-isolated toric singularities. Various further computational details are collected in appendices.

---

[8]Here, "parity" is a slight misnomer, but this terminology is standard. See *e.g.* [56,57] for a detailed discussion in the 3d case, which is completely similar. Some important work related to this aspect of 5d $\mathcal{N} = 1$ gauge theories was recently carried out in [58].

# 2  5d $\mathcal{N} = 1$ gauge theories, parity anomaly and prepotential

In this section, we review and revisit some basic properties of five-dimensional $\mathcal{N} = 1$ supersymmetric gauge theories. We discuss some aspects of the five-dimensional parity anomaly, we revisit the derivation of the 5d prepotential, and we conclude with some remarks about the five-dimensional gauge theory Coulomb phases.

## 2.1  Five-dimensional gauge theory basics

Consider a five-dimensional $\mathcal{N} = 1$ supersymmetric gauge theory with gauge group $\mathbf{G}$ and Lie algebra $\mathfrak{g} = \text{Lie}(\mathbf{G})$. Unless otherwise stated, we take $\mathbf{G}$ compact and connected. We then have:

$$\mathbf{G} = \prod_s \mathbf{G}_s \,, \tag{2.1}$$

where each factor $\mathbf{G}_s$ is a simple Lie group. The vector multiplet $\mathcal{V}$ contains a 5d gauge field $A_\mu$, a real scalar $\varphi$, and the gaugini $\lambda, \widetilde{\lambda}$, all in the adjoint representation of $\mathfrak{g}$. It is coupled to matter fields in hypermultiplets $\mathcal{H}$, in some (generally reducible) representation $\mathfrak{R}$ of the gauge group.

These gauge theories have a non-trivial global symmetry:

$$\mathbf{G}_F \times SU(2)_R \,, \tag{2.2}$$

with $SU(2)_R$ the $R$-symmetry. The "flavor" symmetry group $\mathbf{G}_F$ contains a factor $\mathbf{G}_{\mathcal{H}}$ which acts only on the hypermultiplets. It also contains an abelian factor $U(1)_{T_s}$ for each topological symmetry:

$$\mathbf{G}_F = \mathbf{G}_{\mathcal{H}} \times \prod_s U(1)_{T_s} \,. \tag{2.3}$$

Recall that, for each simple Lie group $\mathbf{G}_s$ with connection $A_\mu^{(s)}$, we have a topological symmetry whose conserved current reads:

$$j_{T_s}^\mu = \frac{1}{32\pi^2} \epsilon^{\mu\nu\rho\sigma\kappa} \, \text{Tr}\big(F_{\nu\rho}^{(s)} F_{\sigma\kappa}^{(s)}\big) \,, \tag{2.4}$$

with $F^{(s)} = dA^{(s)} - iA^{(s)} \wedge A^{(s)}$. This is automatically conserved by virtue of the Bianchi identity. The particles charged under $U(1)_{T_s}$ are the 5d uplift of the 4d Yang-Mills $\mathbf{G}_s$ instantons, therefore the topological symmetry is also called the instanton symmetry.

To keep track of the flavor symmetry, we introduce background vector multiplets $\mathcal{V}_F$, including an abelian vector multiplet $\mathcal{V}_{T_s}$ for each topological current. In flat space, a non-trivial supersymmetric background for $\mathcal{V}_F$ is obtained by turning on constant VEVs for the real scalar $\varphi_F$. We denote these five-dimensional "real masses" by $\mu$:

$$\langle \varphi_F \rangle \equiv \mu = \big(m_\alpha \,, h_{0,s}\big) \,. \tag{2.5}$$

Here, the "flavor masses" for the hypermultiplet flavor group are simply denoted by $m = (m_\alpha)$, where $\alpha = 1, \cdots, \text{rank}(\mathbf{G}_{\mathcal{H}})$ runs over a maximal torus of $\mathbf{G}_{\mathcal{H}}$. The masses for the topological symmetries are denoted by $h_0 = (h_{0,s})$. The $U(1)_{T_s}$ mass term is also the Yang-Mills Lagrangian for the $\mathbf{G}_s$ vector multiplet, with:

$$h_0 = \frac{8\pi^2}{g^2} \tag{2.6}$$

the 5d inverse gauge coupling [1].

## 2.2 Parity anomaly, Chern-Simons terms and real masses

Five-dimensional parity acts by inverting the sign of a single coordinate on $\mathbb{R}^5$, say $x^5 \rightarrow -x^5$. Consider a single Dirac fermion $\psi$, transforming in the **4** of $\mathrm{Spin}(5) \cong Sp(2)$, and coupled to a $U(1)$ gauge field $A_\mu$ with charge 1. It is well-known that $\psi$ suffers from a parity anomaly—one cannot quantize $\psi$ while preserving both parity and gauge invariance.

One can always choose a gauge-invariant quantization, at the expense of violating parity [59]. The presence of the parity-violating term in the effective action is probed by a quantity denoted by:

$$\kappa \quad (\mathrm{mod}\ 1)\,. \tag{2.7}$$

The coefficient $\kappa$, sometimes called the "CS contact term," appears as a parity-odd term in the three-point function of the $U(1)$ current [58, 60]. The mod 1 ambiguity in (2.7) corresponds to the possibility of adding a Chern-Simons term to the 5d action, with integer-quantized level. We will come back to this point below.

**Chern-Simons terms.** Given a five-dimensional $U(1)$ gauge field $A_\mu$, we can write down the 5d Chern-Simons term:

$$S_{\mathrm{CS}} = \frac{ik}{24\pi^2} \int_{\mathcal{N}_5} A \wedge F \wedge F \tag{2.8}$$

on some Euclidean five-manifold $\mathcal{N}_5$. This is well-defined only if the CS level $k$ is integer quantized, $k \in \mathbb{Z}$. [9] The Chern-Simons term (2.8) breaks parity explicitly. It can be completed to the $\mathcal{N} = 1$ supersymmetric action, which takes the schematic form [1, 61]:

$$\frac{k}{24\pi^2} \int_{\mathcal{N}_5} \left( iA \wedge F \wedge F + 6i\bar{\lambda}\left(D + \frac{i}{2}\gamma^{\mu\nu}F_{\mu\nu}\right)\lambda - \frac{1}{6}\varphi F_{\mu\nu}F^{\mu\nu} + 12\varphi D^2 + \cdots \right)\,, \tag{2.9}$$

where $D$ denotes an $SU(2)_R$ triplet of auxiliary fields, and we omitted terms involving derivatives of $\lambda$ and $\varphi$. Note that the real field $\varphi$ is mapped to $-\varphi$ under parity.

More generally, we may consider the mixed $U(1)_a - U(1)_b - U(1)_c$ CS terms:

$$\frac{ik^{abc}}{24\pi^2} \int_{\mathcal{N}_5} A_a \wedge F_b \wedge F_c\,, \tag{2.10}$$

with quantized levels $k^{abc} \in \mathbb{Z}$. We may also have a non-abelian Chern-Simons term for any simple Lie group with non-trivial cubic index—that is, for $\mathfrak{g}_s = \mathrm{Lie}(\mathbf{G}_s) = \mathfrak{su}(N)$ with $N > 2$ [4]. All these CS terms have an $\mathcal{N} = 1$ completion similar to (2.9).

**Chern-Simons contact term and parity anomaly.** Consider again a (background) $U(1)$ gauge field coupled to a free fermion $\psi$, and let $j^\mu$ be the conserved current for the $U(1)$ symmetry that acts as $\psi \rightarrow e^{i\alpha}\psi$. The three-point function of $j^\mu$ contains the parity-odd contact term [58]:

$$\frac{i\kappa}{24\pi^2} \epsilon_{\mu\nu\rho\sigma\kappa} p^\sigma q^\kappa \ \subset\ \langle j_\mu(p) j_\nu(q) j_\rho(-p-q)\rangle\,. \tag{2.11}$$

The Chern-Simons term (2.8) obviously contributes an integer to the coefficient $\kappa$—that is, adding the "local term" (2.8) to the effective action for a free fermion has the effect of shifting $\kappa$ to $\kappa + k$. This explains the ambiguity (2.7). Since $k$ is integer-quantized, the non-integer part of the "CS contact term" $\kappa$ is physical [60].

---

[9]In this normalization, $k = 1$ is the minimal CS term when $\mathcal{N}_5$ is a spin five-manifold with $p_1 = 0$, as required by the M-theory engineering of these gauge theories [47]. On the other hand, $k = 6$ gives the minimal $U(1)$ CS term on an arbitrary $\mathcal{N}_5$.

For a single *massless* Dirac fermion $\psi$, one finds that $\kappa = -\frac{1}{2}$ (mod 1). We fix the integer-valued ambiguity by choosing the "$U(1)_{-\frac{1}{2}}$ quantization," such that:

$$\kappa_\psi = -\frac{1}{2} \tag{2.12}$$

for a massless free fermion. Any other quantization scheme is related to this one by a shift of $\kappa$ by an integer. In most of the literature on 5d $\mathcal{N} = 1$ gauge theories, this is called a "CS level $\kappa = -\frac{1}{2}$." Here, we would like to emphasize that $\kappa$ can be extracted from a gauge-invariant (non-local) effective action, while the CS level $k$ must be an integer by gauge invariance. Therefore, we should distinguish between the CS contact term $\kappa \in \mathbb{R}$ and the properly-quantized CS level $k \in \mathbb{Z}$. [10]

Since the hypermultiplet (coupled to a $U(1)$ gauge field) contains a Dirac fermion, we have the same parity anomaly in $\mathcal{N} = 1$ supersymmetric theories. By defaults, we will choose the $U(1)_{-\frac{1}{2}}$ quantization for every hypermultiplet.

**Massive fermions.**   Another parity-odd term is the real mass:

$$\delta\mathscr{L}_m = im\,\bar{\psi}\psi\,, \qquad m \in \mathbb{R}\,, \tag{2.13}$$

for any Dirac fermion $\psi$. By sending $|m| \to \infty$, one can integrate out $\psi$. A one-loop computation [47, 63] shows that this shifts the parity-odd contact term (2.11) by:

$$\delta\kappa = -\frac{1}{2}\,\mathrm{sign}(m)\,. \tag{2.14}$$

Let $\kappa_\psi(m)$ denote the CS contact term for the free fermion $\psi$, as a function of the real mass. In the $U(1)_{-\frac{1}{2}}$ quantization, we then have:

$$\lim_{m \to -\infty} \kappa_\psi(m) = 0\,, \qquad \lim_{m \to +\infty} \kappa_\psi(m) = -1\,, \tag{2.15}$$

by adding (2.14) to (2.12). This is interpreted as follows: for $m$ large and negative, the effective field theory is simply a CS term at level $k = -1$ for the background gauge field $A_\mu$; for $m$ large and positive, we have a completely empty theory in the infrared. The limit (2.15) can also serve as the definition of what is meant by "$U(1)_{-\frac{1}{2}}$ quantization."

**General charge.**   The above discussion was for a Dirac fermion of unit charge. More generally, for a massless hypermultiplet of $U(1)$ charge $Q \in \mathbb{Z}$, we have

$$\kappa_\psi = -\frac{Q^3}{2}\,, \tag{2.16}$$

and turning on a real mass leads to a shift $\delta\kappa = -\frac{Q^3}{2}\,\mathrm{sign}(m)$ in the IR.

**CS contact terms in general theories.**   More generally, we might consider a more complicated theory whose spectrum might be gapped. In a gapped phase, all the IR contact terms $\kappa$ must be properly quantized, since they should come entirely from Chern-Simons terms in the effective action [60]—this can be understood as a 5d version of the Coleman-Hill theorem [64]. At a gapless point, on the other hand, the observable $\kappa$ mod 1 is generally non-trivial; the simplest example being the free fermion, as in (2.16).

---

[10]We refer to Appendix A of [62] for a pedagogical discussion of this point, in the (completely analogous) three-dimensional case.

## 2.3 The Coulomb branch prepotential

By giving expectation values to the adjoint scalar $\varphi$:

$$\langle \varphi \rangle = \text{diag}(\varphi_a) = (\varphi_1, \cdots, \varphi_{\text{rk}(\mathbf{G})}), \tag{2.17}$$

we break the gauge group to a maximal torus $\mathbf{H}$ times the Weyl group:

$$\mathbf{G} \to \mathbf{H} \rtimes W_{\mathbf{G}}, \qquad \mathbf{H} \cong \prod_{a=1}^{\text{rk}(\mathbf{G})} U(1)_a. \tag{2.18}$$

The $\mathcal{N} = 1$ supersymmetric low-energy effective field theory on the Coulomb branch is fully determined by a prepotential $\mathcal{F}(\varphi, \mu)$. Here and below, $\varphi = (\varphi_a)$ denotes the low-energy Coulomb branch scalars, which sit in abelian vector multiplet $\mathcal{V}_a$, and $\mu = (m, h_0)$ denotes the real masses (2.5)—that is, the flavor masses $m_a$ and the inverse gauge couplings $h_{0,I}$.

The five-dimensional prepotential is a cubic polynomial in the real variables $\varphi$ and $\mu$. It is one-loop exact, with the one-loop contribution coming from integrating out the W-bosons and massive hypermultiplets at a generic point on the Coulomb branch [1, 4, 65]. We have:

$$\begin{aligned}
\mathcal{F}(\varphi, \mu) = & \frac{1}{2} h_{0,s} K_s^{ab} \varphi_a \varphi_b + \frac{k^{abc}}{6} \varphi_a \varphi_b \varphi_c + \frac{1}{6} \sum_{\alpha \in \Delta} \Theta(\alpha(\varphi))(\alpha(\varphi))^3 \\
& - \frac{1}{6} \sum_\omega \sum_{\rho \in \mathfrak{R}} \Theta(\rho(\varphi) + \omega(m))(\rho(\varphi) + \omega(m))^3,
\end{aligned} \tag{2.19}$$

where the sum over repeated indices ($s$ and $a, b, c$) is understood. Here, $\Theta(x)$ is the Heaviside step function:

$$\Theta(x) = \begin{cases} 1 & \text{if } x \geq 0, \\ 0 & \text{if } x < 0. \end{cases} \tag{2.20}$$

The first term in (2.19) is the classical contribution from the Yang-Mills terms, with $h_{0,s} = \frac{8\pi^2}{g_s^2}$ the inverse gauge couplings and $K_s^{ab}$ the Killing forms of the simple factors $\mathfrak{g}_s$. The second term in (2.19) gives the classical Chern-Simons action, with CS levels $k^{abc}$ in the abelianized theory. For $\mathbf{G} = \prod_s \mathbf{G}_s$ with $\mathbf{G}_s$ a simple gauge group, we only have CS contributions from the $SU(N)$ factors (with $N > 2$), with $k^{abc} = k_s d_s^{abc}$ for each $s$, in terms of the cubic Casimir $d_s^{abc}$ of $\mathfrak{g}_s$. The last term on the first line of (2.19) is the one-loop contribution from the W-bosons, with $\Delta$ the set of non-zero roots $\alpha = (\alpha^a)$ of $\mathfrak{g}$, and $\alpha(\varphi) = \alpha^a \varphi_a$ the natural pairing. The second line of (2.19) is the one-loop contribution from the hypermultiplets. The sum is over the flavor and gauge weights, $\omega$ and $\rho$, respectively, with $\omega(m) = \omega^\alpha m_\alpha$ and $\rho(\varphi) = \rho^a \varphi_a$.

This 5d prepotential is the correct result for the hypermultiplet in the "$U(1)_{-\frac{1}{2}}$ quantization," as explained above. That is, for a single hypermultiplet coupled to a $U(1)$ vector multiplet with scalar $\varphi$, we have the continuous function:

$$\mathcal{F}_{\mathcal{H}}(\varphi) = -\frac{1}{6} \Theta(\varphi) \varphi^3. \tag{2.21}$$

Since a $U(1)$ CS term at level $k$ contributes:

$$\mathcal{F}_{U(1)_k}(\varphi) = \frac{k}{6} \varphi^3, \tag{2.22}$$

to the prepotential, the hypermultiplet contribution (2.21) precisely reproduces the decoupling limits (2.15), for either sign of the real mass $\varphi$.

The prepotential (2.19) is derived in Appendix A. It should be compared to the result given by Intriligator, Morrison and Seiberg (IMS) in [4]. As we explain in Appendix, the terms of order $\varphi^3$ in (2.19) are the same as in the IMS prepotential (once we correctly map the CS levels), but our prescription gives a slightly different result for the lower-order terms. This is explained by our different treatment of the parity anomaly.

At a generic point on the Coulomb branch, the theory is gapped and therefore, as discussed above, the Chern-Simons contact terms $\kappa$ should all be integer-quantized. This is true not only for the gauge CS levels, but also for the mixed gauge-flavor and purely flavor CS levels:

$$\kappa^{abc} = \partial_{\varphi_a}\partial_{\varphi_b}\partial_{\varphi_c}\mathcal{F}\,, \quad \kappa^{ab\alpha} = \partial_{\varphi_a}\partial_{\varphi_b}\partial_{m_\alpha}\mathcal{F}\,, \quad \kappa^{a\alpha\beta} = \partial_{\varphi_a}\partial_{m_\alpha}\partial_{m_\beta}\mathcal{F} \quad \in \mathbb{Z}\,, \tag{2.23}$$

and so on (including by taking derivatives with respect to the gauge couplings $h_0$). The effective CS levels that follow from (2.19) are indeed integer quantized, by construction, which is not always the case if we use the IMS prepotential. [11]

## 2.4 BPS particles and strings

Five dimensional gauge theories on their Coulomb branch contain half-BPS particle states and half-BPS string states.

**BPS particles.** The VEV of $\varphi$ and the real masses $\mu$ determine the mass $M$ of BPS particle states, according to:

$$M = |Q^a \varphi_a + Q_F^\alpha m_\alpha + Q_F^s h_{0,s}|\,, \tag{2.24}$$

where $Q^a$, $Q_F^\alpha$ and $Q_F^s$ denote the integer-quantized gauge charges, the $\mathbf{G}_\mathcal{H}$ flavor charges, and the $U(1)_{T_s}$ instanton charges, respectively. The RHS of the formula above can be interpreted as the absolute value of the real central charge that enters in the 5d $\mathcal{N} = 1$ Poincaré superalgebra. The elementary states that carry gauge charge are the W-bosons $W_\alpha$, associated to the roots $\alpha \in \mathfrak{g}$, with:

$$M(W_\alpha) = \alpha(\varphi)\,. \tag{2.25}$$

The hypermultiplet states, with gauge charges $Q^a = \rho^a$ and flavor charges $Q_F^\alpha = \omega^\alpha$, have masses:

$$M(\mathcal{H}_{\rho,\omega}) = \rho(\varphi) + \omega(m)\,. \tag{2.26}$$

The prepotential (2.19) is non-smooth across loci where any particle become massless. We will discuss these Coulomb-branch "walls" further in subsection 2.6 below.

There are also BPS particles charged under the topological symmetries (that is, with $Q_F^s \neq 0$). These "instantonic particles" are solitonic particles in the gauge-theory language, corresponding to **G**-instantons in the four-dimensional sense. They are more subtle to understand from the low-energy point of view, but play a crucial role in the UV completion of the non-abelian gauge theories at strong coupling—see *e.g.* [48, 66–70]. We will discuss them further in subsection 2.5 below.

**BPS strings.** The 5d $\mathcal{N} = 1$ gauge theory also contains BPS strings, corresponding to the five-dimensional uplift of the four-dimensional $\mathcal{N} = 2$ monopoles. They are labelled by GNO-quantized magnetic fluxes through any $S^2$ surrounding them in $\mathbb{R}^5$. In particular, for every $U(1)_a$ in the maximal torus of **G**, there is a string with $U(1)_a$ magnetic flux $\mathfrak{m}_a = 1$, whose

---

[11]Using the IMS prepotential together with the usual treatment of the "parity anomaly," $\kappa^{abc}$ is always integer-quantized and agrees with our result, but some of the mixed flavor-gauge effective CS levels can be half-integer. In the usual (and somewhat misleading) language, our prescription for the prepotential can be understood as a correction to the IMS prepotential that adds some explicit "half-integer CS levels" on the Coulomb branch to "cancel the parity anomalies" for the flavor group.

tension on the Coulomb branch is given by the first derivative of the prepotential with respect to $\varphi_a$ [1]:

$$T_a(\varphi, \mu) = \frac{\partial \mathcal{F}}{\partial \varphi_a} . \tag{2.27}$$

In summary, while the second and third derivatives of $\mathcal{F}$ with respect to $\varphi$ determine the Coulomb-branch low-energy effective action, the first derivatives determine the string tensions.

## 2.5  Instanton particles in $SU(N)$ quivers

All of our examples below will be $SU(N)$ quivers—that is, we will have a gauge group:

$$\mathbf{G} = \prod_s SU(N_s) \tag{2.28}$$

coupled to bifundamental and/or fundamental hypermultiplets. In that simple case, we can compute the masses of the instanton particles by the following trick [71]. Let us first replace each $SU(N)$ gauge group by $U(N)$. The Coulomb branch scalars of $U(N)$ are denoted by:

$$\phi = (\phi_a) , \qquad a = 1, \cdots, N . \tag{2.29}$$

The prepotential for a $U(N)$ theory at CS level $k \in \mathbb{Z}$ reads: [12]

$$\mathcal{F}^{U(N)} = \frac{1}{2} h_0 \sum_{a=1}^N \phi_a^2 + \frac{k}{6} \sum_{a=1}^N \phi_a^3 + \frac{1}{6} \sum_{\substack{a,b=1 \\ a<b}}^N (\phi_a - \phi_b)^3 + \cdots . \tag{2.30}$$

Here, we choose a Weyl chamber of $U(N)$ such that $\phi_1 > \phi_2 > \cdots > \phi_N$, and the ellipsis is the one-loop contribution from hypermultiplets. The generalization to any $U(N)$ quiver is straightforward. Obviously, we recover the $SU(N)$ result by imposing the traceless condition, $\sum_a \phi_a = 0$. We always take:

$$\phi_a = \varphi_a - \varphi_{a-1} , \quad a = 1, \cdots, N , \qquad \text{with} \qquad \varphi_0 = \varphi_N = 0 . \tag{2.31}$$

In the M-theory construction, we will see that, for each $SU(N)$ gauge group, there are $N$ "elementary" instanton particles $\mathcal{I}_a$ ($a = 1, \cdots, N$) with instanton charge 1. Their masses are conveniently given by the second derivatives of (2.30), that is:

$$M(\mathcal{I}_a) = \left. \frac{\partial^2 \mathcal{F}^{U(N)}}{\partial \phi_a^2} \right|_{\phi = \phi(\varphi)} = h_0 + \cdots . \tag{2.32}$$

Thus, we first compute the "effective $U(1)_a$ gauge coupling" on the Coulomb branch of the $U(N)$-quiver theory, and then impose the $SU(N)$ tracelessness conditions (2.31). This heuristic prescription will be natural from the point of view of our string-theory construction. It would be interesting to give a fully gauge-theoretic derivation of the instanton particle masses.

For each $SU(N_s)$ gauge group in a given quiver theory, the instanton particle $\mathcal{I}_s = \mathcal{I}_{s,a}$ with the lowest mass can be viewed as "the" elementary instanton, while any other instanton particle $\mathcal{I}_{s,b}$, $b \neq a$, can be interpreted as a marginal bound state of $\mathcal{I}_s$ with other (W-boson and hypermultiplet) BPS particles [5].

---

[12]In our string-theory construction, we will restrict ourselves to $|k| < N$. For $N = 2$, the $U(2)$ CS level $k = 0$ or $\pm 1$ will correspond to the $\mathbb{Z}_2$-valued $SU(2)$ $\theta$-angle $\theta = 0$ or $\pm\pi$ [4].

## 2.6 Physical chambers on the Coulomb branch

Consider the Coulomb branch $\mathcal{M}^C(\mu)$ of the gauge theory at some fixed value of the masses $\mu = (h_0, m)$. There can be various real codimension-one "walls" on the Coulomb branch, at which locations particles become massless:

$$M(\varphi; \mu) \to 0 \, . \tag{2.33}$$

These walls divide the moduli space into different chambers, characterized by distinct BPS particle spectra. Note that the central charge is real in 5d, hence there can be no marginal-stability walls—only threshold bound-states are allowed among BPS particles. Since the particle masses are linear in $\varphi$, the walls are hyperplanes in $\mathbb{R}^r \cong \{\varphi_a\}$.

From the gauge-theory perspective, we have two kinds of "perturbative" walls, depending on the state that becomes massless:

(i) **An hypermultiplet mode $\mathcal{H}$ becomes massless.** Such a wall is "traversable"—the theory on the other side simply contains an hypermultiplet with the opposite sign of the real mass $M(\mathcal{H})$, with central charge $Z = |M(\mathcal{H})|$. The prepotential is non-smooth across such a wall, as is clear from (2.19).

(ii) **A W-boson becomes massless.** Such a "hard wall" is at the boundary of the Weyl chamber and it is not traversable. [13]

Thus, the "naive" field-theory Coulomb branch consists of a fundamental Weyl chamber of the gauge group **G**, which is further subdivided into "field theory chambers" separated by walls where hypermultiplets go massless.

On the other hand, the low-energy effective field theory on the Coulomb branch is clearly valid only if *all* BPS states, whether perturbative or not, are safely massive. Thus, we should also worry about two other kinds of walls, where:

(iii) **A BPS instanton particle becomes massless.** Such a wall is harder to describe in purely field-theory terms. It might be traversable or not, depending on the theory. We will see examples of this in our study of classic examples.

(iv) **A magnetic string become tensionless,** $T(\varphi; \mu) \to 0$**.** Note that $T(\varphi; \mu) = 0$ is a quadratic equation in $\varphi$, therefore such "magnetic walls" are rather more peculiar.

The existence of these non-perturbative walls was pointed out in [22], where the notion of a *physical Coulomb branch* was introduced. The physical Coulomb branch of a 5d $\mathcal{N} = 1$ gauge theory is the subspace of the "naive" Coulomb branch within the bounds of both hard instanton walls and magnetic walls. [14]

# 3 Type IIA perspective and gauge-theory phases

In this section, we review the M-theory approach to 5d SCFTs, we give a pedagogical introduction to some useful toric geometry tools, and we discuss in detail a new and complementary type-IIA perspective.

---

[13]Indeed, "going through" a Weyl chamber one obtains a gauge-equivalent description, therefore we cannot cross such a wall anymore than we can cross through a mirror. We should fix the gauge once and for all and consider a single fundamental Weyl chamber.

[14]The main focus of [22] was on the strong-coupling limit $\mu \to 0$, while in this work we are interested in the full parameter space.

## 3.1   5d SCFTs from M-theory on a $CY_3$ singularity

The starting point of our analysis is the geometric engineering of the 5d field theory from M-theory on a local Calabi-Yau three-fold ($CY_3$) $\mathbf{X}$, an isolated canonical singularity. It is believed that $\mathbf{X}$ defines an SCFT $\mathcal{T}_{\mathbf{X}}$ on the transverse space:

$$\text{M-theory on } \mathbb{R}^{1,4} \times \mathbf{X} \qquad \leftrightarrow \qquad \mathcal{T}_{\mathbf{X}} \quad \text{SCFT on } \mathbb{R}^{1,4+\varepsilon}. \tag{3.1}$$

If the singularity $\mathbf{X}$ is elliptic, then $\varepsilon = 1$ and $\mathcal{T}_{\mathbf{X}}$ is a 6d SCFT; [15] otherwise, $\varepsilon = 0$ and $\mathcal{T}_{\mathbf{X}}$ is a 5d SCFT. In this paper, we focus on that latter case.

A generic $CY_3$ singularity $\mathbf{X}$ can be smoothed out by a crepant resolution: [16]

$$\pi : \widehat{\mathbf{X}} \to \mathbf{X}, \qquad \pi^* K_{\mathbf{X}} = K_{\widehat{\mathbf{X}}}, \tag{3.2}$$

giving us a smooth local CY threefold $\widehat{\mathbf{X}}$. The exceptional set $\pi^{-1}(0)$ (with $0 \in \mathbf{X}$ the isolated singularity) contains $n_4 \equiv r \geq 0$ compact divisors. The non-negative integer $r$ is called the "rank" of $\mathbf{X}$. It corresponds to the real dimension of the SCFT Coulomb branch:

$$r = \dim \mathcal{M}_{\mathcal{T}_{\mathbf{X}}}^C. \tag{3.3}$$

The resolved space $\widehat{\mathbf{X}}$ also contains compact curves, denoted by $\mathcal{C}$, which may intersect the exceptional divisors non-trivially.

The dictionary between such a smooth geometry and a 5d $\mathcal{N} = 1$ field theory can be established, in principle, as a decoupling limit, starting from a M-theory compactification on compact CY threefold $Y$ (see *e.g.* [21, 80–82]), and scaling the volume of $Y$ to infinity while the volumes of a collection of holomorphic 2-cycles and of holomorphic 4-cycles that are *intersecting* within $Y$ are kept finite; this has the effect of sending the five-dimensional Planck mass to infinity, thus decoupling gravity. We require the collection of 2- and 4- cycles to be intersecting because we are interested in obtaining an interacting SCFT. This gives rise to the local model $\widehat{\mathbf{X}}$.

Given the local CY threefold $\widehat{\mathbf{X}}$, we pick a basis $\mathcal{C}^{\mathbf{a}}$ of compact holomorphic 2-cycles in $H_2(\widehat{\mathbf{X}}, \mathbb{Z})$. Let $n_2 = r + f$ denote the dimension of $H_2(\widehat{\mathbf{X}}, \mathbb{Z})$, with $r$ the rank and $f \geq 0$ some non-negative integer. The two-cycles $\mathcal{C}^{\mathbf{a}}$ are dual to either compact divisors (in the exceptional set) or non-compact divisors. Let us denote by $D_k$ the divisors, compact or not, and let us choose some basis of $n_2$ divisors $\{D_k\}_{k=1}^{n_2}$ such that:

$$\mathcal{C}^{\mathbf{a}} \cdot D_k = \mathbf{Q}^{\mathbf{a}}{}_k, \qquad \det \mathbf{Q} \neq 0. \tag{3.4}$$

Let $J$ denote the Kähler form and let $S$ denote the (Poincaré dual) Kähler class, which is a particular linear combination of divisors over the real numbers:

$$S = \sum_{k=1}^{n_2} \lambda^k D_k = \sum_{j=1}^{f} \mu^j D_j + \sum_{a=1}^{r} \nu^a \mathbf{E}_a. \tag{3.5}$$

Here, we split the set $\{D_k\}$ into $r$ compact divisors, denoted by $\mathbf{E}_a$, and $f$ non-compact divisors, denoted by $D_j$. The Kähler volumes of the compact curves in $\widehat{\mathbf{X}}$ are given by:

$$\xi^{\mathbf{a}}(\mu, \nu) \equiv \int_{\mathcal{C}^{\mathbf{a}}} J = \mathcal{C}^{\mathbf{a}} \cdot S = \mathbf{Q}^{\mathbf{a}}{}_k \lambda^k = \mathbf{Q}^{\mathbf{a}}{}_j \mu^j + \mathbf{Q}^{\mathbf{a}}{}_a \nu^a > 0. \tag{3.6}$$

---

[15] Shrinking the elliptic fiber to zero size uplifts M-theory to F-theory and gives rise to a 6d compactification [72] — see e.g. [73–79] for work in the context of characterizing the singular geometries for 6d SCFTs.

[16] One can also consider complex deformations of the singularity, which characterize the Higgs branch of the SCFT [2, 4]. In this paper, we focus on crepant resolutions and the corresponding Coulomb-branch physics.

Therefore, the parameters $\mu^k \in \mathbb{R}$ and $\nu^a \in \mathbb{R}$ in (3.5) are essentially the Kähler moduli of two-cycles dual to non-compact and compact four-cycles, respectively.

Let us briefly review the correspondence between this geometric structure and the low-energy physics of the SCFT $\mathcal{T}_{\mathbf{X}}$ and of its massive deformations [2,4]. For generic values of the Kähler parameters, the low-energy $\mathcal{N} = 1$ field theory is an abelian theory with gauge group $U(1)^r \cong H^2(\widehat{\mathbf{X}}, \mathbb{R})/H^2(\widehat{\mathbf{X}}, \mathbb{Z})$. The $U(1)$ gauge fields arise from the periods of the M-theory three-form over the curves $\mathcal{C}_a$ dual to compact divisors, in the obvious way. Moreover, the exact prepotential for this abelian theory can be computed from the geometry, as:

$$\mathcal{F}(\mu, \nu) = -\frac{1}{6} \int_{\widehat{\mathbf{X}}} J \wedge J \wedge J = -\frac{1}{6} S \cdot S \cdot S . \tag{3.7}$$

This prepotential is fully determined by the triple-intersection numbers of $\widehat{\mathbf{X}}$, up to some regularization that is needed to compute the triple-intersection of three non-compact divisors; this introduces some ambiguity in (3.7) which, however, only affects the $\nu$-independent part of $\mathcal{F}(\mu, \nu)$. Since those "constant terms" are non-physical for the 5d field theory in flat space, we can mostly ignore them—see Appendix B for further discussion. The Kähler parameters $\mu$ and $\nu$ in (3.5) are the mass parameters and the dynamical fields, respectively. The mass parameters $\mu_i$ correspond to deformations of the SCFT by dimension-four operators:

$$S_{\mathcal{T}_{\mathbf{X}}} \to S_{\mathcal{T}_{\mathbf{X}}} + \mu^j \int d^5 x \, \mathcal{O}_j . \tag{3.8}$$

The operators $\mathcal{O}_j$ sit at level two in short multiplets $C_1[0,0]_3^{(2)}$ of the $\mathfrak{f}(4)$ superconformal algebra [83].

The dynamical fields $\nu$, on the other hand, are the Coulomb branch parameters. At $\mu = 0$, they correspond to real VEVs of SCFT operators, thus spanning the intrinsic Coulomb branch $\mathcal{M}_{\mathcal{T}_{\mathbf{X}}}^C$ of $\mathcal{T}_{\mathbf{X}}$. The Coulomb phase of a 5d SCFT can be defined as the branch of its vacuum moduli space where the $SU(2)_R$ symmetry is unbroken. In particular, we expect to flow to a Coulomb phase by giving VEVs to scalars that are singlets with respect to the $SU(2)_R$ symmetry. However, since the Coulomb branch is a real manifold, one do not expect to have an underlying Coulomb branch chiral ring (as opposed, for instance, to the case of 4d $\mathcal{N} = 2$ SCFTs). From the classification of protected unitary representations of $\mathfrak{f}(4)$ in [84], one can see that few (if any) BPS multiplets have the desired features to be Coulomb branch operators, while many non-BPS multiplets do. Hence, one expects that the Coulomb branch corresponds to VEVs of SCFT operators that sit in long multiplets, which are not protected. [17]

More generally, $\mu$ and $\nu$ can be both non-zero, and the Coulomb branch $\mathcal{M}_{\mathcal{T}_{\mathbf{X}}}^C(\mu)$ is fibered over the parameter space $\{\mu\}$,

$$\mathcal{M}_{\mathcal{T}_{\mathbf{X}}}^C(\mu) \to \mathcal{P}_{\mathcal{T}_{\mathbf{X}}} \to \{\mu\} , \tag{3.9}$$

as discussed in the introduction.

The 5d theory corresponding to $\widehat{\mathbf{X}}$ admits BPS excitations consisting of electrically charged BPS particles and (dual) BPS magnetic strings. In geometry, these are realized by M2-branes wrapping the holomorphic curves $\mathcal{C}^{\mathbf{a}}$, and by M5-branes wrapping holomorphic surfaces $\mathbf{E}_a$, respectively. The masses of the particles are given by the Kähler volumes (3.6). The tensions of the magnetic strings are given by the Kähler volumes of the compact divisors, which coincide with the first derivatives of the prepotential:

$$T_a(\mu, \nu) \equiv -\partial_{\nu^a} \mathcal{F}(\mu, \nu) = \frac{1}{2} \int_{\mathbf{E}_a} J \wedge J = \text{vol}(\mathbf{E}_a) . \tag{3.10}$$

---

[17]We are grateful to Thomas Dumitrescu for illuminating correspondence on this point.

Finally, the metric of the Coulomb branch is given by:

$$\tau_{ab}(\mu, \nu) = \partial_{\nu^a} \partial_{\nu^b} \mathcal{F}(\mu, \nu) = \mathrm{vol}(\mathbf{E}_a \cdot \mathbf{E}_b) \,, \tag{3.11}$$

which is the volume of a curve $\mathcal{C}_{ab} = \mathbf{E}_a \cdot \mathbf{E}_b$ at the intersection of two exceptional divisors.

**The extended parameter space.** The above discussion focussed on a particular resolution $\widehat{\mathbf{X}}$ of the singularity $\mathbf{X}$. In general, there can be many distinct, birationally equivalent local threefolds $\widehat{\mathbf{X}}_\ell$, which all have the same singular limit $\mathbf{X}$. For a given $\widehat{\mathbf{X}}$, the Kähler cone of the singularity $\mathbf{X}$ is the set of all positive Kähler forms:

$$\mathcal{K}(\widehat{\mathbf{X}} \backslash \mathbf{X}) = \{J \mid \mathcal{C} \cdot S > 0 \text{ for all holomorphic curves } \mathcal{C} \subset \widehat{\mathbf{X}}\} \,. \tag{3.12}$$

The extended Kähler cone is the closure of the union of all compatible Kähler cones,

$$\widehat{\mathcal{K}}(\mathbf{X}) = \left\{ \bigcup_\ell \mathcal{K}(\widehat{\mathbf{X}}_\ell \backslash \mathbf{X}) \right\}^c \,. \tag{3.13}$$

The extended Kähler cone is a fan, with pairs of Kähler cones glued along common faces in the interior of $\widehat{\mathcal{K}}(\mathbf{X})$. The boundaries of $\mathcal{K}(\widehat{\mathbf{X}}_\ell \backslash \mathbf{X})$ correspond to loci where the 3-fold $\widehat{\mathbf{X}}_\ell$ develops a singularity. The interior boundaries are regions where a holomorphic curve collapses to zero volume and formally develops negative volume in the adjacent Kähler cone, signaling a flop transition. This corresponds to a BPS particle becoming massless, which triggers a jump in the third derivatives of the prepotential (3.7). By contrast, the exterior boundaries of $\widehat{\mathcal{K}}(\mathbf{X})$ are loci where one of the 4-cycles $\mathbf{E}_a$ can collapse to a 2-cycle or a point. The SCFT point is the origin of $\widehat{\mathcal{K}}(\mathbf{X})$, and corresponds to the singularity $\mathbf{X}$, which is characterized by the connected union of 4-cycles shrinking to a point. All the Coulomb branch VEVs and the massive deformations of the 5d SCFTs obtained from M-theory are encoded in the extended Kähler moduli space of $\mathbf{X}$ [47]:

$$\mathcal{P}_{\mathcal{T}_\mathbf{X}} = \widehat{\mathcal{K}}(\mathbf{X}) \,, \tag{3.14}$$

where $\mathcal{P}_{\mathcal{T}_\mathbf{X}}$ is the "extended parameter space" of the theory $\mathcal{T}_\mathbf{X}$, including all the VEVs and mass parameters. In a sense, the collection of all crepant resolutions of $\mathbf{X}$ provides the 5d analogue of the Seiberg-Witten geometry in 4d $\mathcal{N} = 2$ theory, namely, the smooth M-theory geometry $\widehat{\mathbf{X}}$ corresponding to one such resolution gives a solution to the 5d theory in its Coulomb phase, as captured by the exact prepotential (3.7).

**Gauge-theory phases.** At scales much smaller than the scale set by $|\mu|$, we often have useful field-theory descriptions of $\mathcal{M}^C_{\mathcal{T}_\mathbf{X}}(\mu)$ in (3.9) as the Coulomb branch of a 5d $\mathcal{N} = 1$ gauge theory; in that case, some deformation parameters $\mu_s = h_{0,s}$ correspond to super-Yang-Mills terms in the low-energy description.

When that happens, one can check the correspondence between geometry and field theory by matching the geometric prepotential (3.7) to the one-loop-exact gauge-theory prepotential (2.19), namely:

$$\mathcal{F}(\mu, \nu) = \mathcal{F}(h_0, m, \varphi) \,, \tag{3.15}$$

for some linear map between the Kähler parameters and the field-theory parameters. Surprisingly, in the literature so far, the matching (3.15) has mostly been checked (in numerous examples) in the limit $\mu^j = 0$ (in which case we simply have $\nu_a = -\varphi_a$, in our conventions). This corresponds to a strong-coupling limit in the gauge theory, where the low-energy approximation breaks down. It is also a somewhat degenerate subspace of the extended Kähler cone (3.13). Remarkably, one finds a perfect match nonetheless [2, 4], which probably signals some kind of non-renormalization theorem at work.

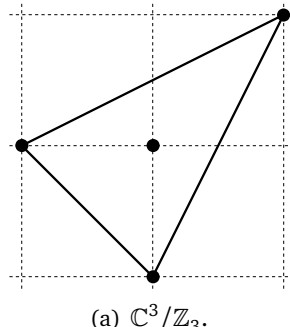

(a) $\mathbb{C}^3/\mathbb{Z}_3$.

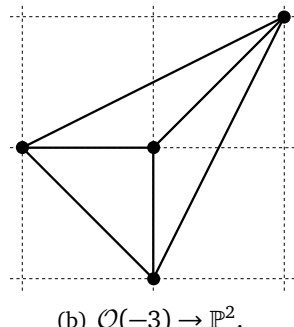

(b) $\mathcal{O}(-3) \to \mathbb{P}^2$.

Figure 3: Toric diagram for the $\mathbb{C}^3/\mathbb{Z}_3$ singularity, and its crepant resolution. In this example, there are three external points $w_1 = (-1, 0)$, $w_2 = (0, -1)$, $w_3 = (1, 1)$ ($n_E = 3$), one internal point $w_0 = (0, 0)$ ($r = 1$), and a unique triangulation of the toric diagram.

One of the aims of this paper is to check (3.15) with generic parameters turned on. Another objective is to explore when, if at all, we can have a *non-abelian* gauge-theory description of the low-energy physics (what we call a "gauge-theory phase"). The standard approach [4, 23] is to look for a "ruling" of the exceptional set $\pi^{-1}(0) \subset \widehat{\mathbf{X}}$—that is, we look for a set of surfaces $\mathbf{E}$ which can be written as a fibration $\mathbb{P}^1 \to \mathbf{E} \to \mathcal{C}$ over a curve $\mathcal{C}$ with $\mathbb{P}^1$ fibers. The M2-branes wrapped over the fibers are identified with the W-bosons, which become massless in a limit to a boundary of the Kähler cone where the rule surfaces shrink to zero size. Instead of following this purely geometric approach, we will propose a closely related description of the gauge-theory phases by exploiting the M-theory/type IIA duality.

In what follows, we restrict ourselves to the case of $\mathbf{X}$ a *toric* singularity, which simplifies the analysis significantly.

## 3.2 CY$_3$ singularity, toric diagram and GLSM

Let us now review some toric geometry tools that we will use extensively below. The toric CY$_3$ singularity $\mathbf{X}$ is an affine toric variety. It is determined by its toric cone $\Delta_0$, a set of $n_E$ vectors $v_i$ in $\mathbb{Z}^3$ which generate a strongly convex rational polyhedral cone. The Calabi-Yau condition implies that all the $v_i$'s are co-planar. In that case, we can perform an $SL(3, \mathbb{Z})$ transformation to bring the toric vectors onto the plane $v^z = 1$, namely:

$$v_i = (w_i, 1), \quad i = 1, \cdots, n_E, \qquad w_i = (w_i^x, w_i^y) \in \mathbb{Z}^2. \tag{3.16}$$

The vectors $w_i$ give us the toric diagram of the singularity, $\Gamma \in \mathbb{Z}^2$. A simple example is shown in Figure 3. The toric diagram also has a number $r \geq 0$ of internal points.

The toric singularity itself can be pictured as a $T^3 \cong U(1)^3$ fibration over the dual cone $\Delta_0^\vee$. The toric vectors $v_i$ are outward-pointing orthogonal to the external facets $F_i \subset \Delta_0^\vee$, corresponding to which $U(1)_i \subset U(1)^3$ degenerates at each facet, with $U(1)_i \times U(1)_j$ degenerating at the edge $E_{ij} = F_i \cap F_j$. Note that each facet corresponds to non-compact four-cycle (the $T^2$ fibration over $F_i$), which is a toric divisor denoted by $D_i$. Each edge $E_{ij} \subset \Delta_0^\vee$ corresponds to a non-compact two-cycle (the $U(1)$ fibration over $E_{ij}$) at the intersection of two toric divisors, denoted by $\mathcal{C}_{ij} \cong D_i \cdot D_j$.

The resolved CY$_3$ $\widehat{\mathbf{X}}$ is obtained by subdividing the toric cone $\Delta_0$ into a more general toric fan $\Delta$. At the level of the toric diagram $\Gamma$, this correspond to including the internal points:

$$v_a = (w_a, 1), \quad a = 1, \cdots, r, \tag{3.17}$$

and to choosing a triangulation of $\Gamma$. A maximal resolution corresponds to a particular complete triangulation of $\Gamma$. Any two maximal resolutions are related by a sequence of flops.

### 3.2.1 GLSM description

We may also describe the toric singularity and its crepant resolutions as a Kähler quotient:

$$\widehat{\mathbf{X}} \cong \mathbb{C}^n //_\xi U(1)^{n-3} \,, \qquad n \equiv n_E + r \,. \tag{3.18}$$

The advantage of this perspective is that it keeps track of the Kähler parameters, $\xi$, which determine the Kähler volumes of the exceptional curves. This construction can be nicely presented, in the physics language, as a "gauged linear sigma-model" (GLSM) [85], with homogeneous coordinates $\{z_i\} \in \mathbb{C}^n$ (with index $i = 1, \cdots, n$) and Kähler parameters $\xi_{\mathbf{a}}$ (with index $\mathbf{a} = 1, \cdots, n-3$) for the $U(1)_{\mathbf{a}}$ actions by which we quotient. The parameters $\xi_{\mathbf{a}}$ are also known as Fayet-Iliopololous (FI) parameters for the auxiliary gauge groups $U(1)_{\mathbf{a}}$. The GLSM data is conveniently summarized by a table:

$$\begin{array}{c|c|c} & z_i & \text{FI} \\ \hline U(1)_{\mathbf{a}} & Q_i^{\mathbf{a}} & \xi_{\mathbf{a}} \end{array} \,, \qquad \text{span}(Q_i^{\mathbf{a}}) = \ker(v_1, \cdots, v_n) \,. \tag{3.19}$$

Here, the charges $Q_i^{\mathbf{a}} \in \mathbb{Z}$ are the $U(1)_{\mathbf{a}}$ charges of the homogenous coordinates $z_i$. The Kähler quotient (3.18) is given explicitly by:

$$\widehat{\mathbf{X}} \cong \left\{ z_i \,\Big|\, \sum_i Q_i^{\mathbf{a}} |z_i|^2 = \xi_{\mathbf{a}} \right\} / U(1)^{n-3} \,. \tag{3.20}$$

Each $z_i$ is associated to a point $w_i \in \Gamma$ in the toric diagram. The corresponding toric divisor is defined by:

$$D_i \cong \{z_i = 0\} \cap \widehat{\mathbf{X}} \,. \tag{3.21}$$

We will also often use the following notation, as in subsection 3.1, which distinguishes between non-compact and compact toric divisors, corresponding to the external and internal points in the toric diagram, respectively:

$$D_j \,, \quad j = 1, \cdots, n_E \,, \qquad \mathbf{E}_a \,, \quad a = 1, \cdots, r \,. \tag{3.22}$$

The charge vectors $Q^{\mathbf{a}}$ are related to the toric vectors $v_i \in \Delta$ (including all the internal points in the toric diagram) by:

$$\sum_{i=1}^n v_i Q_i^{\mathbf{a}} = 0 \,. \tag{3.23}$$

In particular, $\sum_i Q_i^{\mathbf{a}} = 0$ is the condition for the GLSM target space to be Calabi-Yau. These relations also imply three linear relations amongst toric divisors, including the compact ones:

$$\sum_{i=1}^n w_i^x D_i \cong 0 \,, \qquad \sum_{i=1}^n w_i^y D_i \cong 0 \,, \qquad \sum_{i=1}^n D_i \cong 0 \,. \tag{3.24}$$

### 3.2.2 Triangulation, curves and intersection numbers

Given a fully triangulated toric diagram $\Gamma$, there is a convenient way to write down a (redudant) GLSM and to compute all the triple intersections numbers amongs divisors. Let us consider the compact curve:

$$\mathcal{C}_{ij} \cong D_i \cdot D_j \cong \{z_i = 0, z_j = 0\} \cap \widehat{\mathbf{X}} \,. \tag{3.25}$$

It corresponds to an internal line in the toric diagram, denoted by $E_{ij}$. This $E_{ij}$ is at the intersection of two triangles with vertices $w_i, w_j, w_k$ and $w_i, w_j, w_l$, respectively. We can then write a row of the GLSM as:

$$
\begin{array}{c|ccccc|c}
 & z_i & z_j & z_k & z_l & z_{m \neq i,j,k,l} & \\
\hline
\mathcal{C}_{ij} & q_i & q_j & 1 & 1 & 0 & \xi_{\mathcal{C}_{ij}}
\end{array} \ , \qquad v_i q_i + v_j q_j + v_k + v_l = 0 \ . \tag{3.26}
$$

Here and henceforth, it will be convenient to label the GLSM fields $z_i$ by the corresponding toric divisors, and the rows by the corresponding curves. By construction, the FI term $\xi_{\mathcal{C}_{ij}}$ is the Kähler volume of the curve in $\widehat{\mathbf{X}}$, which is positive:

$$
\xi_{ij} = \int_{\mathcal{C}_{ij}} J > 0 \ . \tag{3.27}
$$

Repeating this operation for every internal line $E_{ij}$ in $\Gamma$, we obtain a redundant GLSM that capture all the exceptional curves. We can reduce that redundant description by choosing a set of $n-3$ linearly independent curves $\mathcal{C}_{\mathbf{a}}$, giving us a proper GLSM:

$$
\begin{array}{c|c|c}
 & D_i & \text{FI} \\
\hline
\mathcal{C}^{\mathbf{a}} & Q_i^{\mathbf{a}} & \xi_{\mathbf{a}}
\end{array} \ . \tag{3.28}
$$

This is obviously equivalent to (3.19), but here we have chosen a particular basis for the $U(1)^{n-3}$ gauge group, adapted to a given triangulated toric diagram, such that $\xi_{\mathbf{a}} > 0$, $\forall \mathbf{a}$. We loosely refer to $\{\mathcal{C}^{\mathbf{a}}\}$ as the "generators of the Mori cone." The intersections numbers between divisors and curves are simply given by:

$$
D_i \cdot \mathcal{C}^{\mathbf{a}} = Q_i^{\mathbf{a}} \ . \tag{3.29}
$$

Combining this with (3.25) and the linear relations (3.26), we can compute the triple intersection numbers amongst divisors:

$$
D_i \cdot D_j \cdot D_k \ , \tag{3.30}
$$

where at least one divisor is compact. In the following, we use this simple method to compute the M-theory prepotential (3.7) in numerous examples.

Finally, one might wonder whether one can make sense of the triple intersection number (3.30) when all three divisors are non-compact. It is not well-defined by itself, but we shall briefly discuss a possible regularization in Appendix B.

**Example.** As a simple example, consider the resolved $\mathbb{C}^3/\mathbb{Z}_3$ orbifold of Figure 3. We have a curve $\mathcal{C} \cong \mathcal{C}_{10}$, and the two triangles have vertices $w_1, w_0, w_2$ and $w_1, w_0, w_3$. Then, we have (3.26) with $q_1 = 1$ and $q_0 = -3$, so that:

$$
\begin{array}{c|cccc|c}
 & D_1 & D_2 & D_3 & \mathbf{E}_0 & \\
\hline
\mathcal{C} & 1 & 1 & 1 & -3 & \xi_{\mathcal{C}_{10}}
\end{array} \ . \tag{3.31}
$$

Here, we reordered the divisors in the standard order, and used the notation $D_0 = \mathbf{E}_0$. We can also check that $\mathcal{C}_{20} \cong \mathcal{C}_{30} \cong \mathcal{C}$. This is the standard GLSM description of a local $\mathbb{P}^2$, with $\mathcal{C} \cong H$ the hyperplane class. We also have the linear relations $D_1 \cong D_2 \cong D_3$ amongst toric divisors. The triple intersection numbers are:

$$
D_1^2 \mathbf{E}_0 = \mathcal{C} D_1 = 1 \ , \qquad D_1 \mathbf{E}_0^2 = \mathcal{C} \mathbf{E}_0 = -3 \ , \qquad \mathbf{E}_0^3 = -3 D_1 \mathbf{E}_0^2 = 9 \ . \tag{3.32}
$$

We then have $S = v \mathbf{E}_0$ and the M-theory prepotential $\mathcal{F} = -\frac{1}{6} S^3 = -\frac{3}{2} v^3$.

## 3.3 Toric threefold $\widehat{\mathbf{X}}$ as a $U(1)_M$ fibration

Now, let us pick a $U(1)_M$ inside the $U(1)^3$ toric action. We would like to view the full $\widehat{\mathbf{X}}$ as a $U(1)_M$ fibration over a five-dimensional space $\mathcal{M}_5$:

$$U(1)_M \longrightarrow \widehat{\mathbf{X}} \longrightarrow \mathcal{M}_5 \, . \tag{3.33}$$

Viewing $U(1)_M$ as the M-theory circle, we then have a type-IIA description. Not every $U(1)_M$ gives us a well-understood IIA configuration, however. In the following, we discuss this $U(1)_M$ fibration structure and the conditions for an "allowed" IIA reduction. This approach was first introduced in [51] and developed in [52–54,86,87] in the context of M-theory on CY fourfold singularities.

**GLSM reduction.** Let us first discuss the circle fibration structure (3.33) from the point of the view of the GLSM [51]. We consider the following non-standard parameterization of the GLSM (3.19), as:

$$\begin{array}{c|c|c} & z_i & \text{FI} \\ \hline U(1)_{\mathbf{a}} & Q_i^{\mathbf{a}} & \xi_{\mathbf{a}} \\ \hline U(1)_M & Q_i^M & r_0 \end{array} \, , \qquad \sum_{i=1}^{n} Q_i^M = 0 \, . \tag{3.34}$$

Here, we introduced an additional complex parameter $r_0 + i\theta_0$, together with a new gauge symmetry:

$$\sum_{i=1}^{n} Q_i^M |z_i|^2 = r_0 \, , \qquad \theta_0 \sim \theta_0 + \alpha_M \, , \qquad z_i \sim e^{i\alpha_M Q_i^M} z_i \, . \tag{3.35}$$

This construction describes the same three-fold $\widehat{\mathbf{X}}$, since we can solve for $r_0$ using (3.35) and gauge fix $\theta_0$ to zero. On the other hand, we may consider the further projection of the $\text{CY}_3$ to a CY *two-fold* by "forgetting" $\theta_0$ and fixing $r_0$. In this picture, we interpret $\theta_0$ as the M-theory circle coordinate. At fixed $r_0$, the GLSM (3.34) is a redundant description of a toric $\text{CY}_2$ variety, denoted by $\widehat{\mathbf{Y}}$. The parameter $r_0 \in \mathbb{R}$ is itself interpreted as the $x^9$ coordinate in type IIA string theory, over which $\widehat{\mathbf{Y}}$ is fibered:

$$\widehat{\mathbf{Y}}(r_0) \longrightarrow \mathcal{M}_5 \longrightarrow \mathbb{R} \cong \{r_0\} \, . \tag{3.36}$$

The only two-dimensional toric CY singularity is the *A*-type orbifold:

$$A_{M-1} \cong \mathbb{C}^2/\mathbb{Z}_M \, , \tag{3.37}$$

which can be resolved to the ALE space $\widehat{\mathbf{Y}}$. The corresponding one-dimensional toric diagram is simply a line of $M + 1$ points. Let us denote by $t_k$, $k = 0, \cdots, M$, the corresponding GLSM fields. We have:

$$\begin{array}{c|c|c} & t_k & \text{FI} \\ \hline U(1)_s & -2\delta_{ks} + \delta_{k,s+1} + \delta_{k,s-1} & \chi_s(r_0) \end{array} \, , \qquad s = 1, \cdots, M-1 \, . \tag{3.38}$$

At any fixed $r_0$, we can derive the $\text{CY}_2$ GLSM (3.38) from (3.34), by eliminating the redundant $z_i$ variables using the D-term constraints (3.35). This defines a projection map:

$$z_i \mapsto t_k = z_k(r_0) \, , \tag{3.39}$$

where $t_k$ denotes the independent $z_i$ variables. The Kähler parameters $\chi_s(r_0)$ of $\widehat{\mathbf{Y}}$ are the volumes of the $\mathbb{P}^1$'s in the ALE resolution:

$$\chi_s(r_0) = \int_{\mathbb{P}^1_s} J_{\widehat{\mathbf{Y}}} \, . \tag{3.40}$$

Note that the toric divisors $D_k^{\widehat{\mathbf{Y}}} \cong \{t_k = 0\}$ of $\widehat{\mathbf{Y}}$ for $k = 1, \cdot, M - 1$ are precisely the exceptional curves:

$$D_s^{\widehat{\mathbf{Y}}} \cong \mathbb{P}_s^1 \,. \tag{3.41}$$

They intersect amongst themselves according to the $A_{M-1}$ Dynkin diagram, as is clear from (3.38). The toric divisors $D_0^{\widehat{\mathbf{Y}}}$ and $D_M^{\widehat{\mathbf{Y}}}$ are non-compact, on the other hand.

The M-theory circle is non-trivially fibered over $\mathcal{M}_5$. By the M-theory/IIA duality, this implies the presence of non-trivial RR two-form flux $F_2^{\mathrm{RR}} = dC_1$ in type IIA. Moreover, whenever the M-theory circle degenerates in real-codimension four, there is a magnetic source for $C_1$ in type IIA, namely a number of D6-branes.

**Toric diagram reduction.** We can obtain the one-dimensional toric diagram $\Gamma_{\widehat{\mathbf{Y}}}$ of $\widehat{\mathbf{Y}}$ from the two-dimensional toric diagram $\Gamma$ of $\widehat{\mathbf{X}}$ by a simple projection. The $U(1)_M$ charges $Q_i^M$ introduced in (3.34) do not satisfy $\sum_i v_i Q_i^M = 0$. Instead, let us define:

$$v_M \equiv \text{primitive} \sum_{i=1}^n v_i Q_i^M \,, \tag{3.42}$$

the primitive vector in $\mathbb{Z}^3$ parallel to $\sum_{i=1}^n v_i Q_i^M$. Then, the toric vectors $\widetilde{v}_k \in \mathbb{Z}^2$ of $\widehat{\mathbf{Y}}$ are obtained by projecting the toric vectors $v_i$ to the plane orthogonal to $v_M$. (In general, there will be several $v_i$'s mapping to a single $\widetilde{v}_k$. This corresponds to the map $z_i \mapsto t_i$ in the GLSM.) Due to the CY condition, we must have $v_M = (w_M, 0)$. It is convenient to choose:

$$v_M = (0, 1, 0) \,. \tag{3.43}$$

Reducing along this particular $U(1)_M \subset U(1)^3$ corresponds to a "vertical reduction" of the toric diagram; an example of this is shown in Figure 4(a). Of course, we may (and shall) also consider the "horizontal reduction" of $\Gamma$, or any other projection related to $v_M$ by an $SL(2, \mathbb{Z})$ transformation of the toric diagram.

Given the $\mathrm{CY}_3$ toric variety $\widehat{\mathbf{X}}$, we can understand how the exceptional divisors $\mathbf{E}_j$ and the 2-cycles $\mathcal{C}_{ij}$ (compact or non-compact) are projected onto $\mathcal{M}_5$ by the vertical reduction along $v_M$. Let us consider the 2-cycle $\mathcal{C}_{ij}$ corresponding to the edge $E_{ij}$ (internal or external) in $\Gamma$. We define:

$$\Delta v \equiv v_j - v_i = (m, n, 0) \,. \tag{3.44}$$

Without loss of generality, we choose $n > 0$. Along $\mathcal{C}_{ij}$, the $T_{(ij)}^2$ given by $U(1)_i \times U(1)_j \subset T^3$ degenerates, while the orthogonal $U(1)_\varphi \subset T^3$ is the local angular coordinate on $\mathcal{C}_{ij}$. [18] Note that $T_{(ij)}^2$ corresponds to the span of $\Delta v$ and $(0, 0, 1)$. There are three cases:

(i) **Vertical edge:** If $\Delta v = (0, n, 0)$, we have a vertical edge of length $n$ on the toric diagram, which is really a set of $n$ edges of length one, each corresponding to a distinct 2-cycle $\mathcal{C}_{ij}^{(q)}$, $q = 1, \cdots, n$ with angular direction $U(1)_\varphi$ corresponding to $v_\varphi = (1, 0, 0)$. Since $\Delta v$ is parallel to $v_M$, the M-theory circle degenerates in real-codimension four along each $\mathcal{C}_{ij}^{(q)}$, corresponding to a D6-brane wrapping a curve in type IIA [88]. Thus, each 2-cycle $\mathcal{C}_{ij}^{(q)}$ in $\widehat{\mathbf{X}}$ projects to a D6-brane wrapping a single 2-cycle $D^{\widehat{\mathbf{Y}}}$ in $\widehat{\mathbf{Y}}$, but generally at different values of $r_0$. The D6-branes can be compact (wrapping a $\mathbb{P}^1$) or non-compact (along $D^{\widehat{\mathbf{Y}}} \cong \mathbb{C}$), depending on whether the vertical edges are internal or external in $\Gamma$. Moreover, an M2-brane wrapped over $\mathcal{C}_{ij}^{(q)}$ maps to a D2-brane wrapped over $D^{\widehat{\mathbf{Y}}}$.

---

[18]For an internal edge in $\Gamma$, this is a $U(1)_\varphi$ fibered over an interval, giving rise to a genus-zero curve. For an external edge of $\Gamma$, this corresponds to a rotation along a non-compact 2-cycle $\mathcal{C} \cong \mathbb{C}$.

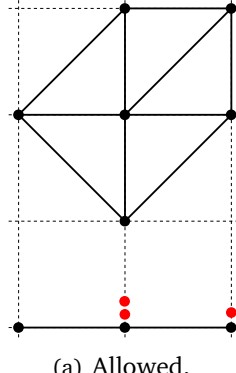

(a) Allowed.

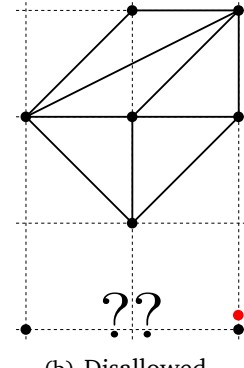

(b) Disallowed.

Figure 4: **Left:** An example of an allowed vertical reduction of a toric diagram $\Gamma$, leading to a resolved $A_1$ singularity $\widehat{\mathbf{Y}}$, pictured by the 1d toric diagram at the bottom. The vertical lines in the middle of $\Gamma$ are mapped to a single exceptional curve $D_1^{\widehat{\mathbf{Y}}} \cong \mathbb{P}^1 \subset \widehat{\mathbf{Y}}$ wrapped by two D6-branes. The vertical line on the right of $\Gamma$ is mapped to a single D6-brane along the non-compact divisor $D_2^{\widehat{\mathbf{Y}}} \cong \mathbb{C} \subset \widehat{\mathbf{Y}}$. **Right:** An example of a triangulation without an allowed vertical reduction, because there is an edge with $\Delta v = (2, 1, 0)$.

(ii) **Allowed oblique edge:** If $\Delta v = (\pm 1, n, 0)$, including the case of an horizontal edge ($n = 0$), we have a two-cycle $\mathcal{C}_{ij}$ with angular direction $v_\varphi = (n, \mp 1, 0)$, which is projected out entirely by the vertical reduction. The curve $\mathcal{C}_{ij}$ maps to an interval along $\mathbb{R} \cong \{r_0\}$ and to a point inside $\widehat{\mathbf{Y}}$ at the intersection of the two adjacent divisors $D_k^{\widehat{\mathbf{Y}}}$ and $D_{k+1}^{\widehat{\mathbf{Y}}}$ to which $v_i$ and $v_j$ map. An M2-brane wrapped over $\mathcal{C}_{ij}$ maps to a fundamental string stretched along the interval.

(iii) **Disallowed oblique edge:** The last case is $\Delta v = (m, n, 0)$ with $|m| > 1$. Without loss of generality, we assume that $m$ and $n$ are mutually prime, so that we have a single edge and a single 2-cycle $\mathcal{C}_{ij}$. The angular direction on the 2-cycle is $v_\varphi = (n, -m, 0)$. In this case, a subgroup $\mathbb{Z}_m$ of $U(1)_M$ degenerates along the curve, and we do not have any simple interpretation of this degeneration in type IIA string theory.

In the following, we will say that a given triangulated toric diagram $\Gamma$ has an "allowed vertical reduction" when all its edges (internal or external) are of type (i) or (ii). An example of a toric diagram without an allowed vertical reduction is given in Figure 4(b). We will further comment on the physical interpretation of the "disallowed" edges in subsection 3.6 below.

## 3.4 Reading off the 5d $\mathcal{N} = 1$ gauge theory from type IIA

Given a toric resolved CY singularity $\widehat{\mathbf{X}}$ with its triangulated toric diagram $\Gamma$, we associate a low-energy gauge-theory phase to a given allowed IIA reduction. By an $SL(2, \mathbb{Z})$ transformation of the toric diagram, we can always view it as a "vertical reduction," as discussed above.

The type IIA configuration consists of a resolved $A_{M-1}$ singularity $\widehat{\mathbf{Y}}$, where $M$ is the horizontal length of $\Gamma$. (For instance, the toric diagram of Fig. 4(a) has $M = 2$.) The ALE space $\widehat{\mathbf{Y}}$ is fibered over the line $\{r_0\}$. As we will see in many examples, the resolution parameters $\chi_s(r_0)$ of the $M - 1$ exceptional curves, (3.40), are piece-wise linear, continuous functions of $r_0$ [52, 54]. The jumps in the first derivative of $\chi_s(r_0)$ occurs at the locations of D6-brane

sources. In fact, supersymmetry imposes the relation:

$$\chi_s'(r_0) = \frac{1}{2\pi} \int_{\mathbb{P}^1_s} F_2^{\mathrm{RR}} \tag{3.45}$$

between the first derivative of $\chi_s(r_0)$ and the RR 2-form flux through the exceptional curve.

For every vertical edge in $\Gamma$—that is, with $\Delta v = (0,1,0)$—, there is a D6-brane wrapping the curve $D_k^{\widehat{\mathbf{Y}}}$ to which the vertical edges maps, at some particular value of $r_0$. Internal vertical edges give rise to D6-branes wrapped over the exceptional curves $\mathbb{P}^1_s$, while external vertical edges give rise to non-compact D6-branes. We refer to these two types of D6-branes as "gauge" and "flavor" D6-branes, respectively.

As we cross a D6-brane wrapped over $D_k^{\widehat{\mathbf{Y}}}$ at some particular $r_0 = \xi_{\mathrm{D6};k}$, the RR fluxes through the exceptional cycles jump. Due to (3.45), the slope of $\chi_s(r_0)$ jumps accordingly, with:

$$\chi_s'(\xi_{\mathrm{D6};k} + \epsilon) - \chi_s'(\xi_{\mathrm{D6};k} - \epsilon) = -\left(\mathbb{P}^1_s \cdot D_k^{\widehat{\mathbf{Y}}}\right), \tag{3.46}$$

for some small $\epsilon > 0$. For instance, for a gauge D6-brane wrapping $\mathbb{P}^1_s$ at $r_0 = \xi_{\mathrm{D6};s}$, the slope of $\chi_s(r_0)$ jumps by $+2$ when crossing the brane, since $\mathbb{P}^1_s$ has self-intersection $-2$ inside $\widehat{\mathbf{Y}}$.

**5d $\mathcal{N} = 1$ quiver from D6-branes.** Given the above configuration of D6-branes wrapped along 2-cycles inside $\mathcal{M}_5$, we can read off the 5d $\mathcal{N} = 1$ gauge theory along the transverse $\mathbb{R}^{1,4}$. For each vertical internal line with $\Delta v = (0,n,0)$ in the toric diagram, we (naively) have a gauge group $U(n)$. By the standard rules for branes at ADE singularities [89], we then obtain a $A$-type gauge-theory quiver of the form:

$$[U(n_0)] \longrightarrow U(n_1)_{k_1} \longrightarrow \quad \cdots \quad \longrightarrow U(n_{M-1})_{k_{M-1}} \longrightarrow [U(n_M)]. \tag{3.47}$$

Here, each line represents an hypermultiplet in a bifundamental representation, and the bracketed groups on either ends are flavor groups, corresponding to the flavor D6-branes. Each gauge group may also have a non-trivial Chern-Simons term at level $k_s$, as we explain below.

The $U(n_s)$ gauge group is realized when the $n_s$ D6-branes wrapped over $\mathbb{P}^1_s$ are brought on top of each other along the $r_0$ direction; this corresponds to a singular limit of $\widehat{\mathbf{X}}$. In general, $\widehat{\mathbf{X}}$ corresponds to a particular point on the Coulomb branch of the quiver (3.47), where the distances between gauge D6-branes are the Coulomb branch VEVs $\langle\varphi\rangle$. Moreover, the low-energy gauge group is actually:

$$[U(n_0)] \longrightarrow SU(n_1)_{k_1} \longrightarrow \quad \cdots \quad \longrightarrow SU(n_{M-1})_{k_{M-1}} \longrightarrow [U(n_M)]. \tag{3.48}$$

Namely, each overall $U(1) \subset U(n_s)$ is massive and decoupled. In M-theory, the gauge fields in the Cartan subalgebra of $SU(n)$ come from the periods of the three-form $C_3$ on the $n-1$ "vertical" curves. The existence or not of the overall $U(1)$, on the other hand, depends on the precise boundary conditions for the "KK monopole." In the present case, it is absent. [19] The corresponding mechanism in type IIA in not entirely clear to us, however.

---

[19]This comes from the difference between the multi-Taub-NUT and the ALE metric. A multi-TN of charge $n$ has $\mathbb{R}^3 \times S^1$ asymptotics; reducing M-theory along the $S^1$ gives us a stack of $n$ flat D6-branes in type IIA. In that case, the overall $U(1)$ comes from the reduction of $C_3$ along a normalizable anti-self-dual two-form of $\mathrm{TN}_n$. The ALE metric, on the other hand, describes the center region of $\mathrm{TN}_n$, with $\mathbb{C}^n/\mathbb{Z}_n$ asymptotics, and does not support the $U(1)$ mode [90].

**"Effective" CS levels.** The presence of the RR flux in $\mathcal{M}_5$ induces Chern-Simons interactions on the gauge D6-branes due to the Wess-Zumino term [51]:

$$\int_{\mathbb{R}^{1,4}\times\mathbb{P}^1_s} C_1 \wedge F \wedge F \wedge F \,. \tag{3.49}$$

By integration by part, this induces a 5d Chern-Simons level along $\mathbb{R}^{1,4}$ for a probe D6-brane wrapped on $\mathbb{P}^1_s$:

$$k_s(r_0) = -\frac{1}{2\pi} \int_{\mathbb{P}^1_s} F_2^{\text{RR}}(r_0) \,. \tag{3.50}$$

Due to (3.45), the CS levels can thus be read off from the slopes of the IIA profiles. For each $U(n_s)$ factor in (3.47), the effective CS level $k_s$ can be computed as follows [54]. Let us denote by $\chi'_{s,\pm}$ the slope to the right and left of the IIA profile $\chi_s$, respectively. That is, for each exceptional curve $\mathbb{P}^1_s$, we define:

$$\chi'_{s,\pm} = \lim_{r_0 \to \pm\infty} \chi'_s(r_0) \,. \tag{3.51}$$

Then, the effective Chern-Simons level $k_s$ is given by minus the average of the slopes:

$$k_s = -\frac{1}{2}\left(\chi'_{s,-} + \chi'_{s,+}\right) \,. \tag{3.52}$$

In fact, this "CS level" (in the common parlance) can be half-integer, and corresponds to the CS contact term $\kappa$, including the half-integer contributions from matter fields—see equation (A.16) in Appendix. We will generally denote $k_s$ in (3.52) by $k_{s,\text{eff}}$, as it is an "effective" CS level.

**Particle states from IIA.** The half-BPS particles on the Coulomb branch are easily read off from the IIA configuration. For $\widehat{\mathbf{X}}$ in a given Kähler chamber, we have the gauge and flavor D6-branes at points along the $x^9 = r_0$ direction. Let us denote these D6-branes by $D6_{k,(a)}$, where $k = 0, \cdots, M$ and $a = 1, \cdots, n_k$ for each $k$, and let $r_0 = \xi_{k,(a)}$ be their $r_0$ positions. The perturbative particles simply arise from open string stretched between two D6-branes:

- The W-bosons of the $SU(n_s)$ gauge group are realized as open strings connecting the D6-branes wrapped over $\mathbb{P}^1_s$. Their masses are given by the distances between any two such gauge D6-branes along $r_0$:

$$M(W_{s;i,j}) = |\xi_{s,(a_i)} - \xi_{s,(a_j)}| \,. \tag{3.53}$$

- The bifundamental hypermultiplets are given by the open strings stretched between two gauge branes wrapping the intersecting curves $\mathbb{P}^1_s$ and $\mathbb{P}^1_{s+1}$, with masses:

$$M(\mathcal{H}_{s,s+1;i,j}) = |\xi_{s,(a_i)} - \xi_{s+1,(a_j)}| \,. \tag{3.54}$$

- The fundamental hypermultiplets are the open string stretched between a flavor D6-brane along $D_0^{\widehat{\mathbf{Y}}}$ and a gauge branes on $\mathbb{P}^1_1$, or between a gauge brane on $\mathbb{P}^1_{M-1}$ and the flavor brane along $D_M^{\widehat{\mathbf{Y}}}$. Their mass is given similarly by the separation distance.

In addition, we have the instanton particles, which arise as D2-branes wrapped over the exceptional curves $\mathbb{P}^1_s$ on top of a D6-brane at $r_0 = \xi_{s,(a)}$. Their mass is given by the size of the curve they wrap:

$$M(\mathcal{I}_{s,(a)}) = \chi_s(\xi_{s,(a)}) \,. \tag{3.55}$$

This should be compared to (2.32) in the field-theory description. As mentioned there, for every $s$, the particle of lowest mass can be viewed as "the" instanton particle $\mathcal{I}_s$, while the other ones are bound state of $\mathcal{I}_s$ with perturbative particles. The "mixed" instanton particle, corresponding to a D2-brane wrapping a curve $\mathbb{P}^1_s$ at a location $r_0 = \xi_{k,(a)}$ with $k \neq s$, are also interpreted as a bound state of $\mathcal{I}_s$ with other perturbative particles in the quiver.

**String states from IIA.** Finally, we should discuss the monopole strings, which correspond to M5-branes wrapped over the exceptional compact divisors $D_j = \mathbf{E}_j \subset \widehat{\mathbf{X}}$. Let $v_j$ denote the corresponding vector in the toric fan. If there is an allowed vertical reduction, $v_j$ is part of a vertical line $v_i, v_j, v_k$ in the toric diagram, with the two curves $\mathcal{C}_{ij} = D_i \cdot D_j$ and $\mathcal{C}_{jk} = D_j \cdot D_j$ projecting down to two D6-branes wrapped over the same $\mathbb{P}^1_s$ (for the appropriate $s$) at some locations $r_0 = \xi_{s,(a)}$ and $\xi_{s,(a+1)}$, respectively. The compact four-cycle $D_j$ then maps to a 2-cycle $\mathbb{P}^1_s$ fibered over the interval $[\xi_{s,(a)}, \xi_{s,(a+1)}]$.

The monopole strings for each $SU(n_s)$ correspond to D4-branes wrapped over $\mathbb{P}^1_s$ and stretched between two subsequent gauge D6-branes. The corresponding string tensions can be computed as the integral of the IIA profile over the interval:

$$T_{s,(a)} = \int_{\xi_{s,(a)}}^{\xi_{s,(a+1)}} dr_0 \, \chi(r_0) \, . \tag{3.56}$$

As we will see in a number of examples, this type IIA perspective on the 5d $\mathcal{N} = 1$ gauge-theory phases of 5d SCFTs allows us to read off the precise map between field-theory parameters and the Kähler parameters of $\widehat{\mathbf{X}}$ systematically.

## 3.5 Global symmetries

To conclude this section, let us briefly discuss global symmetries of the 5d $\mathcal{N} = 1$ theory from the M-theory/IIA perspective.

**Parity.** First of all, the 5d parity operation can be realized as a geometric operation on $\mathbf{X}$. At the level of the toric diagram, it corresponds to an invertion of the toric vectors:

$$w_i \mapsto w_i \cdot C_0 = -w_i \, , \qquad C_0 = \begin{pmatrix} -1 & 0 \\ 0 & -1 \end{pmatrix} \, , \tag{3.57}$$

where $C_0 \equiv S^2$ is the non-trivial central element of $SL(2, \mathbb{Z})$. In the IIA geometry (3.36), parity can be realized as a reflexion of the $x^9 = r_0$ coordinate. In particular, we see from (3.52) that the operation $\chi(r_0) \to \chi(-r_0)$ flips the signs of the effective CS levels.

**Global symmetry.** The rank of the flavor symmetry group $\mathbf{G}_F$ can be read off from the toric geometry as the number of external points in the toric diagram minus 3:

$$\text{rank}(\mathbf{G}_F) = f = n_E - 3 \, , \tag{3.58}$$

namely, the number of linearly-independent non-compact toric divisors. In the IIA setup, the global non-abelian flavor group $\mathbf{G}_\mathcal{H} \subset \mathbf{G}_F$ acting on the hypermultiplets arises from coincident non-compact D6-branes. At the SCFT point, $\mathbf{G}_F$ is sometimes enhanced to a larger global symmetry group [1], although it is not obvious to see this directly from the singular geometry $\mathbf{X}$.

Finally, the $SU(2)_R$ symmetry of the 5d $\mathcal{N} = 1$ gauge theory (which is preserved on its Coulomb branch) is realized geometrically in type IIA as the standard $SU(2)_R$ action on the hyper-Kähler ALE geometry $\widehat{\mathbf{Y}}$ (see *e.g.* [91]).

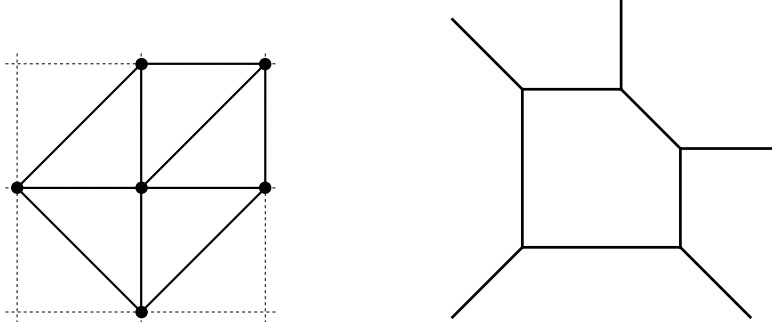

Figure 5: A triangulated toric diagram and its dual $(p,q)$-web.

### 3.6 Comparing to $(p,q)$-webs

To conclude this discussion of the IIA perspective on toric CY three-fold singularities in M-theory, it is interesting to compare it to the $(p,q)$-web description in type IIB [5]. Recall that a five-brane $(p,q)$-web consists of a network of $(p,q)$-fivebranes, parallel along the $x^{0,1,2,3,4}$ directions and forming a web on the $(x^5, x^6)$ plane. The $SO(3)$ rotation group acting on the transverse directions $x^{7,8,9}$ becomes the $SU(2)_R$ R-symmetry of the 5d $\mathcal{N} = 1$ theory. [20]

The $(p,q)$-web is the dual graph to the triangulated toric diagram for $\widehat{\mathbf{X}}$, as shown in a example in Figure 5. Let us choose an $SL(2,\mathbb{Z})$ duality frame in which the D5-brane—corresponding to $(p,q) = (1,0)$—is represented by an horizontal line in the $(p,q)$-web. This corresponds to the "vertical reduction" of the toric diagram to type IIA. Thus, we have a simple correspondence between the 2-cycles (compact and non-compact) in $\widehat{\mathbf{X}}$ and the $(p,q)$-branes in the web:

$$E_{ij} \in \Gamma \text{ with } \Delta v = (m,n,0) \quad \longleftrightarrow \quad \text{fivebrane with } (p,q) = (n,-m) . \tag{3.59}$$

Of course, a vertical edge in the toric diagram corresponds to a D5-brane in the type-IIB $(p,q)$-web, which is indeed T-dual to a D6-brane in type IIA. Similarly, an horizontal edge corresponds to an NS5-brane, which is consistent with the T-duality between a stack of $M$ NS5-branes and the $A_{M-1}$ singularity the in type-IIA description.

The $(p,q)$-web picture also gives us a complementary perspective on the nature of the "disallowed" edges discussed at the end of subsection 3.3: an oblique edge with $\Delta v = (m,n,0)$ and $\gcd(m,n) = 1$ corresponds to an $(n,-m)$-fivebrane, which does not have a perturbative string-theory description if $|m| > 1$. Upon vertical reduction of $\widehat{\mathbf{X}}$ to type IIA, we then expect to find a non-perturbative bound state of D6-branes and geometry. It would be interesting to explore this point of view further. [21]

## 4 Rank-one examples: $SU(2)$ gauge theories

In this section, in order to illustrate our method, we study the well-known $E_{N_f+1}$ series of 5d SCFTs [1]. These are five-dimensional superconformal theories with $E_{N_f+1}$ global symmetry, for $N_f < 8$. They can be engineered by considering M-theory on a $CY_3$ singularity $\mathbf{X}_{(N_f+1)} \equiv E_{N_f+1}$, obtained from the collapse of a del Pezzo surface, $dP_{N_f+1}$, inside a CY threefold. The Coulomb branch of this theory then corresponds to the local del Pezzo threefold:

---

[20]Assuming that all the 5-branes sit at $x^{7,8,9} = 0$. Moving the branes away from each other in the transverse directions corresponds to probing the Higgs branch, thus breaking the R-symmetry [5].

[21]Similar "non-perturbative" degenerations of the M-theory circle were briefly discussed in [53].

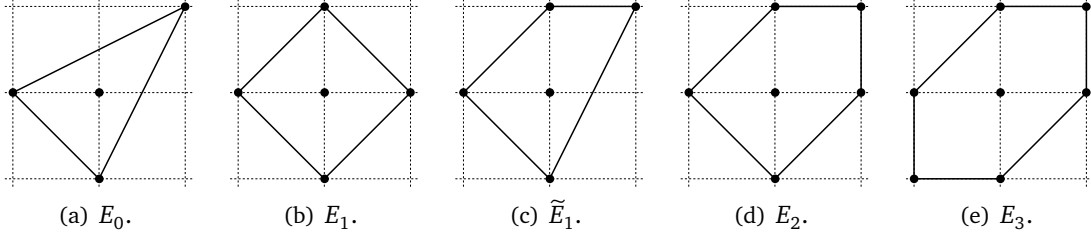

| (a) $E_0$. | (b) $E_1$. | (c) $\widetilde{E}_1$. | (d) $E_2$. | (e) $E_3$. |

Figure 6: Toric diagrams for the toric $CY_3$ singularities that engineer the $E_{N_f+1}$ 5d SCFTs.

$$\widehat{\mathbf{X}}_{(N_f+1)} = \text{Tot}\left(\mathcal{K} \to dP_{N_f+1}\right), \tag{4.1}$$

which is a resolution of the singular variety $\mathbf{X}_{(N_f+1)}$. This variety is toric if and only if $N_f \leq 2$, therefore we restrict ourselves to those cases. [22] The toric diagrams of the corresponding singularities are summarized in Figure 6. Note that $dP_0 \cong \mathbb{P}^2$, and that for $N_f = 0$ we have two distinct singularities, corresponding to $\mathbb{F}_0 \cong \mathbb{P}^1 \times \mathbb{P}^1$ and $dP_1$, respectively; the corresponding SCFTs are denoted by $E_0$, $E_1$ and $\widetilde{E}_1$ [2]. Note also that the singularities $E_1$ ($\mathbb{F}_0$) and $E_3$ ($dP_3$) are "parity invariant" in the sense of (3.57), while the other three singularities in Fig. 6 are not.

For $N_f \geq 0$, the $E_{N_f+1}$ SCFT admits a relevant deformation to an $SU(2)$ gauge theory with $N_f$ fundamental flavors. The gauge theory preserves only an $SO(2N_f) \times U(1)_T$ flavor symmetry, which is enhanced to $E_{N_f+1}$ at the UV fixed point [1,2]. Its prepotential reads:

$$\mathcal{F}_{SU(2),N_f} = h_0 \varphi^2 + \frac{4}{3}\varphi^3 - \frac{1}{6}\sum_{i=1}^{N_f}\sum_{\pm}\Theta(\pm\varphi + m_i)(\pm\varphi + m_i)^3. \tag{4.2}$$

Here, we choose the $SU(2)$ Weyl chamber $\varphi \geq 0$. The single Coulomb branch parameter $\varphi$ corresponds to the single exceptional divisor in $\widehat{\mathbf{X}}_{(N_f+1)}$, which is $dP_{N_f+1}$ itself—in terms of the toric diagrams in Figure 6, this is the single internal point. The $N_f + 1$ parameters $\mu = (h_0, m_i)$ correspond to the $N_f + 1$ linearly-independent non-compact toric divisors—this is $f = n_E - 3 = N_f + 1$ on the toric diagram, with $n_E = N_f + 4$ the number of external points.

For later convenience, we also introduce the prepotential of the associated $U(2)_{k-\frac{1}{2}N_f}$ theory with $N_f$ flavors:

$$\mathcal{F}_{U(2),N_f} = \sum_{a=1}^{2}\left(\frac{h_0}{2}\phi_a^2 + \frac{k}{6}\phi_a^3 - \frac{1}{6}\sum_{i=1}^{N_f}\Theta(\phi_a + m_i)(\phi_a + m_i)^3\right) + \frac{1}{6}(\phi_1 - \phi_2)^3, \tag{4.3}$$

as discussed in section 2.5. This reduces to (4.2) for $\phi_1 = -\phi_2 = \varphi$.

By inspection of Figure 6, it is clear that the $E_1$, $\widetilde{E}_1$, $E_2$ and $E_3$ singularities all admit some "vertical reductions," depending on the partial resolution, as explained in section 3. We will explore the corresponding gauge-theory phases in the following. On the other hand, the $E_0$ singularity in Fig. 6(a) admits neither vertical nor horizontal (nor any other) reduction; its $(p,q)$-web contains a $(-1,2)$-fivebrane, and in that sense it is indeed a "non-Lagrangian" theory [2,92].

---

[22]In section 7, we will explore some non-isolated toric singularities which give access to a subspace of the parameter space of the $E_{N_f+1}$ theory for $N_f = 3, 4, 5$.

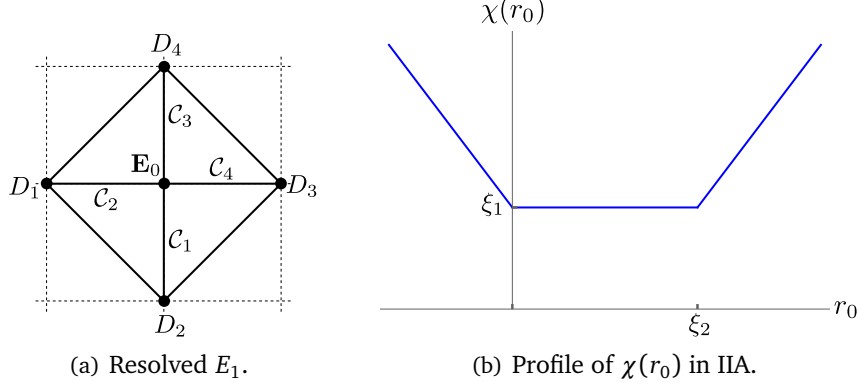

(a) Resolved $E_1$.      (b) Profile of $\chi(r_0)$ in IIA.

Figure 7: The resolved $E_1$ singularity and its vertical reduction. We have $\xi_2 = 2\varphi$ in the gauge-theory language, and the $SU(2)$ gauge symmetry is restored at $\xi_2 = 0$.

## 4.1   $E_1$ SCFT and $SU(2)_0$ gauge theory

Consider the toric singularity $E_1$, namely the complex cone over $\mathbb{F}_0 \cong \mathbb{P}^1 \times \mathbb{P}^1$. It has a unique resolution, corresponding to blowing up $\mathbb{F}_0$, as shown in Figure 7(a). The corresponding GLSM description gives:

| | $D_1$ | $D_2$ | $D_3$ | $D_4$ | $\mathbf{E}_0$ | vol($\mathcal{C}$) |
|---|---|---|---|---|---|---|
| $\mathcal{C}_1$ | 1 | 0 | 1 | 0 | $-2$ | $\xi_1$ |
| $\mathcal{C}_2$ | 0 | 1 | 0 | 1 | $-2$ | $\xi_2$ |
| $U(1)_M$ | 0 | 1 | 0 | 0 | $-1$ | $r_0$ |

$$(4.4)$$

We have four non-compact toric divisors $D_i$, and one compact toric divisor $\mathbf{E}_0 \cong \mathbb{F}_0$, with the following linear relations:

$$D_3 \cong D_1 \,, \qquad D_4 \cong D_2 \,, \qquad \mathbf{E}_0 \cong -2D_1 - 2D_2 \,. \tag{4.5}$$

The curves $\mathcal{C}$ are given as intersections of pairs of divisors according to:

$$\mathcal{C}_1 = D_2 \cdot \mathbf{E}_0 \,, \qquad \mathcal{C}_2 = D_1 \cdot \mathbf{E}_0 \,, \qquad \mathcal{C}_3 = D_4 \cdot \mathbf{E}_0 \,, \qquad \mathcal{C}_4 = D_3 \cdot \mathbf{E}_0 \,, \tag{4.6}$$

and we have the linear equivalences $\mathcal{C}_3 \cong \mathcal{C}_1$ and $\mathcal{C}_4 \cong \mathcal{C}_2$. The volume of the curves are non-negative, $\xi_1 \geq 0$ and $\xi_2 \geq 0$.

**Geometric prepotential.**    Let us first compute the geometric prepotential from M-theory. We may parametrize the Kähler cone by:

$$S = \mu D_1 + \nu \mathbf{E}_0 \,. \tag{4.7}$$

The parameters $(\mu, \nu)$ are related to the FI parameters as:

$$\xi_1 = \mu - 2\nu \geq 0 \,, \qquad \xi_2 = -2\nu \geq 0 \,. \tag{4.8}$$

One can easily compute the relevant triple-intersection numbers:

$$D_1^3 = 0 \,, \qquad D_1^2 \mathbf{E}_0 = 0 \,, \qquad D_1 \mathbf{E}_0^2 = -2 \,, \qquad \mathbf{E}_0^3 = 8 \,. \tag{4.9}$$

Here, the result for $D_1^3$ for the triple-intersection of the non-compact divisor $D_1$ depends on a choice of regulator, as we discuss in Appendix B, and we can choose $D_1^3 = 0$ for future

convenience—see Appendix B.3.1. The other intersection numbers are non-ambiguously defined, for any smooth resolution strictly inside the Kähler cone (that is, for $\xi_i > 0$). This directly gives:

$$\mathcal{F}(\nu, \mu) = -\frac{1}{6}S^3 = \mu\nu^2 - \frac{4}{3}\nu^3 . \tag{4.10}$$

This is a purely geometric result, which gives a prepotential for some dynamical $U(1)$ 5d $\mathcal{N} = 1$ vector multiplet (including the real scalar field $\nu$) in M-theory on the local CY threefold $\widehat{\mathbf{X}}$. To make contact with the non-abelian gauge-theory description of the low-energy theory, we need to choose a type-IIA string theory reduction.

**Type IIA reduction and gauge-theory parameters.** Let us consider the "vertical reduction" of the toric diagram of Fig. 7(a), with the $U(1)_M$ charges indicated in the last line of (4.4). The type IIA background is a resolved $A_1$ singularity (that is, the resolved $\mathbb{C}^2/\mathbb{Z}_2$ orbifold), fibered over the $x^9 = r_0$ direction. The three vertical points in the toric diagram give rise to two D6-branes wrapping the exceptional $\mathbb{P}^1$ in the resolved $A_1$ singularity, thus realizing a pure $SU(2)$ gauge theory.

The exceptional $\mathbb{P}^1$ corresponds to the curve $\mathcal{C}_1$ in the M-theory description, but its volume $\chi$ varies along $r_0$, in a piecewise-linear fashion, as we explained in section 3.3. By reducing the GLSM (4.4), we easily find the GLSM of the $A_1$ singularity:

$$\begin{array}{c|ccc|c} & t_1 & t_2 & t_0 & \\ \hline \mathbb{P}^1 & 1 & 1 & -2 & \chi(r_0) \end{array} , \tag{4.11}$$

with:

$$\chi(r_0) = \begin{cases} \xi_1 + 2r_0 - 2\xi_2 & \text{if } r_0 \geq \xi_2, \\ \xi_1 & \text{if } 0 \leq r_0 \leq \xi_2, \\ \xi_1 - 2r_0 & \text{if } r_0 \leq 0. \end{cases} \tag{4.12}$$

This profile is shown in Figure 7(b). The kinks of $\chi(r_0)$, where the slope jumps by $+2$ from left to right, indicate the $r_0$ positions of the D6-branes wrapped over the exceptional $\mathbb{P}^1$.

From this IIA picture, we directly read off the gauge theory description, and the map between geometry and field-theory parameters. First of all, when $\xi_2 = 0$, the two wrapped D6-branes realize a 5d $SU(2)$ gauge group at $r_0 = 0$. The $SU(2)$ inverse gauge coupling is then given by the size of the $\mathbb{P}^1$ at $r_0 = 0$, thus $h_0 = \xi_1$ when $\xi_2 = 0$. The effective CS level (3.52) vanishes, [23] and the theory is parity-invariant.

Separating the D6-branes in the $r_0$ direction corresponds to going onto the Coulomb branch. The open strings stretched between the two wrapped D6-branes give us the W-bosons and their superpartners, with mass equal to the separation $\xi_2$. Finally, the instantonic particle on the Coulomb branch corresponds to a D2-brane wrapped over the $\mathbb{P}^1$, at either $r_0 = 0$ or $r_0 = \xi_2$, and its mass is therefore equal to $\xi_1$. In the gauge theory, we have the particle masses:

$$M(W_\alpha) = 2\varphi = \xi_2 , \qquad M(\mathcal{I}_1) = M(\mathcal{I}_2) = h_0 + 2\varphi = \xi_1 , \tag{4.13}$$

which are mapped to the Kähler parameters $\xi_1, \xi_2$ as indicated. While this is a well-known result from the $(p, q)$-web point of view [5], the IIA perspective allow us to perform this computation systematically in complicated examples. [24] The masses of the $SU(2)$ instantons were computed from (4.3), following the prescription of section 2.5. Using (4.8), the relations (4.13) are equivalent to:

$$\mu = h_0 , \qquad \nu = -\varphi . \tag{4.14}$$

---

[23]For $SU(2)$, the CS level $k_{SU(2)} \in \mathbb{Z}$ mod 2 in interpreted as a $\mathbb{Z}_2$-valued $\theta$-angle.

[24]By contrast, the particle states in 5-brane $(p, q)$-webs are generally themselves complicated string-webs, which can be rather more subtle to understand [5].

Plugging these relations into the geometric prepotential (4.10), we indeed recover the $SU(2)$ prepotential:

$$\mathcal{F} = h_0 \varphi^2 + \frac{4}{3} \varphi^3 \, . \tag{4.15}$$

As another consistency check of these identification, let us also compute the string tension. From the field theory, we have:

$$T = \partial_\varphi \mathcal{F} = 2\varphi(h_0 + 2\varphi) \, . \tag{4.16}$$

From the IIA geometry, a string is a D4-brane wrapped over the $\mathbb{P}^1$ and stretched between the wrapped D6-branes. Its tension is therefore given by $T = \xi_1 \xi_2$, in agreement with (4.16). In M-theory, this corresponds to an M5-brane wrapping the exceptional divisor $\mathbf{E}_0$, whose volume is indeed $\xi_1 \xi_2$.

Notice that in this simple case, along the magnetic wall $T = 0$, at least one BPS particle is getting massless, also for the deformed theory.

Finally, note that we could also have chosen the S-dual "horizontal reduction" of the $E_1$ singularity to type IIA. From the symmetry of the toric diagram, it is clear that this gives an isomorphic gauge theory description in terms of pure $SU(2)$, but with the roles of $\xi_1$ and $\xi_2$ interchanged.

## 4.2 $\widetilde{E}_1$ SCFT and $SU(2)_\pi$ gauge theory

Our next example is the $\widetilde{E}_1$ theory, corresponding to the local del Pezzo surface $dP_1$—that is, $\mathbb{P}^2$ blown up at one smooth point. The toric diagram of the threefold is shown in Figure 8(a), with the blown-up $\mathbb{P}^1$ corresponding to $\mathcal{C}_3$. The GLSM can be chosen to be:

| | $D_1$ | $D_2$ | $D_3$ | $D_4$ | $\mathbf{E}_0$ | vol($\mathcal{C}$) |
|---|---|---|---|---|---|---|
| $\mathcal{C}_2$ | 0 | 1 | 0 | 1 | $-2$ | $\xi_2$ |
| $\mathcal{C}_3$ | 1 | 0 | 1 | $-1$ | $-1$ | $\xi_3$ |
| $U(1)_M$ | 0 | 1 | 0 | 0 | $-1$ | $r_0$ |

$$\tag{4.17}$$

The linear relations amongst toric divisors are:

$$D_3 \cong D_1 \, , \qquad D_4 \cong -D_1 + D_2 \, , \qquad \mathbf{E}_0 \cong -D_1 - 2D_2 \, . \tag{4.18}$$

The linear relation amongst curves are:

$$\mathcal{C}_4 \cong \mathcal{C}_2 \, , \qquad \mathcal{C}_1 \cong \mathcal{C}_2 + \mathcal{C}_3 \, . \tag{4.19}$$

In particular, the curves $\{\mathcal{C}_2, \mathcal{C}_3$ can be chosen as generators of the Mori cone.

**Geometric prepotential.** Let us consider:

$$S = \mu D_1 + \nu \mathbf{E}_0 \, , \tag{4.20}$$

with

$$\xi_2 = -2\nu \geq 0 \, , \qquad \xi_3 = \mu - \nu \geq 0 \, . \tag{4.21}$$

Here, the inequalities are the ones that define the Kähler chamber for the resolution of Figure 8(a). For that resolution, the relevant intersection numbers are $D_1^3 = 0$, $D_1^2 \mathbf{E}_0 = 0$, $D_1 \mathbf{E}_0^2 = -2$ and $\mathbf{E}_0^3 = 8$, and therefore the prepotential is the same as in (4.10), namely:

$$\mathcal{F}(\nu, \mu) = \mu \nu^2 - \frac{4}{3} \nu^3 \, . \tag{4.22}$$

The $\widetilde{E}_1$ geometry admits another resolution, shown in 8(c), which admits no type-IIA reduction. More details on the intersection numbers are given in Appendix B.3.2.

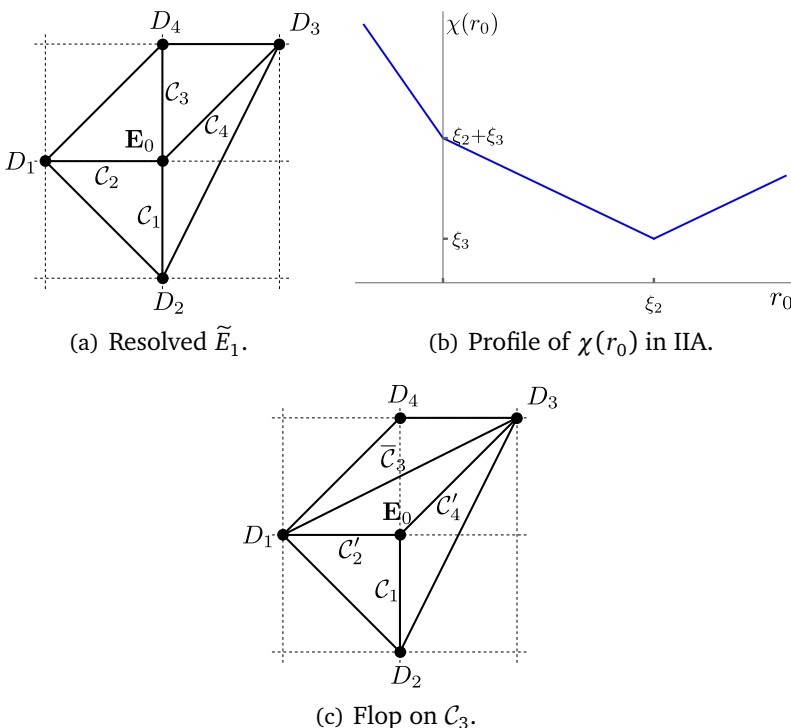

(a) Resolved $\widetilde{E}_1$.

(b) Profile of $\chi(r_0)$ in IIA.

(c) Flop on $\mathcal{C}_3$.

Figure 8: The resolved $\widetilde{E}_1$ singularity and its vertical reduction. By flopping $\mathcal{C}_3$ and sending vol($\bar{\mathcal{C}}_3$) to infinity (that is, $\xi_3 \to -\infty$), one obtains the isolated $E_0$ theory.

**Gauge theory description.** Consider the vertical reduction of Fig. 8(a). We again obtain a pure $SU(2)$ theory, from D6-branes wrapping the resolved $A_1$ singularity, but the IIA profile is different from the $E_1$ case, as shown in Fig. 8(b). We now have:

$$
\chi(r_0) = \begin{cases} r_0 - \xi_2 + \xi_3 & \text{if } \xi_2 \leq r_0, \\ -r_0 + \xi_2 + \xi_3 & \text{if } 0 \leq r_0 \leq \xi_2, \\ -3r_0 + \xi_2 + \xi_3 & \text{if } r_0 \leq 0. \end{cases}
\tag{4.23}
$$

According to (3.52), we naively have the Chern-Simons level:

$$
k_{SU(2)} = 1 \,,
\tag{4.24}
$$

by abuse of notation. More precisely, we would obtain a $U(2)$ theory with level $k = 1$. In the $SU(2)$ theory, this is interpreted as having a non-zero $\theta$-angle valued in $\pi_4(SU(2)) = \mathbb{Z}_2$. Thus, the gauge-theory phases of the $E_1$ and $\widetilde{E}_1$ theories correspond to $\theta = 0$ and $\theta = \pi$, respectively [4]. We denote the $\theta = \pi$ theory by $SU(2)_\pi$. Note that the $\theta$-angle, like the CS level, breaks 5d parity. Interestingly, when considering the "vertical reduction," parity in the gauge theory corresponds to a reflection $r_0 \to -r_0$ direction. Indeed the main difference between the two IIA profiles for $E_1$ and $\widetilde{E}_1$ in Figures 7(b) and 8(b), respectively, in the non-abelian limit $\xi_2 = 0$, is that one is parity-symmetric and the other is not.

The masses of the W-boson and instanton particles are:

$$
M(W_\alpha) = 2\varphi = \xi_2 \,, \qquad M(\mathcal{I}_1) = h_0 + 3\varphi = \xi_2 + \xi_3 \,, \qquad M(\mathcal{I}_2) = h_0 + \varphi = \xi_3 \,.
\tag{4.25}
$$

Comparing to (4.21), we see that $\mu = h_0$ and $\nu = -\varphi$, therefore the gauge theory prepotential is given by (4.15), and it is unaffected by the non-zero $\theta$-angle. On the other hand, the

spectrum of instantonic particles $\mathcal{I}_a$ is different. [25] That spectrum was first obtained in the $(p, q)$-web language in type IIB [5], and the type IIA perspective of course gives the same answer. Note also that we have:

$$M(\mathcal{I}_1) = M(\mathcal{I}_2) + M(W_\alpha) \,. \tag{4.26}$$

In M-theory, the particles $\mathcal{I}_1, \mathcal{I}_2$ and $W_\alpha$ correspond to M2-branes wrapped over $\mathcal{C}_1, \mathcal{C}_3$ and $\mathcal{C}_2$, respectively. Then the marginal bound state relation $\mathcal{I}_1 \cong \mathcal{I}_2 + W_\alpha$ corresponds to the second linear relation in (4.19) amongst the curves.

Since the prepotential for $SU(2)_\pi$ is the same as for $SU(2)_0$, the string tension is also the same and given by (4.16). The IIA prescription for the wrapped D4-brane tension gives:

$$T = \int_0^{\xi_2} \chi(r_0) dr_0 = \xi_2 \left( \frac{1}{2} \xi_2 + \xi_3 \right), \tag{4.27}$$

which indeed reproduces (4.16) upon using (4.25).

The second resolution of the $\widetilde{E}_1$ singularity, shown in Fig. 8(c), can be obtained from the one in Fig. 8(a) by flopping the curve $\mathcal{C}_3$. This corresponds to sending the GLSM parameter $\xi_3$ through $\xi_3 = 0$ and taking it negative. In particular, one can consider the limit $\xi_3 \to -\infty$, which leads to the $E_0$ singularity. Since this would correspond to $h_0 < 0$, we cannot describe this flow in the $SU(2)_\pi$ language. Its end point is the SCFT known as $E_0$, corresponding to a collapsing $\mathbb{P}^2$ [2].

**Magnetic wall.** For this theory the magnetic wall, $T = 0$ splits into two loci

$$(I): \xi_2 = 0 \quad \text{and} \quad (II): \frac{1}{2} \xi_2 + \xi_3 = 0 \,. \tag{4.28}$$

From the spectrum of BPS particle masses we see that the region $(I)$ indeed coincides with the hard-wall where the W-boson becomes massless. The region $(II)$, on the other hand, is not part of the Kähler chamber (a). We note that the BPS instanton $\mathcal{I}_3$ can become massless at $\xi_3 = 0$, away from any magnetic wall, giving rise to a traversable instantonic wall. The theory in this case flows to a chamber that does not have a gauge theory interpretation.

### 4.3 $E_2$ SCFT and $N_f = 1$ $SU(2)$ gauge theory

Consider the $E_2$ singularity. It has five distinct resolutions, shown in Figure 10. Only the first three of them admit a vertical reduction to type IIA. The resolutions (b), (c), (d) admit a horizontal reduction, giving us an S-dual gauge theory description. The resolution (e) admit no gauge theory description. We will focus here on the vertical reduction. The GLSM describing the $E_2$ singularity can be chosen to be:

|  | $D_1$ | $D_2$ | $D_3$ | $D_4$ | $D_5$ | $\mathbf{E}_0$ |  |
|---|---|---|---|---|---|---|---|
| $\mathcal{C}_1$ | 1 | 0 | 1 | 0 | 0 | $-2$ | $\xi_1$ |
| $\mathcal{C}_2$ | 0 | 1 | 0 | 1 | 0 | $-2$ | $\xi_2$ |
| $\mathcal{C}_5$ | 0 | 0 | 1 | 1 | $-1$ | $-1$ | $\xi_5$ |
| $U(1)_M$ | 0 | 1 | 0 | 0 | 0 | $-1$ | $r_0$ |

$$\tag{4.29}$$

Here, the rows are labelled by the curves $\mathcal{C}_1, \mathcal{C}_2$ and $\mathcal{C}_5$ in resolution (a), as shown in Figure 10(a). They have positive volume if $\xi_1 > 0$, $\xi_2 > 0$ and $\xi_5 > 0$. By allowing more general

---

[25]To obtain (4.25), we use the prescription (2.32) for a $U(2)_1$ theory.

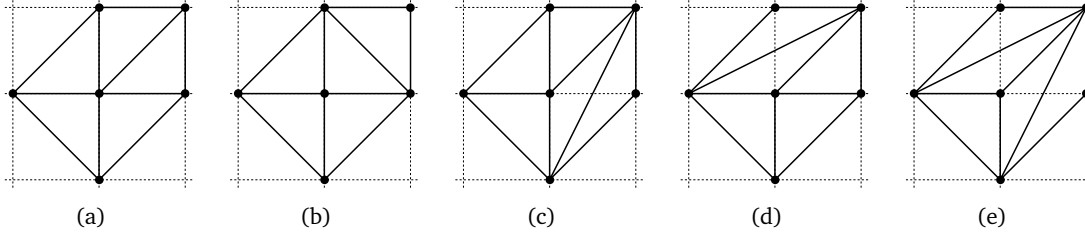

Figure 9: The five resolutions of the $E_2$ singularity.

values of the "FI parameters" $\xi_1$, $\xi_2$ and $\xi_5$, the same GLSM also describes all the resolutions shown in Figure 9. Note also the linear relations amongst toric divisors:

$$D_1 \cong D_3 + D_5 \,, \qquad D_2 \cong D_4 + D_5 \,, \qquad \mathbf{E}_0 \cong -2D_1 - 2D_2 + D_5 \,, \tag{4.30}$$

which hold independently of the partial resolution.

The resolutions (a), (b) and (c) give rise to an $SU(2)$ gauge theory with a single fundamental flavor ($N_f = 1$). Depending on the values of the parameters, we have three distinct chambers on the Coulomb branch. The gauge-theory prepotential (4.2) with $N_f = 1$ takes the values:

$$\mathcal{F} = \begin{cases} \left(h_0 - \frac{m}{2}\right)\varphi^2 + \frac{7}{6}\varphi^3 - \frac{1}{2}m^2\varphi - \frac{1}{6}m^3 & \text{if } \varphi + m > 0,\ -\varphi + m < 0, \\ (h_0 - m)\varphi^2 + \frac{4}{3}\varphi^3 - \frac{1}{3}m^3 & \text{if } \varphi + m > 0,\ -\varphi + m > 0, \\ h_0\varphi^2 + \frac{4}{3}\varphi^3 & \text{if } \varphi + m < 0,\ -\varphi + m < 0. \end{cases} \tag{4.31}$$

As we will show, these three Coulomb-branch chambers correspond to the first three resolutions in Fig. 9:

$$\begin{array}{llll} \text{(a)} & \longleftrightarrow & \varphi + m > 0\,, & -\varphi + m < 0\,, \\ \text{(b)} & \longleftrightarrow & \varphi + m > 0\,, & -\varphi + m > 0\,, \\ \text{(c)} & \longleftrightarrow & \varphi + m < 0\,, & -\varphi + m < 0\,. \end{array} \tag{4.32}$$

Recall that we have $\varphi > 0$, while the real mass $m$ can take both signs.

**Geometric prepotential.** The M-theory prepotential $\mathcal{F} = -\frac{1}{6}S^3$ depends on the partial resolution we consider. Let us define:

$$S = \mu_1 D_1 + \mu_5 D_5 + \nu \mathbf{E}_0 \,. \tag{4.33}$$

The parameters $\mu, \nu$ are related to the FI parameters in (B.25) by:

$$\xi_1 = \mu_1 - 2\nu \,, \qquad \xi_2 = -2\nu \,, \qquad \xi_5 = -\mu_5 - \nu \,. \tag{4.34}$$

By a direct computation of the intersection numbers (see Appendix B.3.3 for more details), we find the following result: [26]

$$\begin{aligned} \mathcal{F}_{(a)} &= -\frac{7}{6}\nu^3 + \left(\mu_1 + \frac{1}{2}\mu_5\right)\nu^2 + \frac{1}{2}\mu_5^2\nu - \frac{1}{6}\mu_5^3 \,, \\ \mathcal{F}_{(b)} &= -\frac{4}{3}\nu^3 + \mu_1\nu^2 - \frac{1}{3}\mu_5^3 \,, \\ \mathcal{F}_{(c)} &= -\frac{4}{3}\nu^3 + (\mu_1 + \mu_5)\nu^2 \,, \end{aligned} \tag{4.35}$$

---

[26]Here, as before, we defined the "non-compact" intersection numbers in a convenient fashion.

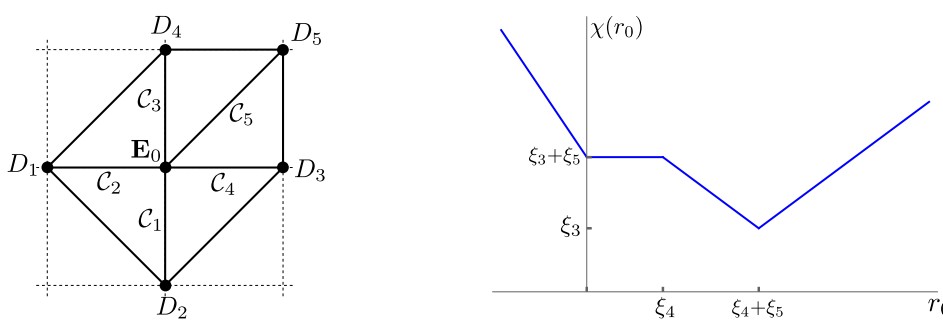

(a) Chamber (a). The flavor D6-brane lies between the two gauge D6-branes.

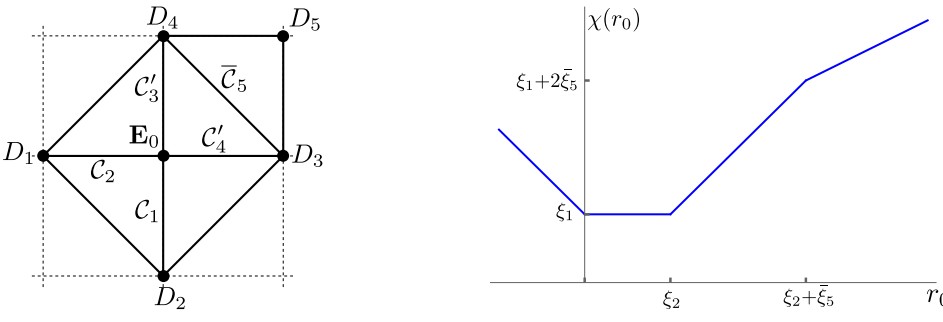

(b) Chamber (b). The flavor D6-brane lies to the right of the gauge branes.

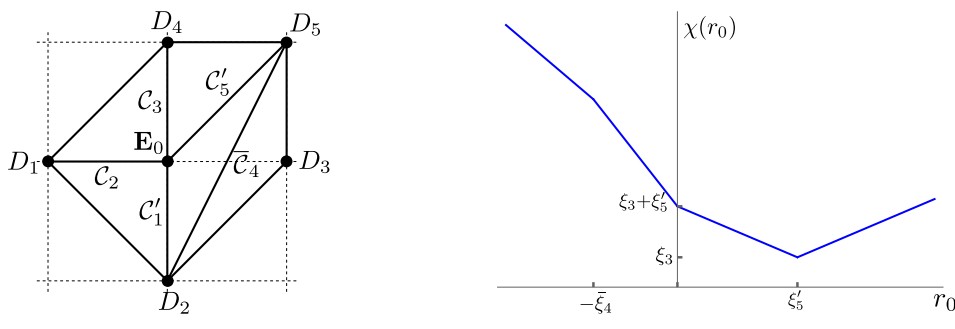

(c) Chamber (c). The flavor D6-brane lies to the left of the gauge branes.

Figure 10: The three resolutions of the $E_2$ singularity with a "vertical reduction," and the corresponding IIA profiles for $\chi(r_0)$.

for the first three resolutions. This matches perfectly with the field-theory result (4.31), given the map of parameters:

$$\mu_1 = h_0 - m \,, \qquad \mu_5 = m \,, \qquad \nu = -\varphi \,, \tag{4.36}$$

To derive (4.36), we again turn to the IIA reduction.

**Comparing to the results obtained using the IMS prepotential.** At this point, it may be useful to make some comments related to our new prescription for the prepotential. [27] The IMS prepotential [4] in this theory reads:

$$\mathcal{F}_{\text{IMS}} = \widetilde{h}_0 \varphi^2 + \frac{4}{3}\varphi^3 + \frac{1}{12}|\varphi + m|^3 + \frac{1}{12}|-\varphi + m|^3 = \begin{cases} \widetilde{h}_0 \varphi^2 + \frac{7}{6}\varphi^3 - \frac{1}{2}m^2\varphi \,, \\ \left(\widetilde{h}_0 - \frac{m}{2}\right)\varphi^2 + \frac{4}{3}\varphi^3 \,, \\ \left(\widetilde{h}_0 + \frac{m}{2}\right)\varphi^2 + \frac{4}{3}\varphi^3 \,, \end{cases} \tag{4.37}$$

---

[27]We thank the SciPost referee for this suggestion.

in the three chambers, up to the constant term. It is clear that, modulo the constant term, this agrees with (4.31) upon redefining the gauge coupling parameter:

$$\widetilde{h}_0 = h_0 - \frac{m}{2} \ . \tag{4.38}$$

Thus, the prepotential (4.37) can be matched to the M-theory prepotential (4.35), but only by using a different map between geometry and field theory parameters, with $\widetilde{h}_0 = \mu_1 + \frac{1}{2}\mu_5$ instead of $h_0 = \mu_1 + \mu_5$ identified as the gauge coupling. Precisely this alternative parameterization was found in [5] when matching the string tension computed as $T = \partial_\varphi \mathcal{F}_{\text{IMS}}$ with the area of the face on the $(p,q)$-web. [28] Similar comments hold for all the other examples below.

**Gauge-theory chamber (a).** Consider the vertical reduction of resolution (a), which is shown in Fig. 10(a). In this geometry, we have the relations:

$$\mathcal{C}_1 \cong \mathcal{C}_3 + \mathcal{C}_5 \ , \qquad \mathcal{C}_2 \cong \mathcal{C}_4 + \mathcal{C}_5 \ , \tag{4.39}$$

amongst the curves, and therefore $\{\mathcal{C}_3, \mathcal{C}_4, \mathcal{C}_5\}$ generates the Mori cone. We denote by $\xi_i$ the volume of $\mathcal{C}_i$, and therefore we have $\xi_3 = \xi_1 - \xi_5$ and $\xi_4 = \xi_2 - \xi_5$. The IIA reduction gives again a resolved $A_1$ singularity fibered over $r_0$. One finds:

$$(a) \ : \qquad \chi(r_0) = \begin{cases} r_0 + \xi_3 - \xi_4 - \xi_5 & \text{if} \quad \xi_4 + \xi_5 \geq r_0, \\ -r_0 + \xi_3 + \xi_4 + \xi_5 & \text{if} \quad \xi_4 \leq r_0 \leq \xi_4 + \xi_5, \\ \xi_3 + \xi_5 & \text{if} \quad 0 \leq r_0 \leq \xi_4, \\ -2r_0 + \xi_3 + \xi_5 & \text{if} \quad r_0 \leq 0, \end{cases} \tag{4.40}$$

as shown in Fig. 10(a). We see that there are now two wrapped D6-branes at $r_0 = 0$ and $r_0 = \xi_4 + \xi_5 = \xi_2$, corresponding to the gauge group, and one non-compact "flavor" D6-brane at $r_0 = \xi_4$. (At the location of the flavor brane, the slope of $\chi(r_0)$ decreases by $-1$ from left to right.) The "effective CS level" (3.52) is now given by:

$$k_{SU(2),\text{eff}} = \frac{1}{2} \ . \tag{4.41}$$

In our conventions, this is interpreted as a bare CS level $k = 1$ plus the contribution $-\frac{1}{2}$ from the hypermultiplet.

The open string between the gauge D6-branes provide the W-boson, and the open strings stretching from the gauge branes to the flavor brane provide the hypermultiplet modes. Therefore, we find:

$$M(\mathcal{H}_+) = \varphi + m = \xi_4 \ , \qquad M(\mathcal{H}_-) = \varphi - m = \xi_5 \ , \qquad M(W_\alpha) = 2\varphi = \xi_4 + \xi_5 \ . \tag{4.42}$$

In particular, we see that, in this particular resolution, the hypermultiplet comes from M2-branes wrapped over $\mathcal{C}_4$ and $\mathcal{C}_5$ in M-theory (this is to be contrasted with the rather more complicated description of the hypermultiplet states in the $(p;q)$-web [5]). The masses of the instantonic particles are given by the volumes of the D2-branes wrapped at the $r_0$ locations of the gauge D6-branes. We find:

$$M(\mathcal{I}_1) = h_0 + \varphi = \xi_3 \ , \qquad M(\mathcal{I}_2) = h_0 + 2\varphi - m = \xi_3 + \xi_5 \ . \tag{4.43}$$

---

[28]It is instructive to compare section III.D of [5] to our formalism in more detail. Their parameters $(1/g_0^2, m, \phi)$ for $SU(2), N_f = 1$ correspond to our $(\widetilde{h}_0, m, 2\varphi)$, respectively. The brane configuration in their Fig.16(a) and (b) correspond to our resolution (a) and (b), respectively. The 'simple' and 'subtle' quark states in their discussion correspond to $\mathcal{H}_+$ and $\mathcal{H}_-$, respectively, in our discussion around (4.42) below. They then identify the 'simplest' instanton mass as $\widetilde{h}_0 + 2\varphi - \frac{m}{2}$, in our notation; given the shift (4.38), this agrees with $M(\mathcal{I}_2)$ which we find in (4.43) below; the lightest instanton, $\mathcal{I}_1$, would correspond to a D-string along the top D5-brane on the $(p,q)$-web.

In particular, $\mathcal{I}_1$ comes from an M2-brane wrapped over $\mathcal{C}_3$. Together with (4.34), the relations (4.42)-(4.43) imply the map (4.36) between $(\mu_1, \mu_5, \nu)$ and the gauge-theory variables, as anticipated. The gauge-theory formula for the instanton masses in (4.43) is derived from (4.3) with the $U(2)$ CS level $k = 1$.

As a consistency check, let us consider the string tension. The gauge-theory prepotential implies:

$$T^{(a)} = \partial_\varphi \mathcal{F} = \frac{7}{2}\varphi^2 + (2h_0 - m)\varphi - \frac{1}{2}m^2 . \tag{4.44}$$

From the IIA D4-brane description, we find:

$$T^{(a)} = \int_0^{\xi_4 + \xi_5} \chi(r_0)dr_0 = (\xi_3 + \xi_5)\xi_4 + \left(\xi_3 + \frac{1}{2}\xi_5\right)\xi_5 . \tag{4.45}$$

Plugging in (4.42)-(4.43), this precisely reproduces the gauge-theory result (4.44).

**Magnetic wall.** In this case, we see that the tension does not factorize into a product of masses of BPS particles. We obtain:

$$T^{(a)} = (\xi_3 + \xi_5)(\xi_4 + \xi_5) - \tfrac{1}{2}\xi_5^2 = M(\mathcal{I}_1)M(W_\alpha) - \tfrac{1}{2}(M(\mathcal{H}_-))^2 . \tag{4.46}$$

A magnetic wall would correspond to solving $T = 0$. Solving this equations for $\xi_5$ gives $\xi_5 = -\xi_3 - \xi_4 \pm \sqrt{\xi_3^2 + \xi_4^2}$, which is always negative in chamber (a). Hence, there is no magnetic wall in this chamber (except at the origin, corresponding to the SCFT).

**Gauge-theory chamber (b).** It is instructive to consider the other resolutions with a vertical reduction, as a consistency check. The toric diagram of resolution (b) and its IIA profile are shown in Figure 10(b). We have:

$$\text{(b)} : \quad \chi(r_0) = \begin{cases} r_0 + \xi_1 - \xi_2 + \bar{\bar{\xi}}_5 & \text{if } \xi_2 + \bar{\bar{\xi}}_5 \geq r_0, \\ 2r_0 + \xi_1 - 2\xi_2 & \text{if } \xi_2 \leq r_0 \leq \xi_2 + \bar{\bar{\xi}}_5, \\ \xi_1 & \text{if } 0 \leq r_0 \leq \xi_2, \\ -2r_0 + \xi_1 & \text{if } r_0 \leq 0. \end{cases} \tag{4.47}$$

Here, the parameters $\xi_1$ and $\xi_2$ are the same as above, and $\bar{\bar{\xi}}_5 = -\xi_5$ (while $\xi_3' = \xi_3 + \xi_5$ and $\xi_4' = \xi_4 + \xi_5$). Indeed, resolution (b) is obtained from resolution (a) by flopping the curve $\mathcal{C}_5$ to $\bar{\mathcal{C}}_5 = -\mathcal{C}_5$. In the gauge theory, this corresponds to changing the sign of the hypermultiplet mode $\mathcal{H}_-$. Note that the flavor brane is to the right of the gauge branes along the $r_0$ direction. We now have:

$$M(\mathcal{H}_+) = \varphi + m = \xi_2 + \bar{\bar{\xi}}_5 , \quad M(\mathcal{H}_-) = -\varphi + m = \bar{\bar{\xi}}_5 , \quad M(W_\alpha) = 2\varphi = \xi_2 , \tag{4.48}$$

and

$$M(\mathcal{I}_1) = M(\mathcal{I}_2) = h_0 - m + 2\varphi = \xi_1 . \tag{4.49}$$

The instanton mass is again obtained from a $U(2)_1$ description. We also have the string tension:

$$T^{(b)} = \xi_1 \xi_2 = 2\varphi(h_0 - m + 2\varphi) = \partial_\varphi \mathcal{F} . \tag{4.50}$$

Note that we can take the limit $m \to +\infty$ within this chamber. In the field theory, this corresponds to integrating out the hypermultiplet with positive mass, which gives the $SU(2)_0$ gauge theory. [29] This is clearly also what happens in the geometry: the $E_2$ geometry reduces

---

[29]Since we start from a theory with $k_{\text{eff}} = \frac{1}{2}$, integrating out the hypermultiplet with positive mass gives rise to a theory with $k_{\text{eff}} = 0$.

to the $E_1$ geometry as size of $\bar{\mathcal{C}}_5$ is sent to infinity. (This is clear from the toric diagram of Fig. 10(b), since that limit effectively decouples the divisor $D_5$ from the rest of the geometry as we "decompactify" $\bar{\mathcal{C}}_5$.)

**Gauge-theory chamber (c).** The third resolution and its associated IIA profile are shown in Figure 10(c). The relations amongst curves are:

$$\mathcal{C}_1' = \mathcal{C}_3 + \mathcal{C}_2 , \qquad \mathcal{C}_5' \cong \mathcal{C}_2 . \tag{4.51}$$

We now have:

$$(c) : \qquad \chi(r_0) = \begin{cases} r_0 + \xi_3 - \xi_5' + \bar{\xi}_5 & \text{if } \xi_5' \geq r_0, \\ -r_0 + \xi_3 + \xi_5' & \text{if } 0 \leq r_0 \leq \xi_5', \\ -3r_0 + \xi_3 + \xi_5' & \text{if } -\bar{\xi}_4 \leq r_0 \leq 0, \\ -2r_0 + \xi_3 + \bar{\xi}_4 + \xi_5' & \text{if } r_0 \leq -\bar{\xi}_4. \end{cases} \tag{4.52}$$

Here we introduced the new parameters:

$$\bar{\xi}_4 = -\xi_4 , \qquad \xi_5' = \xi_2 \tag{4.53}$$

corresponding to the positive sizes of the curves $\bar{\mathcal{C}}_4$ and $\mathcal{C}_5'$, respectively. This resolution can be obtained from resolution (a) by flopping the curve $\mathcal{C}_4$. Now the flavor brane is located to the left of the gauge branes. We find

$$M(\mathcal{H}_+) = -\varphi - m = \bar{\xi}_4 , \quad M(\mathcal{H}_-) = \varphi - m = \bar{\xi}_4 + \xi_5' , \quad M(W_\alpha) = 2\varphi = \xi_5' , \tag{4.54}$$

and

$$M(\mathcal{I}_1) = h_0 + 3\varphi = \xi_3 + \xi_5' , \qquad M(\mathcal{I}_2) = h_0 + \varphi = \xi_3 . \tag{4.55}$$

The string tension reads

$$T^{(c)} = \int_0^{\xi_5'} \chi(r_0) dr_0 = \xi_5'\left(\xi_3 + \frac{1}{2}\xi_5'\right) = 2\varphi(h_0 + 2\varphi) = \partial_\varphi \mathcal{F} . \tag{4.56}$$

In this chamber, we can take the limit $m \to -\infty$, corresponding to sending the size of $\bar{\mathcal{C}}_4$ to infinity. This leads to the parity-violating $SU(2)_\pi$ gauge theory discussed above (with "$k_{\text{eff}} = 1$"), and correspondingly we recover the $\widetilde{E}_1$ geometry in that limit. Note that the fact that we obtain distinct geometries depending on the sign of $m$ is a manifestation of the parity anomaly carried by this single hypermultiplet.

In summary, we have verified that the three complete resolutions of the $E_2$ singularity with an allowed vertical reduction can be precisely matched to the gauge theory description, with all the mass parameters turned on. The perturbative RG flows also have a simple geometric interpretation as partial resolutions of the $E_2$ singularity which results in either the $E_1$ or the $\widetilde{E}_1$ singularity.

## 4.4 $E_3$ SCFT and $N_f = 2$ $SU(2)$ gauge theory

Consider the $E_3$ singularity. It has 18 distinct resolutions, 9 of which admit a vertical reduction to type IIA, as shown in Figure 11. Its GLSM can be chosen to be:

| | $D_1$ | $D_2$ | $D_3$ | $D_4$ | $D_5$ | $D_6$ | $\mathbf{E}_0$ | |
|---|---|---|---|---|---|---|---|---|
| $\mathcal{C}_1$ | $-1$ | $1$ | $0$ | $0$ | $0$ | $1$ | $-1$ | $\xi_1$ |
| $\mathcal{C}_2$ | $1$ | $-1$ | $1$ | $0$ | $0$ | $0$ | $-1$ | $\xi_2$ |
| $\mathcal{C}_3$ | $0$ | $1$ | $-1$ | $1$ | $0$ | $0$ | $-1$ | $\xi_3$ |
| $\mathcal{C}_5$ | $0$ | $0$ | $0$ | $1$ | $-1$ | $1$ | $-1$ | $\xi_5$ |
| $U(1)_M$ | $0$ | $0$ | $-1$ | $0$ | $0$ | $0$ | $1$ | $r_0$ |

$$\tag{4.57}$$

The rows are labeled by the curves $\mathcal{C}_1$, $\mathcal{C}_2$, $\mathcal{C}_3$ and $\mathcal{C}_5$ in resolution (a), as shown in Figure 12(a). They have positive volume if $\xi_1 > 0$, $\xi_2 > 0$, $\xi_3 > 0$ and $\xi_5 > 0$. By allowing more general values of the FI parameters, the same GLSM also describes all the resolutions shown in Figure 11. Note also the linear relations among toric divisors:

$$D_4 \cong D_1 + D_2 - D_5 \ , \quad D_6 \cong D_2 + D_3 - D_5 \ , \quad E_0 \cong -2D_1 - 3D_2 - 2D_3 + D_5 \ , \tag{4.58}$$

which hold independently of the partial resolution.

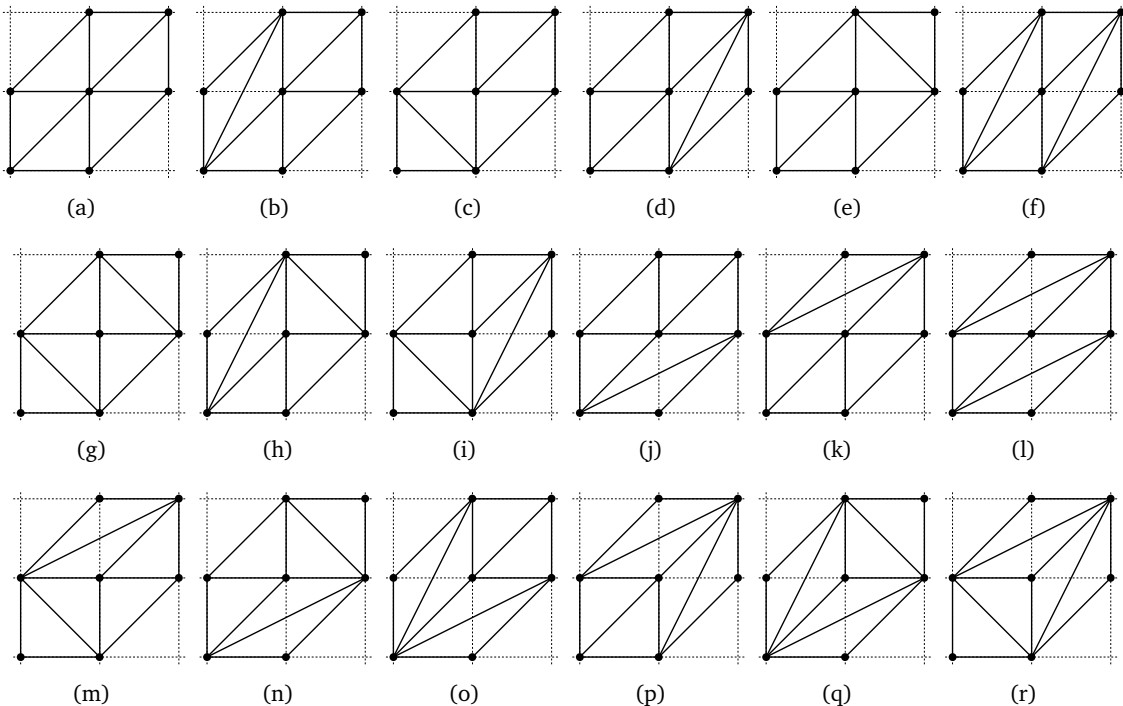

Figure 11: The 18 resolutions of the $E_3$ singularity. The 9 resolutions (a)-(i) admit a vertical reduction. There are also 9 resolutions that admit an horizontal reduction (namely: (a), (c), (e), (g), (j)-(n)), and 9 resolutions that admit an "oblique" reduction (namely: (a), (b), (d), (f), (g)-(l), (o), (p)). The last three resolutions have no "allowed" IIA reduction.

The 9 resolutions (a)-(i) give rise to an $SU(2)$ gauge theory with two fundamental flavors ($N_f = 2$). Depending on the values of the parameters, we have nine distinct chambers on the Coulomb branch. The gauge theory prepotential (4.2) with $N_f = 2$ takes the form:

$$
\begin{aligned}
\mathcal{F}_{(a)} &= \left(h_0 - \frac{m_1 + m_2}{2}\right)\varphi^2 + \varphi^3 - \frac{1}{2}(m_1^2 + m_2^2)\varphi - \frac{1}{6}(m_1^3 + m_2^3) \ , \\
\mathcal{F}_{(b)} &= \left(h_0 - \frac{m_1}{2} - m_2\right)\varphi^2 + \frac{7}{6}\varphi^3 - \frac{1}{2}m_1^2\varphi - \frac{1}{6}(m_1^3 + m_2^3) \ , \\
\mathcal{F}_{(c)} &= \left(h_0 - \frac{m_1}{2}\right)\varphi^2 + \frac{7}{6}\varphi^3 - \frac{1}{2}m_1^2\varphi - \frac{1}{6}m_1^3 \ ,
\end{aligned}
\tag{4.59}
$$

and

$$
\begin{aligned}
\mathcal{F}_{(d)} &= \left(h_0 - \frac{m_2}{2}\right)\varphi^2 + \frac{7}{6}\varphi^3 - \frac{1}{2}m_2^2\varphi - \frac{1}{6}m_2^3 , \\
\mathcal{F}_{(e)} &= \left(h_0 - m_1 - \frac{m_2}{2}\right)\varphi^2 + \frac{7}{6}\varphi^3 - \frac{1}{2}m_2^2\varphi - \frac{1}{6}(m_1^3 + m_2^3) , \\
\mathcal{F}_{(f)} &= (h_0 - m_2)\varphi^2 + \frac{4}{3}\varphi^3 - \frac{1}{2}m_2^3 , \\
\mathcal{F}_{(g)} &= (h_0 - m_1)\varphi^2 + \frac{4}{3}\varphi^3 - \frac{1}{3}m_1^3 , \\
\mathcal{F}_{(h)} &= (h_0 - m_1 - m_2)\varphi^2 + \frac{4}{3}\varphi^3 - \frac{1}{3}(m_1^3 + m_2^3) , \\
\mathcal{F}_{(i)} &= h_0\varphi^2 + \frac{4}{3}\varphi^3 .
\end{aligned}
\tag{4.60}
$$

Here, the subscripts correspond to the field theory chambers:

$$
\begin{aligned}
\text{(a)} &\leftrightarrow \begin{cases} \varphi + m_1 \geq 0, \ -\varphi + m_1 < 0 \\ \varphi + m_2 \geq 0, \ -\varphi + m_2 < 0 \end{cases} &
\text{(b)} &\leftrightarrow \begin{cases} \varphi + m_1 \geq 0, \ -\varphi + m_1 < 0 \\ \varphi + m_2 \geq 0, \ -\varphi + m_2 \geq 0 \end{cases} \\
\text{(c)} &\leftrightarrow \begin{cases} \varphi + m_1 \geq 0, \ -\varphi + m_1 < 0 \\ \varphi + m_2 < 0, \ -\varphi + m_2 < 0 \end{cases} &
\text{(d)} &\leftrightarrow \begin{cases} \varphi + m_1 < 0, \ -\varphi + m_1 < 0 \\ \varphi + m_2 \geq 0, \ -\varphi + m_2 < 0 \end{cases} \\
\text{(e)} &\leftrightarrow \begin{cases} \varphi + m_1 \geq 0, \ -\varphi + m_1 \geq 0 \\ \varphi + m_2 \geq 0, \ -\varphi + m_2 < 0 \end{cases} &
\text{(f)} &\leftrightarrow \begin{cases} \varphi + m_1 < 0, \ -\varphi + m_1 < 0 \\ \varphi + m_2 \geq 0, \ -\varphi + m_2 \geq 0 \end{cases} \\
\text{(g)} &\leftrightarrow \begin{cases} \varphi + m_1 \geq 0, \ -\varphi + m_1 \geq 0 \\ \varphi + m_2 < 0, \ -\varphi + m_2 < 0 \end{cases} &
\text{(h)} &\leftrightarrow \begin{cases} \varphi + m_1 \geq 0, \ -\varphi + m_1 \geq 0 \\ \varphi + m_2 \geq 0, \ -\varphi + m_2 \geq 0 \end{cases} \\
\text{(i)} &\leftrightarrow \begin{cases} \varphi + m_1 < 0, \ -\varphi + m_1 < 0 \\ \varphi + m_2 < 0, \ -\varphi + m_2 < 0 \end{cases}
\end{aligned}
\tag{4.61}
$$

Recall that we have $\varphi > 0$, while the real masses $m_1$ and $m_2$ can take both signs. Nonetheless, it is convenient to choose $m_1 > m_2$ (using the U(2) flavor symmetry), in which case the field theory chambers (4.61) precisely correspond to the first 9 resolutions denoted by the same letters in Figure 11.

**Geometric prepotential.** To compute the M-theory prepotential $\mathcal{F} = -\frac{1}{6}S^3$, let us define:

$$
S = \mu_1 D_1 + \mu_2 D_2 + \mu_3 D_3 + \nu \mathbf{E}_0 .
\tag{4.62}
$$

The parameters $\mu, \nu$ are related to the FI terms of (4.57) by:

$$
\mu_1 = \xi_2 + \xi_3 - 2\xi_5 , \quad \mu_2 = \xi_1 + \xi_2 + \xi_3 - 3\xi_5 , \quad \mu_3 = \xi_1 + \xi_2 - 2\xi_5 , \quad \nu = -\xi_5 .
\tag{4.63}
$$

By direct computation of the intersection numbers (see Appendix B.3.4), we find: [30]

$$
\begin{aligned}
\mathcal{F}_{(a)} &= -\nu^3 + \left(\frac{\mu_1}{2} + \frac{\mu_2}{2} + \frac{\mu_3}{2}\right)\nu^2 + \left(\frac{\mu_1^2}{2} - \mu_2\mu_1 + \frac{\mu_2^2}{2} + \frac{\mu_3^2}{2} - \mu_2\mu_3\right)\nu , \\
\mathcal{F}_{(b)} &= -\frac{7\nu^3}{6} + \left(\mu_2 + \frac{\mu_3}{2}\right)\nu^2 + \left(\frac{\mu_3^2}{2} - \mu_2\mu_3\right)\nu , \\
\mathcal{F}_{(c)} &= -\frac{7\nu^3}{6} + (\mu_1 + \mu_3)\nu^2 - \mu_1\mu_3\nu ,
\end{aligned}
\tag{4.64}
$$

---

[30]Here, for simplicity, we did not keep track of the "non-compact" intersection numbers, so those expressions are valid up to $\nu$-independent terms. Correspondingly, we only match to field theory up to a $\varphi$-independent term. The same comment will apply to the other geometries that we study in the rest of the paper.

for the first two resolutions of the $E_3$ singularity, up to a $\nu$-independent constant term. As we will show, the geometric parameters are matched to the field-theory parameters according to:

$$\mu_1 = h_0 + m_1 \,, \quad \mu_2 = h_0 + 2m_1 - m_2 \,, \quad \mu_3 = 2m_1 \,, \quad \nu = -\varphi + m_1 \,. \tag{4.65}$$

Indeed, plugging (4.65) into (4.64) reproduces the field-theory result (4.59), up to the constant term. The same geometric computation can be carried out for the six other resolutions (d)-(i), and one finds perfect agreement with (4.60).

**Gauge-theory chamber (a).**  Consider the vertical reduction of resolution (a), which is shown in Figure 12(a). In this geometry we have the relations

$$\mathcal{C}_4^a \cong \mathcal{C}_1^a + \mathcal{C}_2^a - \mathcal{C}_5^a \,, \qquad \mathcal{C}_6^a \cong \mathcal{C}_2^a + \mathcal{C}_3^a - \mathcal{C}_5^a \,, \tag{4.66}$$

amongst the curves, and therefore $\{\mathcal{C}_1, \mathcal{C}_2, \mathcal{C}_3, \mathcal{C}_5\}$ generates the Mori cone. We denote by $\xi_i \equiv \xi_i^a$, the volume of $\mathcal{C}_i \equiv \mathcal{C}_i^a$, so that we have $\xi_4 = \xi_1 + \xi_2 - \xi_5 > 0$ and $\xi_6 = \xi_2 + \xi_3 - \xi_5 > 0$. The IIA reduction gives again a resolved $A_1$ singularity fibered over $r_0$. One finds:

$$(a) \;:\; \chi(r_0) = \begin{cases} r_0 + \xi_3^a & \text{if} \quad r_0 \geq 0, \\ -r_0 + \xi_3^a & \text{if} \quad -\xi_2^a \leq r_0 \leq 0, \\ \xi_2^a + \xi_3^a & \text{if} \quad \xi_5^a - \xi_1^a - \xi_2^a \leq r_0 \leq -\xi_2^a, \\ r_0 + \xi_1^a + 2\xi_2^a + \xi_3^a - \xi_5^a & \text{if} \quad -\xi_1^a - \xi_2^a \leq r_0 \leq \xi_5^a - \xi_1^a - \xi_2^a, \\ -r_0 - \xi_1^a + \xi_3^a - \xi_5^a & \text{if} \quad r_0 \leq -\xi_1^a - \xi_2^a, \end{cases} \tag{4.67}$$

as shown in Figure 12(a). We directly see that there are two gauge D6-branes at $r_0 = 0$ and $r_0 = -\xi_1^a - \xi_2^a$, respectively, and two non-compact "flavor" D6-branes at $r_0 = \xi_5^a - \xi_1^a - \xi_2^a$ and $r_0 = -\xi_2^a$. The effective CS level (3.52) vanishes in this theory, $k_{\text{eff}} = 0$. In our conventions, this is interpreted as a bare CS level $k = 1$ (for the "naive" $U(2)$ gauge group) plus the contribution $-\frac{1}{2} - \frac{1}{2} = -1$ from the two hypermultiplets.

The open string between the gauge D6-branes provide the W-boson, and the open strings stretching from the gauge branes to the two flavor branes provide the hypermultiplet modes. Therefore, we find

$$M(W_\alpha) = 2\varphi = \xi_1^a + \xi_2^a \,, \tag{4.68}$$

and

$$\begin{aligned} M(\mathcal{H}_{1,1}) &= \varphi - m_1 = \xi_5^a \,, & M(\mathcal{H}_{1,2}) &= \varphi - m_2 = \xi_1^a, \\ M(\mathcal{H}_{2,1}) &= \varphi + m_1 = \xi_1^a + \xi_2^a - \xi_5^a \,, & M(\mathcal{H}_{2,2}) &= \varphi + m_2 = \xi_2^a. \end{aligned} \tag{4.69}$$

The masses of the instanton particles are given by the volumes of the wrapped D2-branes at the $r_0$ locations of the gauge D6-branes:

$$M(\mathcal{I}_1) = h_0 + \varphi - m_1 - m_2 = \xi_3^a, \qquad M(\mathcal{I}_2) = h_0 + \varphi = \xi_2^a + \xi_3^a - \xi_5^a \,. \tag{4.70}$$

Here, as before, the instanton masses can be calculated using the prepotential for the $U(2)$ $N_f = 2$ theory (at bare CS level $k = 1$). Note that the instanton particle $\mathcal{I}_1$ comes from an M2-brane wrapped over $\mathcal{C}_3^a$, whereas $\mathcal{I}_2$ comes from an M2-brane wrapped over $\mathcal{C}_6^a$. Plugging the above identifications into (4.63), we derive the geometry-to-field-theory map (4.65). One can also check that the IIA computation of the string tension:

$$T = \int_{-\xi_1 - \xi_2}^0 \chi(r_0) dr_0 = \xi_1 \xi_2 + \xi_2 \xi_3 + \xi_3 \xi_1 + \frac{1}{2}(\xi_2^2 - \xi_5^2) \,, \tag{4.71}$$

agrees with the field-theory result, $T = \partial_\varphi \mathcal{F}_{(a)}$ in this chamber. As in previous examples, we see clearly that an instanton can become massless while the string tension is kept finite.

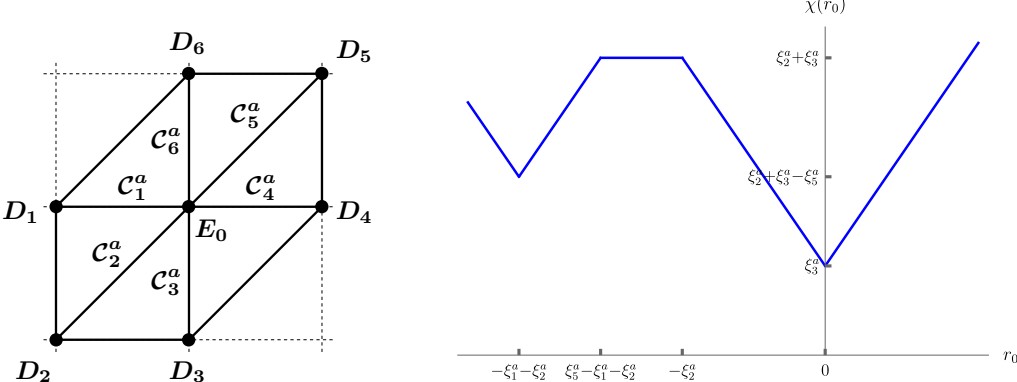

(a) The two flavor D6-branes lie between the two gauge D6-branes.

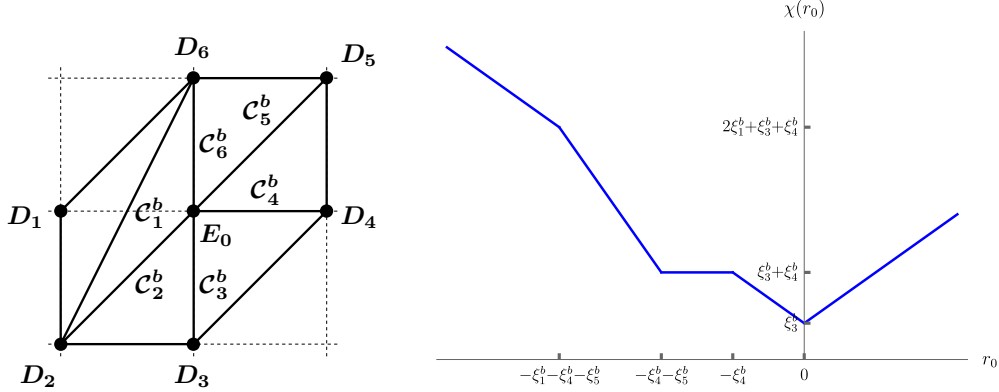

(b) One flavor brane between the two gauge branes, and the other to the left.

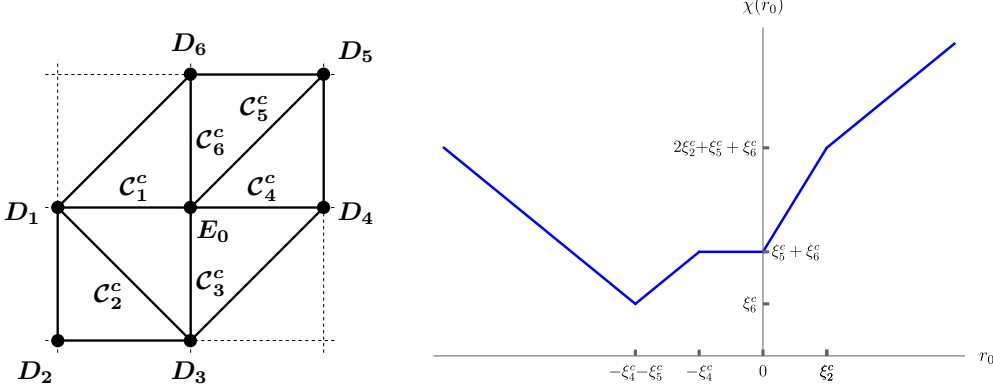

(c) One flavor brane between the two gauge branes, and the other to the right.

Figure 12: The first three resolutions, (a), (b) and (c), of the $E_3$ singularity and their IIA profiles $\chi(r_0)$ upon vertical reduction.

**Gauge-theory chamber (b).** The toric diagram of resolution (b) and its IIA profile are shown in Figure 12(b). In this geometry, we have the relations:

$$\mathcal{C}_2^b \cong \mathcal{C}_4^b + \mathcal{C}_5^b , \qquad \mathcal{C}_6^b \cong \mathcal{C}_3^b + \mathcal{C}_4^b , \tag{4.72}$$

amongst the curves; we therefore pick $\{\mathcal{C}_1^b, \mathcal{C}_3^b, \mathcal{C}_4^b, \mathcal{C}_5^b\}$ as a basis. The Kähler volume of $\mathcal{C}_i^a$, denoted by $\xi_i^b$, are related to parameters from resolution (a) by:

$$\xi_1^b = -\xi_1^a = -\xi_1 , \quad \xi_3^b = \xi_3^a = \xi_3 , \quad \xi_4^b = \xi_4^a = \xi_1 + \xi_2 - \xi_5 , \quad \xi_5^b = \xi_5^a = \xi_5 . \tag{4.73}$$

Indeed, the resolution (b) is obtained from resolution (a) by flopping the curve $\mathcal{C}_1^a$ to $\mathcal{C}_1^b \cong -\mathcal{C}_1^a$. We find the IIA profile:

$$
(b) : \chi(r_0) = \begin{cases}
r_0 + \xi_3^b & \text{if } r_0 \geq 0, \\
-r_0 + \xi_3^b & \text{if } -\xi_4^b \leq r_0 \leq 0, \\
\xi_3^b + \xi_4^b & \text{if } -\xi_4^b - \xi_5^b \leq r_0 \leq -\xi_4^b, \\
-2r_0 + \xi_3^b - \xi_4^b - 2\xi_5^b & \text{if } -\xi_1^b - \xi_4^b - \xi_5^b \leq r_0 \leq -\xi_4^b - \xi_5^b, \\
-r_0 + \xi_1^b + \xi_3^b - \xi_5^b & \text{if } r_0 \leq -\xi_1^b - \xi_4^b - \xi_5^b.
\end{cases}
\tag{4.74}
$$

In this case, the flavor D6-branes are at $r_0 = -\xi_1^b - \xi_4^b - \xi_5^b$ and $r_0 = -\xi_4^b$, whereas the gauge D6-branes are at $r_0 = -\xi_4^b - \xi_5^b$ and $r_0 = 0$. Thus, one flavor brane lies between the gauge D6-branes and the other flavor brane lies to the left of the gauge D6-branes along the $r_0$ direction. The W-boson mass is given by $M(W) = 2\varphi = \xi_4^b + \xi_5^b$ and the hypermultiplets masses are:

$$
\begin{aligned}
M(\mathcal{H}_{1,1}) &= \varphi - m_1 = \xi_5^b, & M(\mathcal{H}_{1,2}) &= -\varphi + m_2 = \xi_1^b, \\
M(\mathcal{H}_{2,1}) &= \varphi + m_1 = \xi_4^b, & M(\mathcal{H}_{2,2}) &= \varphi + m_2 = \xi_1^b + \xi_4^b + \xi_5^b.
\end{aligned}
\tag{4.75}
$$

The instanton masses are:

$$
M(\mathcal{I}_1) = h_0 + \varphi - m_1 - m_2 = \xi_3^b, \qquad M(\mathcal{I}_2) = h_0 + 2\varphi - m_2 = \xi_3^b + \xi_4^b.
\tag{4.76}
$$

These results of course agree with (4.65). The string tension computed from IIA also agrees with $T = \partial_\varphi \mathcal{F}_{(b)}$.

**Gauge-theory chamber (c).** The third resolution and its associated IIA profile are shown in Figure 12(c). It can be obtained from resolution (a) by flopping the curve $\mathcal{C}_2^a$. The relations among curves are:

$$
\mathcal{C}_1^c \cong \mathcal{C}_4^c + \mathcal{C}_5^c, \qquad \mathcal{C}_3^c \cong \mathcal{C}_5^c + \mathcal{C}_6^c.
\tag{4.77}
$$

We now have:

$$
(c) : \qquad \chi(r_0) = \begin{cases}
r_0 + \xi_2^c + \xi_5^c + \xi_6^c & \text{if } r_0 \geq \xi_2^c, \\
2r_0 + \xi_5^c + \xi_6^c & \text{if } 0 \leq r_0 \leq \xi_2^c, \\
\xi_5^c + \xi_6^c & \text{if } -\xi_4^c \leq r_0 \leq 0, \\
r_0 + \xi_4^c + \xi_5^c + \xi_6^c & \text{if } -\xi_4^c - \xi_5^c \leq r_0 \leq -\xi_4^c, \\
-r_0 - \xi_4^c - \xi_5^c + \xi_6^c & \text{if } r_0 \leq -\xi_4^c - \xi_5^c.
\end{cases}
\tag{4.78}
$$

Here, we have:

$$
\xi_1^c = \xi_1 + \xi_2, \qquad \xi_2^c = -\xi_2, \qquad \xi_3^c = \xi_2 + \xi_3, \qquad \xi_{4,5,6}^c = \xi_{4,5,6}.
\tag{4.79}
$$

We then find $M(W_\alpha) = 2\varphi = \xi_4^c + \xi_5^c$ for the W-boson, as well as:

$$
\begin{aligned}
M(\mathcal{H}_{1,1}) &= \varphi - m_1 = \xi_5^c, & M(\mathcal{H}_{1,2}) &= \varphi - m_2 = \xi_2^c + \xi_4^c + \xi_5^c, \\
M(\mathcal{H}_{2,1}) &= \varphi + m_1 = \xi_4^c, & M(\mathcal{H}_{2,2}) &= -\varphi - m_2 = \xi_2^c,
\end{aligned}
\tag{4.80}
$$

for the hypermultiplets, and:

$$
M(\mathcal{I}_1) = h_0 + 2\varphi - m_1 = \xi_5^c + \xi_6^c, \qquad M(\mathcal{I}_2) = h_0 + \varphi = \xi_6^c,
\tag{4.81}
$$

for the instanton particles, again in perfect agreement with (4.65).

Note that resolutions (b) and (c) are compatible with the limits $m_2 \to \infty$ and $m_2 \to -\infty$, respectively. In the field theory, this leads to an $SU(2)$ $N_f = 1$ low energy theory. This correspond to removing the toric divisors $D_1$ (resp., $D_2$) by sending the size of $\mathcal{C}_1^b$ (resp., $\mathcal{C}_2^c$) to infinity, giving us the resolved $E_2$ singularity (the $dP_2$ geometry).

**Other resolutions and decoupling limits.** The vertical reduction of the other 6 resolutions (d) to (i) can be performed in exactly the same way. The various geometric decoupling limits also agree with the field-theory RG flows. For instance, resolution (f) is compatible with the limit $m_1 \to -\infty$, $m_2 \to \infty$, which leads to a pure $SU(2)_0$ theory. (That is, $k_{\text{eff}} = 0$ in the IR because the two flavors have real masses of opposite signs.) Indeed, we get the $E_1$ ($\mathbb{F}_0$) geometry upon decouling the divisors $D_1$ and $D_4$. A similar comment holds for resolution (g). On the other hand, resolutions (h) and (i) are compatible with the decoupling limits $m_{1,2} \to \infty$ and $m_{1,2} \to -\infty$, respectively, leading to the $SU(2)_\pi$ theory (that is, $k_{\text{eff}} = \pm 1$) from the $\widetilde{E}_1$ ($dP_1$) geometry.

# 5 Higher-rank example: $SU(N)_k$

To illustrate our methods in some simple higher-rank example, let us consider a family of toric geometries that admit a resolution leading to an $SU(N)_k$ gauge theory [4, 5].

## 5.1 The $SU(N)_k$ gauge theory and its Coulomb branch

Consider a 5d $\mathcal{N} = 1$ gauge theory with gauge group $SU(N)$ and CS level $k \in \mathbb{Z}$. Let $\varphi_a$ ($a = 1, \cdots, N-1$) denote the Coulomb branch scalars, and let $\phi_a$ denote the associated $U(N)$ parameters, as in (2.31). We denote by $A_{ab} = A_{ba}$ the Cartan matrix of $SU(N)$, namely:

$$A_{ab} = 2\delta_{a,b} - \delta_{a,b+1} - \delta_{a+1,b} . \tag{5.1}$$

The $N-1$ simple roots are then given by:

$$\alpha_{(b)}^a = A_{ab}. \tag{5.2}$$

The prepotential on the Coulomb branch of the $SU(N)_k$ theory is easily computed. We have:

$$\mathcal{F}(\varphi, h_0) = \mathcal{F}_{\text{kin}} + \mathcal{F}_{\text{CS}} + \mathcal{F}_{\text{1-loop}} , \tag{5.3}$$

with

$$
\begin{aligned}
\mathcal{F}_{\text{kin}} &= \frac{1}{2} h_0 \sum_{a=1}^{N} \phi_a^2 &&= \frac{1}{2} h_0 A^{ab} \varphi_a \varphi_b , \\
\mathcal{F}_{\text{CS}} &= \frac{k}{6} \sum_{a=1}^{N} \phi_a^3 &&= \frac{k}{2} \sum_{a=1}^{N-1} \left( \varphi_{a-1}^2 \varphi_a - \varphi_a \varphi_{a+1}^2 \right) , \\
\mathcal{F}_{\text{1-loop}} &= \frac{1}{6} \sum_{a<b} (\phi_a - \phi_b)^3 &&= \frac{4}{3} \sum_{a=1}^{N-1} \varphi_a^3 + \frac{1}{2} \sum_{a=1}^{N-1} (N - 2a) \left( \varphi_{a-1}^2 \varphi_a - \varphi_a \varphi_{a+1}^2 \right) .
\end{aligned}
\tag{5.4}
$$

The W-bosons associated to the simple roots have masses:

$$M(W_{\alpha_{(b)}}) = A^{ab} \varphi_a , \tag{5.5}$$

and become massless at the boundaries of the fundamental Weyl chamber.

## 5.2 $SU(N)_k$ gauge theory from a toric singularity

Consider a convex toric diagram with external points at:

$$w_0 = (0,0) , \quad w_N = (0,N) , \quad w_x = (-1, h_x) , \quad w_y = (1, h_y) , \tag{5.6}$$

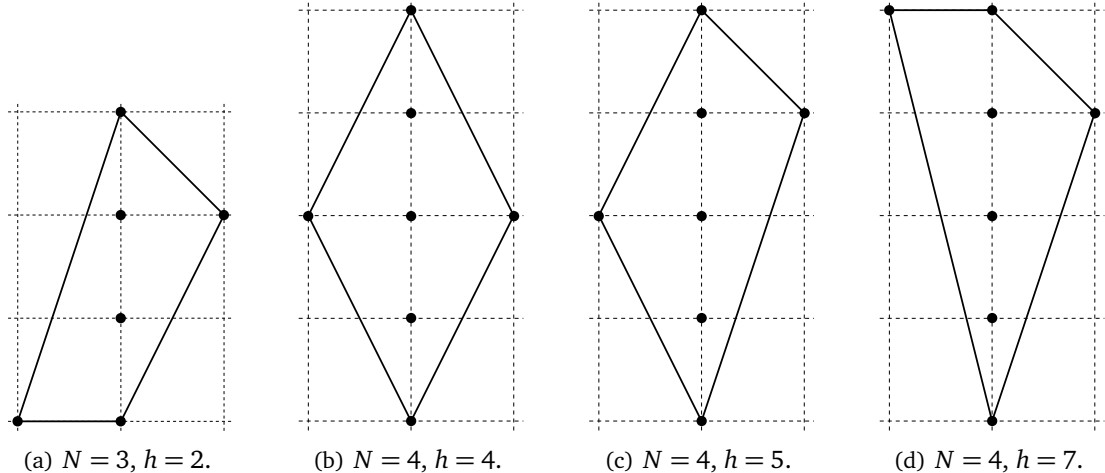

(a) $N = 3, h = 2$.     (b) $N = 4, h = 4$.     (c) $N = 4, h = 5$.     (d) $N = 4, h = 7$.

Figure 13: Examples of "$SU(N)_k$ toric singularities." Upon vertical reduction, the corresponding gauge theories are $SU(3)_{-1}$, $SU(4)_0$, $SU(4)_1$ and $SU(4)_3$, respectively.

with $h_x, h_y \in \mathbb{Z}$ and $N > 2$. (The case $N = 2$ has been discussed above.) We impose the condition:

$$0 < h < 2N , \qquad h \equiv h_x + h_y , \tag{5.7}$$

to ensure that the toric diagram is strictly convex. Some examples are shown in Figure 13. Note that the toric diagram preserves parity, in the sense of (3.57), if and only if $h = N$. This suggests that the $SU(N)$ CS level should be identified with $k = h - N$, up to the overall sign which is a matter of convention. We will confirm this expectation using the IIA setup.

While this toric singularity may admits many distinct resolutions, there is a unique triangulation of the toric diagram with an allowed vertical reductions, as shown in Figure 14. Let us denote by $D_0$, $D_N$, $D_x$ and $D_y$ the toric divisors associated to the four external points (5.6). We also have the exceptional toric divisors $\mathbf{E}_a$ ($a = 1, \cdots, N-1$), corresponding to the internal points $w_a = (0, a)$ in the toric diagram. We have the linear equivalences:

$$D_x \cong D_y , \qquad D_0 \cong (N-1)D_N + (h-2)D_x + \sum_{a=1}^{N-1}(a-1)\mathbf{E}_a ,$$

$$D_N \cong -D_0 - 2D_x - \sum_{a=1}^{N-1}\mathbf{E}_a , \tag{5.8}$$

amongst toric divisors. The resolution shown in Fig. 14(b) contains the exceptional curves:

$$\mathcal{C}_a^x \cong D_x \cdot \mathbf{E}_a , \qquad \mathcal{C}_a^y \cong D_y \cdot \mathbf{E}_a , \qquad a = 1, \cdots N-1 ,$$

$$\mathcal{C}_a^0 \cong \mathbf{E}_{a-1} \cdot \mathbf{E}_a , \qquad a = 1, \cdots N , \tag{5.9}$$

with $\mathbf{E}_0 \equiv D_0$ and $\mathbf{E}_N \equiv D_N$. One finds the following linear equivalences amongst them:

$$\mathcal{C}_a^x \cong \mathcal{C}_a^y , \qquad \mathcal{C}_a^0 - \mathcal{C}_{a+1}^0 \cong (h-2a)\mathcal{C}_a^x , \qquad a = 1, \cdots, N-1 . \tag{5.10}$$

Thus we may consider the $N$ independent curves $\mathcal{C}_a^x$ and $\mathcal{C}_1^0$, for instance. According to our general discussion in section 3.4, M2-branes wrapped on $\mathcal{C}_a^x$ and on $\mathcal{C}_a^0$ map to the W-bosons $W_{\alpha_{(a)}}$ (associated to the simple roots $\alpha_{(a)}$) and to the instanton particles $\mathcal{I}_a$, respectively.

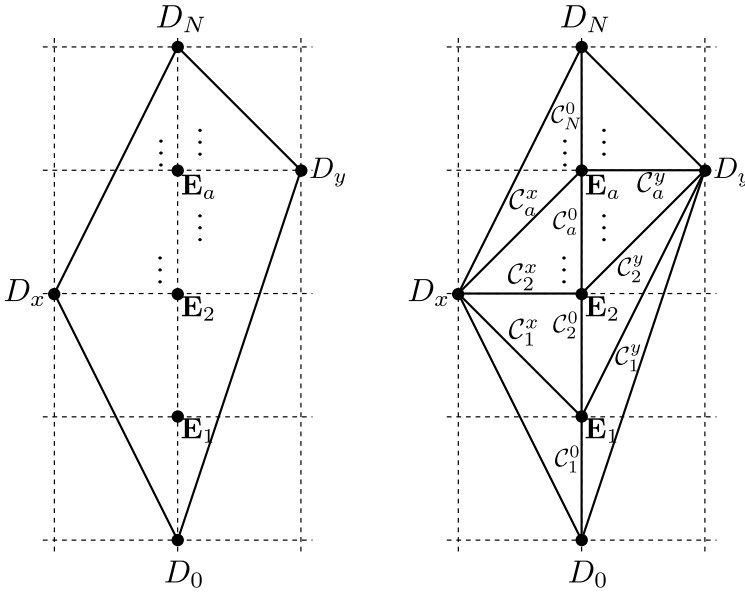

(a) Toric diagram and toric divisors.

(b) Unique resolution with allowed vertical reduction.

Figure 14: A rank-$(N-1)$ toric singularity and its unique resolution with an "allowed" vertical reduction, leading to an $SU(N)_k$ gauge theory on D6-branes in type IIA.

### 5.2.1 Type IIA reduction

Consider the vertical reduction of the triangulated toric diagram of Fig. 14(b). We have the GLSM description:

|  | $D_0$ | $\mathbf{E}_b$ | $D_N$ | $D_x$ | $D_y$ |  |
|---|---|---|---|---|---|---|
| $\mathcal{C}_1^0$ | $h-2$ | $-h\delta_{b,1}$ | $0$ | $1$ | $1$ | $\eta$ |
| $\mathcal{C}_a^x$ | $\delta_{0,1}$ | $-A_{ab}$ | $\delta_{a,N-1}$ | $0$ | $0$ | $\xi_a^x$ |
| $U(1)_M$ | $1$ | $-\delta_{b,1}$ | $0$ | $0$ | $0$ | $r_0$ |

$$(5.11)$$

with $a, b = 1, \cdots, N-1$, and $A_{ab}$ as defined in (5.1). It is convenient to introduce the parameters:

$$\zeta_a = \sum_{b=1}^{a} \xi_b^x, \qquad a = 0, \cdots, N-1, \qquad (5.12)$$

with $\zeta_0 = 0$. It is clear from the vertical projection of the toric diagram that the IIA background is a resolved $A_1 \cong \mathbb{C}^2/\mathbb{Z}_2$ singularity, fibered over $\mathbb{R}$, with $N$ D6-branes wrapped over the exceptional $\mathbb{P}^1$. By a direct computation, one finds the IIA profile:

$$\chi(r_0) = \eta + (2a-h)r_0 - 2\sum_{b=0}^{a-1}\zeta_b, \qquad \text{if} \quad \zeta_{a-1} \leq r_0 \leq \zeta_a, \qquad a = 0, \cdots, N, \qquad (5.13)$$

with the understanding that $\zeta_{-1} \equiv -\infty$ and $\zeta_N \equiv \infty$. An example is shown in Figure 15. Here, each kink corresponds to a single wrapped D6-brane. According to (3.52), this gives rise to a CS level:

$$k = h - N, \qquad (5.14)$$

as anticipated. The convexity condition (5.7) on the toric diagram implies that:

$$|k| < N. \qquad (5.15)$$

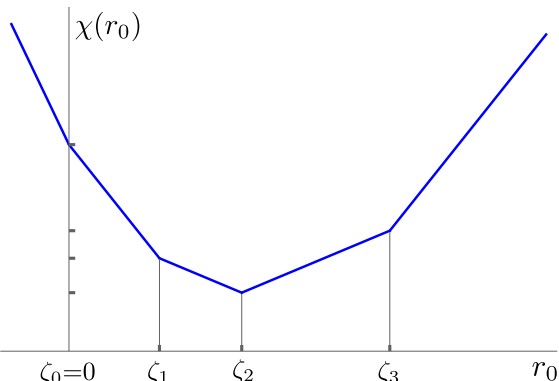

Figure 15: Type IIA profile for $\chi(r_0)$ in the case of the $SU(4)_1$ singularity ($N = 4, h = 5$). Each area under the curve between $\zeta_{a-1}$ and $\zeta_a$ corresponds to the tension $T_a$ of an elementary magnetic string.

Thus, we recover the known fact that the $SU(N)_k$ theory with $|k| < N$ has a UV fixed point described by an isolated toric singularity [5]. The limiting case $|k| = N$, on the other hand, corresponds to a non-isolated toric singularity (with a line of $A_1$ singularities that has been related to an enhanced $SU(2)$ global symmetry at strong coupling [9]). Incidentally, we note that UV fixed points are expected for $SU(N)_k$ with $k > N$ as well [23,50], but they cannot be realized in the strictly toric framework.

From the IIA profile (5.13), we can read off the map between the geometry and the field-theory parameters. The open strings stretched between two adjacent D6-branes give rise to the simple-root W-bosons—more generally, open strings stretched between any two distinct D6-branes span the full set of positive roots, in the obvious way. The simple-root W-bosons have masses:

$$M(W_{\alpha_{(b)}}) = A^{ab}\varphi_a = \zeta_b - \zeta_{b-1} = \xi_b^x , \qquad b = 1, \cdots, N-1 . \tag{5.16}$$

On the other hand, the instanton particles have masses:

$$M(\mathcal{I}_a) = h_0 + 2\varphi_a + (k + N - 2a)(\varphi_a - \varphi_{a-1}) , \qquad a = 1, \cdots, N , \tag{5.17}$$

according to (2.32), which should be identified with the volumes of the wrapped D2-branes at the kinks of (5.13):

$$M(\mathcal{I}_a) = \eta + (2a - h)\zeta_{a-1} - 2\sum_{b=0}^{a-1} \zeta_b . \tag{5.18}$$

This implies the relations:

$$\xi_a^x = A^{ab}\varphi_b , \qquad \eta = h_0 + (N + k)\varphi_1 . \tag{5.19}$$

Note that $\eta$ is also the volume of the curve $\mathcal{C}_1^0$ in M-theory. More generally, the instanton $\mathcal{I}_a$ corresponds to an M2-brane wrapped over $\mathcal{C}_a^0$. Moreover, given the geometry/gauge-theory map (5.19), we can easily check that the type-IIA and gauge-theory computations of the string tensions also agree, with:

$$T_a = \partial_{\varphi_a}\mathcal{F} = \int_{\zeta_{a-1}}^{\zeta_a} \chi(r_0) , \tag{5.20}$$

in terms of the prepotential (5.3)-(5.4) and of the IIA profile (5.13), respectively.

### 5.2.2 Prepotential from M-theory

Finally, let us also compute the prepotential directly from M-theory. We have:

$$S = \mu_x D_x + \sum_{a=1}^{N-1} \nu_a \mathbf{E}_a \ . \tag{5.21}$$

The parameters $\mu_x$ and $\nu_a$ are related to the GLSM FI parameters by:

$$\eta = \mu_x - h\nu_1 \ , \qquad \xi_a^x = -A^{ab}\nu_b \ . \tag{5.22}$$

This follows from (5.11). Comparing to (5.19), we directly see that:

$$\nu_a = -\varphi_a \ , \qquad \mu_x = h_0 \ . \tag{5.23}$$

Therefore, one finds:

$$\mathcal{F} = -\frac{1}{6}S^3 = \frac{1}{2}h_0^2 \varphi_a(D_x^2 \mathbf{E}^a) - \frac{1}{2}h_0 \varphi_a \varphi_b(D_x \mathbf{E}^a \mathbf{E}^b) + \frac{1}{6}\varphi_a \varphi_b \varphi_c(\mathbf{E}^a \mathbf{E}^b \mathbf{E}^c) \tag{5.24}$$

up to an ambiguous $\varphi$-independent term, which we can set to zero in conventions in which $D_x^3 = 0$. One can check that, for the resolution of Fig. 14(b), the only non-zero triple-intersection numbers are:

$$D_x \mathbf{E}_a \mathbf{E}_b = -A_{ab} \ , \quad \mathbf{E}_a^3 = 8 \ , \quad \mathbf{E}_{a-1}^2 \mathbf{E}_a = h - 2a \ , \quad \mathbf{E}_{a-1}\mathbf{E}_a^2 = 2a - 2 - h \ . \tag{5.25}$$

Plugging this into (5.24), one reproduces the field theory prepotential (5.3)-(5.4).

## 6 S-duality for 5d $\mathcal{N} = 1$ gauge theories

A given toric CY$_3$ singularity **X** may admit several type-IIA reductions, giving rise to several distinct gauge-theory "phases." This phenomenon is sometimes called "UV duality," since different 5d $\mathcal{N} = 1$ gauge theories can have the same UV completion as a 5d SCFT [5, 8–13]. In our setup, we have an $SL(2,\mathbb{Z})$-worth family of potential choices of the "M-theory circle $U(1)_M$" along which to reduce to type IIA, corresponding exactly to doing an $SL(2,\mathbb{Z})$ transformation of the toric diagram before peforming the "vertical reduction." In type IIB string theory, this corresponds to S-duality on the $(p,q)$-web.

In this section, we study examples of 5d "UV dualities" with all the possible mass parameters $\mu_j$ turned on. After briefly discussing the rank-one case, we will focus on one particularly nice rank-two example, the "beetle" singularity.

As anticipated in the introduction, when sending all the massive deformation parameters to zero, $\mu_i \to 0$, while keeping $\nu_i \neq 0$ finite, we obtain a partial resolution of the singularity which is "universal" across gauge-theory descriptions. Most IR phases disappear in that limit, and the gauge theory phases coalesce in a much simpler geometry. This universality indicates that we can identify the resulting smaller extended Kähler cone with the genuine Coulomb branch of the 5d SCFT, $\mathcal{M}_{\mathcal{T}_X}^C$. We stress that, in that limit, the gauge theory interpretation breaks down as the gauge coupling is sent to infinity. Nevertheless, perhaps surprisingly, the extrapolated gauge theory result agrees perfectly with the geometric analysis.

### 6.1 "Self-duality" of $SU(2)$ gauge theories

Consider first the rank-one examples. It is clear that the toric diagrams for $E_{N_f+1}$ in Figure 6 can have both an horizontal and a vertical reduction. We then have a self-similar S-duality:

$$SU(2), N_f \text{ hypers} \qquad \longleftrightarrow \qquad SU(2), N_f \text{ hypers} \ , \tag{6.1}$$

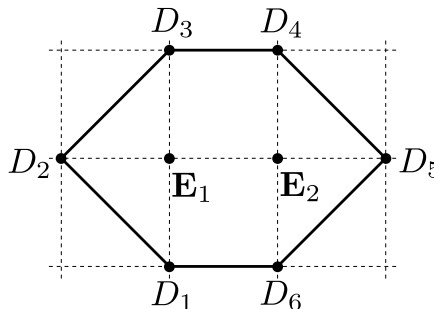

Figure 16: The toric diagram of the beetle geometry

for $N_f = 0, 1, 2$.

As the simplest example, consider the $E_1$ singularity, which is resolved to a local $\mathbb{F}_0 \cong \mathbb{P}^1 \times \mathbb{P}^1$. As discussed in section 4.1, we have a single Kähler chamber with two exceptional curves $\mathcal{C}_1$, $\mathcal{C}_2$ of non-negative volume, $\xi_1 \geq 0$, $\xi_2 \geq 0$. Upon vertical reduction, one finds the relations:

$$\xi_1 = h_0 + 2\varphi \ , \qquad \xi_2 = 2\varphi \ , \tag{6.2}$$

between the Kähler parameters and the field-theory parameters of the pure $SU(2)_0$ theory. If we perform an horizontal reduction instead, we have:

$$\xi_1 = 2\varphi^D \ , \qquad \xi_2 = h_0^D + 2\varphi^D \ , \tag{6.3}$$

where $h_0^D$ and $\varphi^D$ are the parameters of a *dual* pure $SU(2)_0$. This S-duality exchanges the $W$-boson and the instanton particle. Interestingly, strictly speaking, the first $SU(2)$ description is only valid for $\xi_1 > \xi_2$, so that $h_0 > 0$. To explore the rest of the Kähler chamber using a low-energy gauge-theory perspective, one should use the S-dual description with the identifications (6.3) instead. The loci $\xi_1 = \xi_2$ is at the "strong coupling" limit $h_0 = 0$, thus corresponding to the Coulomb branch of the $E_1$ SCFT. (It is not particularly special from the point of view of geometry, however.) The limit $\xi_2 \to 0$ corresponds to a non-traversable Kähler wall, where the W-boson in the first $SU(2)$ description becomes massless [47], while the limit $\xi_2 \to 1$ corresponds to an instantonic wall, where the instanton particle in the first $SU(2)_0$ description becomes massless. In the S-dual description, the instantonic wall is simply the Weyl chamber wall of the dual $SU(2)$ gauge group, and it is therefore non-transversable.

Similar comments holds about the S-duality (6.1) for $N_f > 0$. We should also note that the $E_2$ and $E_3$ strongly-coupled SCFT Coulomb branches, $\mathcal{M}_{\mathcal{T}_\mathbf{x}}^C$, with $h_0 = 0$ and $m_i = 0$ (equivalently, $\mu = 0$ and $\nu = -\varphi < 0$), are only accessible in the "star-shaped" triangulations, Figures 10(a) and 12(a), respectively. The SCFT prepotential on $\mathcal{M}_{\mathcal{T}_\mathbf{x}}^C$ reads:

$$\mathcal{F} = \frac{8 - N_f}{6} \varphi^3 \ . \tag{6.4}$$

## 6.2 The beetle geometry and its gauge-theory phases

Let us now consider some more interesting "rank-two" geometry. The toric singularity is shown in Figure 16—for lack of a better name, we call it "the beetle." It has 24 distinct resolutions, shown in Figure 17. We easily see that there are 6 allowed vertical projections and 16 allowed horizontal projections. The four resolutions (a), (b), (c) and (d) admit both a vertical and a horizontal reduction.

It is also immediately clear that the vertical reduction leads to an $SU(2) \times SU(2)$ quiver gauge theory description, while the horizontal reduction leads to an $SU(3)$ gauge theory with $k_{\text{eff}} = 0$ and $N_f = 2$ fundamental flavors. Thus, we have the S-duality relation:

$$SU(2) \times SU(2) \qquad \longleftrightarrow \qquad SU(3)_0 \, , \, N_f = 2 \, . \tag{6.5}$$

We will explore this "UV duality" relation in the following. Let us insist again on the fact that, for most resolutions in Fig. 17, only one of the two gauge-theory descriptions is valid (or neither is valid, in the case of the resolutions (s) to (x)); yet, for the resolutions (a), (b), (d) and (e), both gauge-theory descriptions can be true *simultaneously*.

Note also that the toric diagram of Figure 17 is parity invariant, in the sense of (3.57), therefore the SCFT is parity invariant. This will be confirmed by the IIA analysis.

### 6.2.1 M-theory geometry and prepotential

The beetle singularity can be described by the following GLSM:

|  | $D_1$ | $D_2$ | $D_3$ | $D_4$ | $D_5$ | $D_6$ | $\mathbf{E}_1$ | $\mathbf{E}_2$ | FI |
|---|---|---|---|---|---|---|---|---|---|
| $\mathcal{C}_2^a$ | 0 | 1 | 0 | 0 | 0 | 0 | $-2$ | 1 | $\xi_2$ |
| $\mathcal{C}_3^a$ | 0 | 0 | $-1$ | 1 | 0 | 0 | 1 | $-1$ | $\xi_3$ |
| $\mathcal{C}_6^a$ | 1 | 0 | 0 | 0 | 1 | $-1$ | 0 | $-1$ | $\xi_6$ |
| $\mathcal{C}_7^a$ | $-1$ | 0 | 0 | 0 | 0 | 1 | 1 | $-1$ | $\xi_7$ |
| $\mathcal{C}_9^a$ | 1 | 0 | 1 | 0 | 0 | 0 | $-2$ | 0 | $\xi_9$ |

$$\tag{6.6}$$

Here, the rows are labelled by the curves in resolution (a), to be discussed below, in which case the FI terms $\xi_2, \xi_3, \xi_6, \xi_7$ and $\xi_9$ are all positive, but the same GLSM describes all 24 resolutions. We have the following relations amongst toric divisors:

$$\begin{aligned} D_1 + D_6 &\cong D_3 + D_4 \, , \\ \mathbf{E}_1 &\cong -D_1 - 2D_2 - D_3 + D_5 \, , \qquad \mathbf{E}_2 \cong D_2 - D_4 - 2D_5 - D_6 \, , \end{aligned} \tag{6.7}$$

which are independent of the particular resolution. Let us consider:

$$S = \mu_1 D_1 + \mu_2 D_2 + \mu_6 D_6 + \nu_1 \mathbf{E}_1 + \nu_2 \mathbf{E}_2 \, . \tag{6.8}$$

The map between the FI parameters in (6.6) and the parameters $\mu, \nu$ is:

$$\begin{aligned} \mu_1 &= 4\xi_3 - 2\xi_6 - 2\xi_7 + \xi_9 \, , & \nu_1 &= 2\xi_3 - \xi_6 - \xi_7 \, , \\ \mu_2 &= \xi_2 + 3\xi_3 - \xi_6 - \xi_7 \, , & \nu_2 &= \xi_3 - \xi_6 - \xi_7 \, . \\ \mu_6 &= 3\xi_3 - 2\xi_6 - \xi_7 + \xi_9 \, , \end{aligned} \tag{6.9}$$

The geometric prepotential:

$$\mathcal{F}(\mu, \nu) = \mathcal{F}(\xi) = -\frac{1}{6} S^3 \tag{6.10}$$

depends on the resolution. A complete list of the geometric prepotentials in the 24 Kähler chambers is given in Appendix C.

### 6.2.2 $SU(2) \times SU(2)$ phases

Consider the quiver theory consisting of two $SU(2)$ gauge groups, $SU(2)_{(1)} \times SU(2)_{(2)}$, with one hypermultiplet $\mathcal{H}$ in the bifundamental representation. We denote by $h_1, h_2$ and $\varphi_1, \varphi_2$

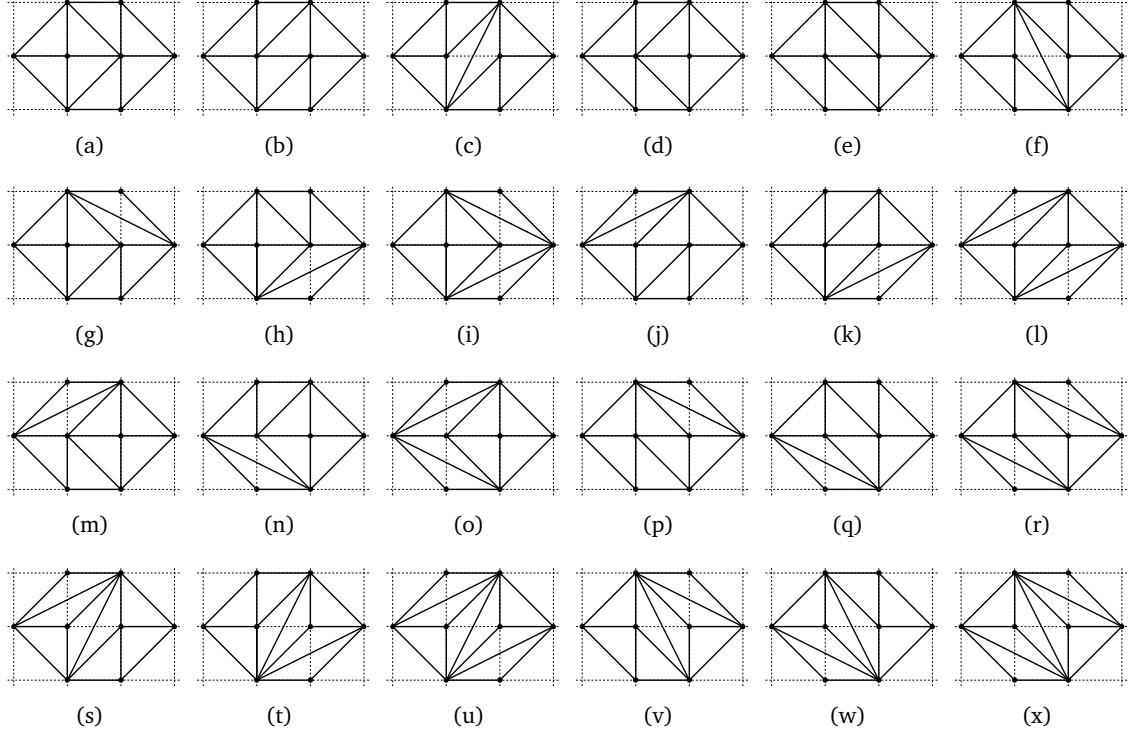

Figure 17: All 24 resolutions of the beetle geometry. The 6 resolutions (a)-(f) admit a vertical projection, the 16 resolutions (a), (b), (d), (e) and (g)-(r) admit a horizontal projection. Note that the 4 resolutions (a), (b), (d) and (e) admit both. The last 6 resolutions, (s)-(x), admit neither, hence do not have a gauge theory interpretation.

the two gauge couplings and the two Coulomb branch scalars, respectively, and by $m$ the mass of the bifundamental hypermultiplet. The prepotential reads:

$$
\begin{aligned}
\mathcal{F}_{SU(2)\times SU(2)} &= h_1\varphi_1^2 + h_2\varphi_2^2 + \frac{4}{3}\varphi_1^3 + \frac{4}{3}\varphi_2^3 \\
&\quad - \frac{1}{6}\Theta(\varphi_1 + \varphi_2 + m)(\varphi_1 + \varphi_2 + m)^3 \\
&\quad - \frac{1}{6}\Theta(\varphi_1 - \varphi_2 + m)(\varphi_1 - \varphi_2 + m)^3 \\
&\quad - \frac{1}{6}\Theta(-\varphi_1 + \varphi_2 + m)(-\varphi_1 + \varphi_2 + m)^3 \\
&\quad - \frac{1}{6}\Theta(-\varphi_1 - \varphi_2 + m)(-\varphi_1 - \varphi_2 + m)^3 .
\end{aligned}
\tag{6.11}
$$

By performing the vertical reduction, one can easily derive the precise relation between the geometry and field-theory parameters. In terms of the GLSM (6.6), we find:

$$
\begin{aligned}
\xi_2 &= h_1 + 2\varphi_1 - \varphi_2 - m , & \xi_3 &= -\varphi_1 + \varphi_2 - m , \\
\xi_6 &= h_2 + \varphi_2 - 2m , & \xi_7 &= -\varphi_1 + \varphi_2 + m , \\
\xi_9 &= 2\varphi_1 .
\end{aligned}
\tag{6.12}
$$

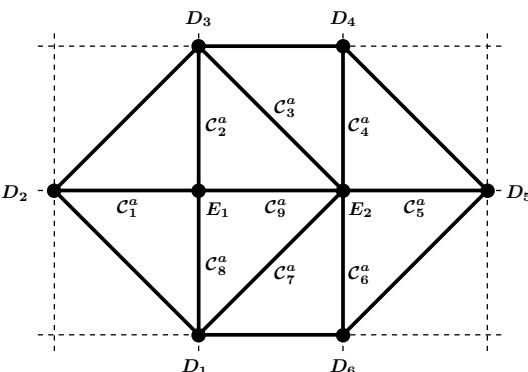

Figure 18: Resolution (a) of the beetle geometry, with all the curves indicated.

Equivalently, plugging the above into (6.9) we find:

$$
\begin{aligned}
&\mu_1 = -2h_2 - 2m\,, &\quad &\nu_1 = -\varphi_1 - h_2 - m\,,\\
&\mu_2 = h_1 - h_2 - 3m\,, &\quad &\nu_2 = -\varphi_2 - h_2\,,\\
&\mu_6 = -2h_2\,.
\end{aligned}
\tag{6.13}
$$

We will also find that the resolutions (a) to (f) from Figure 17 correspond to the following field-theory chambers:

$$
\begin{aligned}
\text{(a)} \quad &\longleftrightarrow \quad
\begin{cases}
\varphi_1 + \varphi_2 + m > 0\,, & \varphi_1 - \varphi_2 + m < 0\,,\\
-\varphi_1 + \varphi_2 + m > 0\,, & -\varphi_1 - \varphi_2 + m < 0\,,
\end{cases}\\[4pt]
\text{(b)} \quad &\longleftrightarrow \quad
\begin{cases}
\varphi_1 + \varphi_2 + m > 0\,, & \varphi_1 - \varphi_2 + m > 0\,,\\
-\varphi_1 + \varphi_2 + m > 0\,, & -\varphi_1 - \varphi_2 + m < 0\,,
\end{cases}\\[4pt]
\text{(c)} \quad &\longleftrightarrow \quad
\begin{cases}
\varphi_1 + \varphi_2 + m > 0\,, & \varphi_1 - \varphi_2 + m > 0\,,\\
-\varphi_1 + \varphi_2 + m > 0\,, & -\varphi_1 - \varphi_2 + m > 0\,,
\end{cases}\\[4pt]
\text{(d)} \quad &\longleftrightarrow \quad
\begin{cases}
\varphi_1 + \varphi_2 + m > 0\,, & \varphi_1 - \varphi_2 + m > 0\,,\\
-\varphi_1 + \varphi_2 + m < 0\,, & -\varphi_1 - \varphi_2 + m < 0\,,
\end{cases}\\[4pt]
\text{(e)} \quad &\longleftrightarrow \quad
\begin{cases}
\varphi_1 + \varphi_2 + m > 0\,, & \varphi_1 - \varphi_2 + m < 0\,,\\
-\varphi_1 + \varphi_2 + m < 0\,, & -\varphi_1 - \varphi_2 + m < 0\,,
\end{cases}\\[4pt]
\text{(f)} \quad &\longleftrightarrow \quad
\begin{cases}
\varphi_1 + \varphi_2 + m < 0\,, & \varphi_1 - \varphi_2 + m < 0\,,\\
-\varphi_1 + \varphi_2 + m < 0\,, & -\varphi_1 - \varphi_2 + m < 0\,.
\end{cases}
\end{aligned}
\tag{6.14}
$$

This is the expected match between the gauge-theory resolutions and the six distinct gauge-theory chambers. Note that we chose the fundamental $SU(2) \times SU(2)$ Weyl chamber, $\varphi_1 \geq 0$ and $\varphi_2 \geq 0$, by convention, while the real mass $m$ can be of either sign. In Appendix C, we check that the M-theory and field-theory prepotentials precisely match, in those 6 chambers.

Note also that only resolution (c) is compatible with the field-theory decoupling limit $m \to \infty$, and similarly only resolution (f) is compatible with the limit $m \to -\infty$. In these limits, we integrate out the bifundamental hypermultiplet and obtain two decoupled $SU(2)_\pi$ gauge theories. This is compatible with the toric geometry shown in Fig. 17(c) and Fig. 17(f), where $|m| \to \infty$ corresponds to blowing up a curve to infinite volume in such a way that the beetle singularity splits into two resolved $E_1$ singularities.

**Resolution (a).** Consider resolution (a) with the curves $\mathcal{C}_1^a, \cdots, \mathcal{C}_1^9$ as shown in Figure 18. We have the linear relations:

$$
\mathcal{C}_1^a \cong \mathcal{C}_9^a\,, \qquad \mathcal{C}_8^a \cong \mathcal{C}_2^a\,, \qquad \mathcal{C}_3^a + \mathcal{C}_4^a \cong \mathcal{C}_6^a + \mathcal{C}_7^a\,, \qquad \mathcal{C}_5^a \cong \mathcal{C}_3^a + \mathcal{C}_7^a + \mathcal{C}_9^a\,.
\tag{6.15}
$$

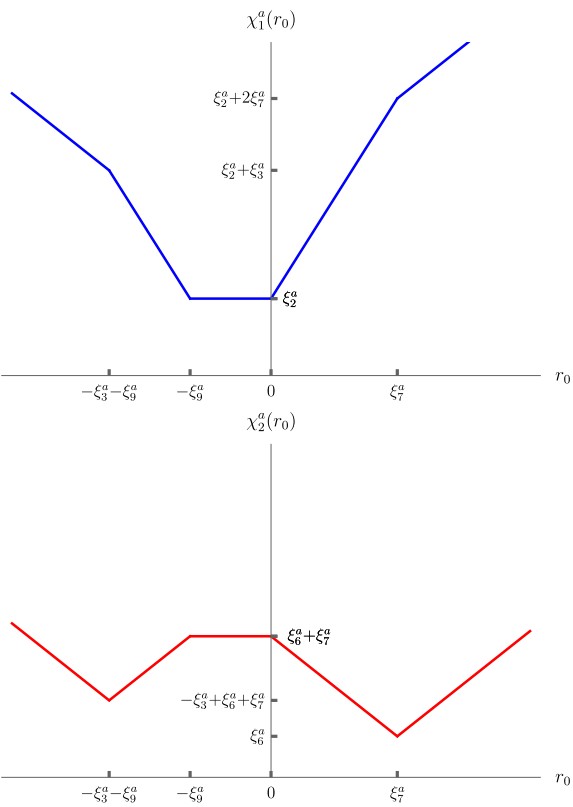

Figure 19: IIA profiles for the vertical reduction of resolution (a).

Let $\xi_i^a$ denote the Kähler volume of $\mathcal{C}_i^a$. The vertical reduction is obtained from the GLSM:

|          | $D_1$ | $D_2$ | $D_3$ | $D_4$ | $D_5$ | $D_6$ | $\mathbf{E}_1$ | $\mathbf{E}_2$ | FI |
|----------|-------|-------|-------|-------|-------|-------|------|------|--------|
| $\mathcal{C}_2^a$ | 0 | 1 | 0 | 0 | 0 | 0 | $-2$ | 1 | $\xi_2^a$ |
| $\mathcal{C}_3^a$ | 0 | 0 | $-1$ | 1 | 0 | 0 | 1 | $-1$ | $\xi_3^a$ |
| $\mathcal{C}_6^a$ | 1 | 0 | 0 | 0 | 1 | $-1$ | 0 | $-1$ | $\xi_6^a$ |
| $\mathcal{C}_7^a$ | $-1$ | 0 | 0 | 0 | 0 | 1 | 1 | $-1$ | $\xi_7^a$ |
| $\mathcal{C}_9^a$ | 1 | 0 | 1 | 0 | 0 | 0 | $-2$ | 0 | $\xi_9^a$ |
| $U(1)_M$ | $-1$ | 0 | 0 | 0 | 0 | 0 | 1 | 0 | $r_0$ |

(6.16)

As anticipated, this is the same as in (6.6), with $\xi_i^a \equiv \xi_i \geq 0$, and we choose the curves $\{\mathcal{C}_2^a, \mathcal{C}_3^a, \mathcal{C}_6^a, \mathcal{C}_7^a, \mathcal{C}_9^a\}$ to be the Mori cone generators, with the condition $-\xi_3^a + \xi_6^a + \xi_7^a \geq 0$ understood in the definition of the Kähler cone, due to the third relation in (6.15).

Upon vertical reduction, the IIA background consists a (resolved) $A_2$ singularity with the exceptional curves $\mathbb{P}_s^1$ ($s = 1, 2$), with two D6-branes wrapping each $\mathbb{P}^1$, thus realizing the $SU(2) \times SU(2)$ gauge group, with the single bifundamental arising at the intersection of the $\mathbb{P}^1$'s. The $A_2$ singularity is non-trivially fibered over $\mathbb{R} \cong \{r_0\}$, with:

$$\chi_1^a(r_0) = \begin{cases} \xi_2 + \xi_7 + r_0 & \text{if} \quad r_0 \geq \xi_7, \\ \xi_2 + 2r_0 & \text{if} \quad 0 \leq r_0 \leq \xi_7, \\ \xi_2 & \text{if} \quad -\xi_9 \leq r_0 \leq 0, \\ \xi_2 - 2\xi_9 - 2r_0 & \text{if} \quad -\xi_3 - \xi_9 \leq r_0 \leq -\xi_9, \\ \xi_2 + \xi_3 - \xi_9 - r_0 & \text{if} \quad r_0 \leq -\xi_3 - \xi_9, \end{cases}$$

(6.17)

and

$$
\chi_2^a(r_0) =
\begin{cases}
\xi_6 - \xi_7 + r_0 & \text{if} \quad r_0 \geq \xi_7, \\
\xi_6 + \xi_7 - r_0 & \text{if} \quad 0 \leq r_0 \leq \xi_7, \\
\xi_6 + \xi_7 & \text{if} \quad -\xi_9 \leq r_0 \leq 0, \\
\xi_6 + \xi_7 + \xi_9 + r_0 & \text{if} \quad -\xi_3 - \xi_9 \leq r_0 \leq -\xi_9, \\
-2\xi_3 + \xi_6 + \xi_7 - \xi_9 - r_0 & \text{if} \quad r_0 \leq -\xi_3 - \xi_9,
\end{cases}
\tag{6.18}
$$

for $\mathbb{P}_1^1$ and $\mathbb{P}_2^1$, respectively. This IIA profile is shown in Figure 19. Note that $k_{s,\text{eff}} = 0$ here (as well as for all the other gauge-theory chambers), so the theory is indeed parity-invariant.

From this IIA description, we immediately read off the perturbative particles. The W-bosons for both gauge groups have masses:

$$
M(W_{(1)}) = 2\varphi_1 = \xi_9 , \qquad M(W_{(2)}) = 2\varphi_2 = \xi_3 + \xi_7 + \xi_9 ,
\tag{6.19}
$$

and the masses of the four hypermultiplets modes are:

$$
\begin{aligned}
M(\mathcal{H}_{++}) &= \varphi_1 + \varphi_2 + m = \xi_7 + \xi_9 , & M(\mathcal{H}_{+-}) &= -\varphi_1 + \varphi_2 - m = \xi_3 , \\
M(\mathcal{H}_{-+}) &= -\varphi_1 + \varphi_2 + m = \xi_7 , & M(\mathcal{H}_{--}) &= \varphi_1 + \varphi_2 - m = \xi_3 + \xi_9 .
\end{aligned}
\tag{6.20}
$$

Finally, one can compute the instanton masses:

$$
\begin{aligned}
M(\mathcal{I}_{(1),1}) = M(\mathcal{I}_{(1),2}) &= h_1 + 2\varphi_1 - \varphi_2 - m = \xi_2 , \\
M(\mathcal{I}_{(2),1}) = h_2 + \varphi_2 - 2m = \xi_6 , & M(\mathcal{I}_{(2),2}) = h_2 + \varphi_2 = -\xi_3 + \xi_6 + \xi_7 .
\end{aligned}
\tag{6.21}
$$

This completes the derivation of the geometry-to-gauge-theory map (6.12). One can also easily check that the string tensions computed from the IIA geometry agree with the field theory result:

$$
T_1^{(a)} = \partial_{\varphi_1} \mathcal{F}^{(a)} = \int_{-\xi_9}^0 \chi_1(r_0) dr_0 , \qquad T_2^{(a)} = \partial_{\varphi_2} \mathcal{F}^{(a)} = \int_{-\xi_3 - \xi_9}^{\xi_7} \chi_2(r_0) dr_0 .
\tag{6.22}
$$

Here, $\mathcal{F}^{(a)}$ denotes the prepotential (6.11) in the gauge-theory chamber (a), as defined in (6.14).

**Resolution (b).** Consider resolution (b), shown in Figure 20(a). This is obtained from resolution (a) by flopping the curve $\mathcal{C}_3^a$. The relations amongst the curves are:

$$
\mathcal{C}_2^b + \mathcal{C}_3^b \cong \mathcal{C}_8^b, \qquad \mathcal{C}_6^b + \mathcal{C}_7^b \cong \mathcal{C}_4^b, \qquad \mathcal{C}_1^b \cong \mathcal{C}_3^b + \mathcal{C}_9^b, \qquad \mathcal{C}_5^b = \mathcal{C}_7^b + \mathcal{C}_9^b .
\tag{6.23}
$$

We can take $\{\mathcal{C}_2^b, \mathcal{C}_3^b, \mathcal{C}_6^b, \mathcal{C}_7^b, \mathcal{C}_9^b\}$ to generate the Mori cone. The Kähler volumes of the curves $\mathcal{C}_i^b$, denoted by $\xi_i^b$, are related to the FI terms $\xi_i$ of the GLSM (6.6) by:

$$
\xi_2^b = \xi_2 + \xi_3 , \qquad \xi_3^b = -\xi_3 , \qquad \xi_6^b = \xi_6 , \qquad \xi_7^b = \xi_7 , \qquad \xi_9^b = \xi_9 + \xi_3 ,
\tag{6.24}
$$

The IIA background is easily obtained from the GLSM:

|  | $D_1$ | $D_2$ | $D_3$ | $D_4$ | $D_5$ | $D_6$ | $\mathbf{E}_1$ | $\mathbf{E}_2$ | FI |
|---|---|---|---|---|---|---|---|---|---|
| $\mathcal{C}_2^b$ | 0 | 1 | −1 | 1 | 0 | 0 | −1 | 0 | $\xi_2^b$ |
| $\mathcal{C}_3^b$ | 0 | 0 | 1 | −1 | 0 | 0 | −1 | 1 | $\xi_3^b$ |
| $\mathcal{C}_6^b$ | 1 | 0 | 0 | 0 | 1 | −1 | 0 | −1 | $\xi_6^b$ |
| $\mathcal{C}_7^b$ | −1 | 0 | 0 | 0 | 0 | 1 | 1 | −1 | $\xi_7^b$ |
| $\mathcal{C}_9^b$ | 1 | 0 | 0 | 1 | 0 | 0 | −1 | −1 | $\xi_9^b$ |
| $U(1)_M$ | −1 | 0 | 0 | 0 | 0 | 0 | 1 | 0 | $r_0$ |

$$
\tag{6.25}
$$

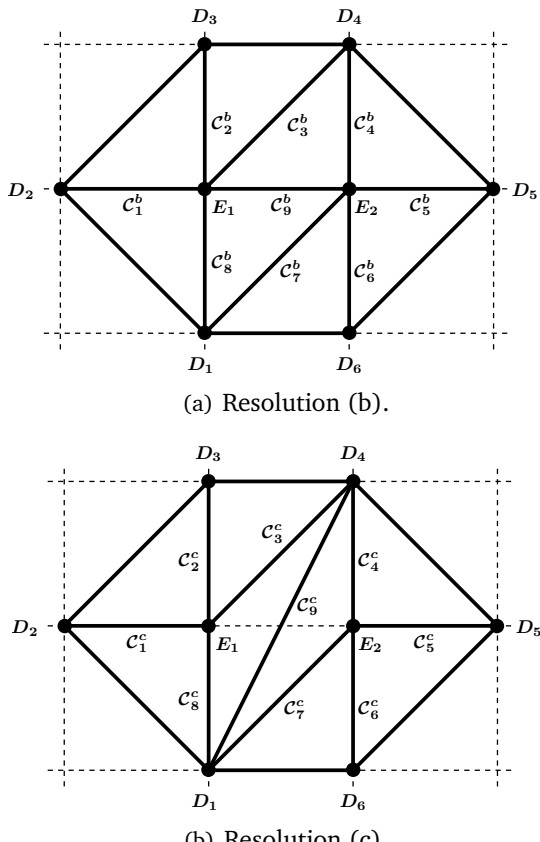

(a) Resolution (b).

(b) Resolution (c).

Figure 20: Resolutions (b) and (c), with all the curves indicated.

This is equivalent to (6.16), with the rows corresponding to the Mori cone generators. We find:

$$\chi_1^b(r_0) = \begin{cases} \xi_2^b + \xi_3^b + \xi_7^b + r_0 & \text{if } r_0 \geq \xi_7^b, \\ \xi_2^b + \xi_3^b + 2r_0 & \text{if } 0 \leq r_0 \leq \xi_7^b, \\ \xi_2^b + \xi_3^b & \text{if } -\xi_9^b \leq r_0 \leq 0, \\ \xi_2^b + \xi_3^b + \xi_9^b + r_0 & \text{if } -\xi_3^b - \xi_9^b \leq r_0 \leq -\xi_9^b, \\ \xi_2^b - \xi_3^b - \xi_9^b - r_0 & \text{if } r_0 \leq -\xi_3^b - \xi_9^b, \end{cases} \quad (6.26)$$

$$\chi_2^b(r_0) = \begin{cases} \xi_6^b - \xi_7^b + r_0 & \text{if } r_0 \geq \xi_7^b, \\ \xi_6^b + \xi_7^b - r_0 & \text{if } 0 \leq r_0 \leq \xi_7^b, \\ \xi_6^b + \xi_7^b & \text{if } -\xi_3^b \leq r_0 \leq 0, \\ \xi_6^b + \xi_7^b - 2\xi_9^b - 2r_0 & \text{if } -\xi_3^b - \xi_9^b, \leq r_0 \leq -\xi_9^b, \\ \xi_3^b + \xi_6^b + \xi_7^b - \xi_9^b - r_0 & \text{if } r_0 \leq -\xi_3^b - \xi_9^b. \end{cases} \quad (6.27)$$

This is pictured in Fig. 21. From this profile, we read off the W-bosons and hypermultiplet masses:

$$
\begin{aligned}
M(W_{(1)}) &= 2\varphi_1 = \xi_3^b + \xi_9^b, & M(W_{(2)}) &= 2\varphi_2 = \xi_7^b + \xi_9^b, \\
M(\mathcal{H}_{++}) &= \varphi_1 + \varphi_2 + m = \xi_3^b + \xi_7^b + \xi_9^b, & M(\mathcal{H}_{+-}) &= \varphi_1 - \varphi_2 + m = \xi_3^b, \\
M(\mathcal{H}_{-+}) &= -\varphi_1 + \varphi_2 + m = \xi_7^b, & M(\mathcal{H}_{--}) &= \varphi_1 + \varphi_2 - m = \xi_9^b.
\end{aligned} \quad (6.28)
$$

We also have the instanton masses:

$$
\begin{aligned}
M(\mathcal{I}_{(1),1}) &= h_1 + \varphi_1 - 2m = \xi_2^b, & M(\mathcal{I}_{(1),2}) &= h_1 + 2\varphi_1 - \varphi_2 - m = \xi_2^b + \xi_3^b, \\
M(\mathcal{I}_{(2),1}) &= h_2 + \varphi_2 - 2m = \xi_6^b, & M(\mathcal{I}_{(2),2}) &= h_2 - \varphi_1 + 2\varphi_2 - m = \xi_6^b + \xi_7^b.
\end{aligned} \quad (6.29)
$$

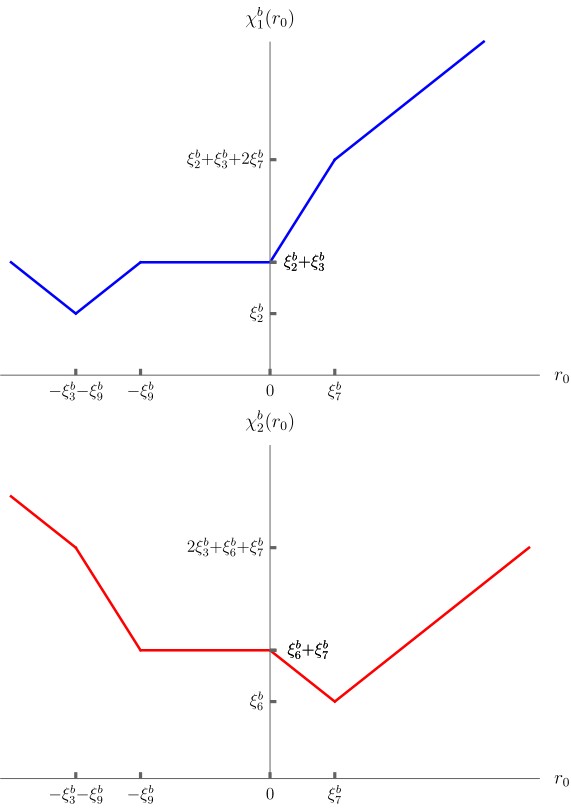

Figure 21: IIA profiles for the vertical reduction of resolution (b).

Of course, this reproduces (6.12). The string tensions similarly match.

**Resolution (c).** Consider the resolution (c), shown in Figure 20(b). It can be obtained from resolution (b) by flopping the curve $\mathcal{C}_9^b$ to $\mathcal{C}_9^c \cong -\mathcal{C}_9^b$. Here, we have the following linear relations amongst curves:

$$\mathcal{C}_1^c \cong \mathcal{C}_3^c, \qquad \mathcal{C}_4^c \cong \mathcal{C}_6^c + \mathcal{C}_7^c, \qquad \mathcal{C}_5^c \cong \mathcal{C}_7^c, \qquad \mathcal{C}_8^c \cong \mathcal{C}_2^c + \mathcal{C}_3^c. \tag{6.30}$$

Thus we can choose $\{\mathcal{C}_2^c, \mathcal{C}_3^c, \mathcal{C}_6^c, \mathcal{C}_7^c, \mathcal{C}_9^c\}$ as generators of the Mori cone. The Kähler volumes $\xi_i^c$ are related to the FI terms $\xi_i$ in (6.6) by:

$$\xi_2^c = \xi_2 + \xi_3, \qquad \xi_3^c = \xi_9, \qquad \xi_6^c = \xi_6, \qquad \xi_7^c = \xi_3 + \xi_7 + \xi_9, \qquad \xi_9^c = -\xi_3 - \xi_9. \tag{6.31}$$

We derive the IIA background from the GLSM:

|  | $D_1$ | $D_2$ | $D_3$ | $D_4$ | $D_5$ | $D_6$ | $E_1$ | $E_2$ | FI |
|---|---|---|---|---|---|---|---|---|---|
| $\mathcal{C}_2^c$ | 0 | 1 | −1 | 1 | 0 | 0 | −1 | 0 | $\xi_2^c$ |
| $\mathcal{C}_3^c$ | 1 | 0 | 1 | 0 | 0 | 0 | −2 | 0 | $\xi_3^c$ |
| $\mathcal{C}_6^c$ | 1 | 0 | 0 | 0 | 1 | −1 | 0 | −1 | $\xi_6^c$ |
| $\mathcal{C}_7^c$ | 0 | 0 | 0 | 1 | 0 | 1 | 0 | −2 | $\xi_7^c$ |
| $\mathcal{C}_9^c$ | −1 | 0 | 0 | −1 | 0 | 0 | 1 | 1 | $\xi_9^c$ |
| $U(1)_M$ | −1 | 0 | 0 | 0 | 0 | 0 | 1 | 0 | $r_0$ |

$$\tag{6.32}$$

We find the following profiles for $\chi_I(r_0)$:

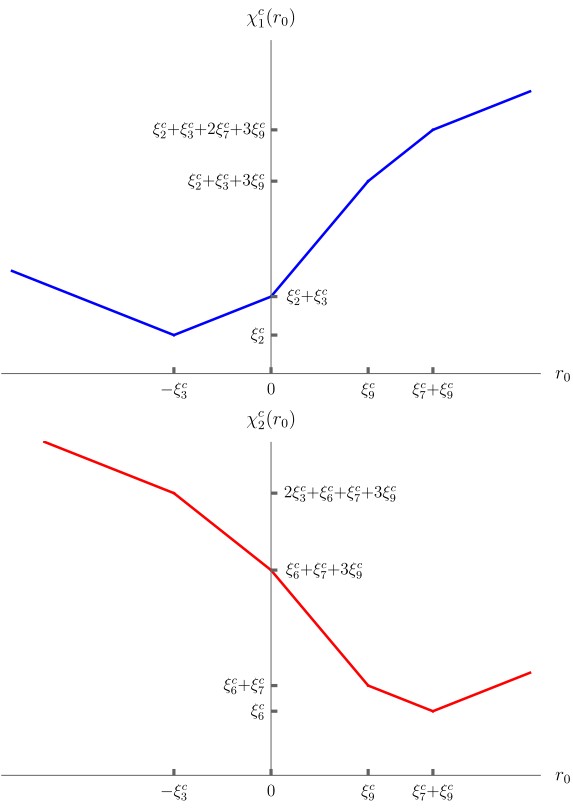

Figure 22: IIA profiles for the vertical reduction of resolution (c).

$$
\chi_1^c(r_0) = \begin{cases} \xi_2^c + \xi_3^c + \xi_7^c + 2\xi_9^c + r_0 & \text{if} \quad r_0 \geq \xi_7^c + \xi_9^c, \\ \xi_2^c + \xi_3^c + \xi_9^c + 2r_0 & \text{if} \quad \xi_9^c \leq r_0 \leq \xi_7^c + \xi_9^c, \\ \xi_2^c + \xi_3^c + 3r_0 & \text{if} \quad 0 \leq r_0 \leq \xi_9^c, \\ \xi_2^c + \xi_3^c + r_0 & \text{if} \quad -\xi_3^c \leq r_0 \leq 0, \\ \xi_2^c - \xi_3^c - r_0 & \text{if} \quad r_0 \leq -\xi_3^c, \end{cases}
\tag{6.33}
$$

$$
\chi_2^c(r_0) = \begin{cases} \xi_6^c - \xi_7^c - \xi_9^c + r_0 & \text{if} \quad r_0 \geq \xi_7^c + \xi_9^c, \\ \xi_6^c + \xi_7^c + \xi_9^c - r_0 & \text{if} \quad \xi_9^c \leq r_0 \leq \xi_7^c + \xi_9^c, \\ \xi_6^c + \xi_7^c + 3\xi_9^c - 3r_0 & \text{if} \quad 0 \leq r_0 \leq \xi_9^c, \\ \xi_6^c + \xi_7^c + 3\xi_9^c - 2r_0 & \text{if} \quad -\xi_3^c \leq r_0 \leq 0, \\ \xi_3^c + \xi_6^c + \xi_7^c + 3\xi_9^c - r_0 & \text{if} \quad r_0 \leq -\xi_3^c. \end{cases}
\tag{6.34}
$$

This is shown in Figure 22. The masses of the perturbative particles are:

$$
\begin{aligned}
M(W_{(1)}) &= 2\varphi_1 = \xi_3^c, & M(W_{(2)}) &= 2\varphi_2 = \xi_7^c, \\
M(\mathcal{H}_{++}) &= \varphi_1 + \varphi_2 + m = \xi_3^c + \xi_7^c + \xi_9^c, & M(\mathcal{H}_{+-}) &= \varphi_1 - \varphi_2 + m = \xi_3^c + \xi_9^c, \\
M(\mathcal{H}_{-+}) &= -\varphi_1 + \varphi_2 + m = \xi_7^c + \xi_9^c, & M(\mathcal{H}_{--}) &= -\varphi_1 - \varphi_2 + m = \xi_9^c,
\end{aligned}
\tag{6.35}
$$

and the instanton masses are

$$
\begin{aligned}
M(\mathcal{I}_{(1),1}) &= h_1 + \varphi_1 - 2m = \xi_2^c, & M(\mathcal{I}_{(1),2}) &= h_1 + 3\varphi_1 - 2m = \xi_2^c + \xi_3^c, \\
M(\mathcal{I}_{(2),1}) &= h_2 + \varphi_2 - 2m = \xi_6^c, & M(\mathcal{I}_{(2),2}) &= h_2 + 3\varphi_2 - 2m = \xi_6^c + \xi_7^c.
\end{aligned}
\tag{6.36}
$$

As mentioned above, this resolution is compatible with the limit $m \to \infty$, corresponding to $\xi_9^c \to \infty$. That this limit gives rise to two $SU(2)_\pi$ gauge groups is readily apparent from Figure 22.

**Resolution (d), (e) and (f).** The three other resolutions admitting a vertical reduction are completely similar, and can be obtained from the cases (a), (b) and (c), respectively, by a reflection along the vertical axis, thus exchanging the role of the two $SU(2)$ gauge groups.

### 6.2.3 $SU(3)$ **phases**

The horizontal reduction of the beetle geometry, when allowed, leads to an $SU(3)$ gauge theory coupled to two hypermultiplets in the fundamental representation, with $k_{\text{eff}=0}$. The prepotential of this gauge theory reads:

$$
\begin{aligned}
\mathcal{F}^{N_f=2}_{SU(3)} = {} & \widetilde{h}_0(\widetilde{\varphi}_1^2 + \widetilde{\varphi}_2^2 - \widetilde{\varphi}_1\widetilde{\varphi}_2) + \frac{1}{2}(\widetilde{\varphi}_1^2\widetilde{\varphi}_2 - \widetilde{\varphi}_1\widetilde{\varphi}_2^2) \\
& + \frac{4}{3}(\widetilde{\varphi}_1^3 + \widetilde{\varphi}_2^3) - \frac{1}{2}(\widetilde{\varphi}_1^2\widetilde{\varphi}_2 + \widetilde{\varphi}_1\widetilde{\varphi}_2^2) - \frac{1}{6}\sum_{i=1}^{2}\Big(\Theta(\widetilde{\varphi}_1 + \widetilde{m}_i)(\widetilde{\varphi}_1 + \widetilde{m}_i)^3 \\
& + \Theta(-\widetilde{\varphi}_1 + \widetilde{\varphi}_2 + \widetilde{m}_i)(-\widetilde{\varphi}_1 + \widetilde{\varphi}_2 + \widetilde{m}_i)^3 + \Theta(-\widetilde{\varphi}_2 + \widetilde{m}_i)(-\widetilde{\varphi}_2 + \widetilde{m}_i)^3\Big).
\end{aligned}
\tag{6.37}
$$

Here and in the rest of this section, we dress the $SU(3)$ variables with tildes, to distinguish them from the $SU(2) \times SU(2)$ gauge-theory variables. Note that the last term on the first line of (C.8) is a bare $SU(3)$ CS level $k = 1$, which is necessary so that $k_{\text{eff}} = 0$, given our choice of "$\kappa = -\frac{1}{2}$" quantization for the two fundamental hypermultiplets.

We have chosen the fundamental Weyl chamber:

$$
2\widetilde{\varphi}_1 - \widetilde{\varphi}_2 \geq 0, \qquad -\widetilde{\varphi}_1 + 2\widetilde{\varphi}_2 \geq 0.
\tag{6.38}
$$

There are 16 distinct gauge-theory chambers that can be obtained by varying $\widetilde{\varphi}_1, \widetilde{\varphi}_2$ and the real masses $\widetilde{m}_1, \widetilde{m}_2 \in \mathbb{R}$. The chambers correspond to all the possible choices of signs for the masses of the 6 hypermultiplets modes $\widetilde{\varphi}_1 + \widetilde{m}_i$, $-\widetilde{\varphi}_1 + \widetilde{\varphi}_2 + \widetilde{m}_i$, and $-\widetilde{\varphi}_2 + \widetilde{m}_i$ ($i = 1, 2$), compatible with the Weyl-chamber condition (6.38).

We will now derive the following map between the GLSM parameters and the $SU(3)$ variables:

$$
\begin{aligned}
\xi_2 &= 2\widetilde{\varphi}_1 - \widetilde{\varphi}_2, & \xi_3 &= -\widetilde{\varphi}_1 + \widetilde{\varphi}_2 + \widetilde{m}_1, \\
\xi_6 &= \widetilde{\varphi}_2 - \widetilde{m}_2, & \xi_7 &= -\widetilde{\varphi}_1 + \widetilde{\varphi}_2 + \widetilde{m}_2, \\
\xi_9 &= \widetilde{h}_0 + 2\widetilde{\varphi}_1 - \widetilde{m}_1 - \widetilde{m}_2.
\end{aligned}
\tag{6.39}
$$

Equivalently, we find:

$$
\begin{aligned}
\mu_1 &= \widetilde{h}_0 + 3\widetilde{m}_1 - \widetilde{m}_2, & \nu_1 &= -\widetilde{\varphi}_1 + 2\widetilde{m}_1, \\
\mu_2 &= 3\widetilde{m}_1, & \nu_2 &= -\varphi_2 + \widetilde{m}_1, \\
\mu_6 &= \widetilde{h}_0 + 2\widetilde{m}_1.
\end{aligned}
\tag{6.40}
$$

By comparing to (6.13), this implies an "S-duality map" which we discuss in subsection 6.2.4 below.

**Resolution (a).** Consider the horizontal reduction of resolution (a) in Fig. 18. We use the same GLSM as in (6.16), but with a different $U(1)_M$ charge:

$$
\begin{array}{c|cccccccc|c}
 & D_1 & D_2 & D_3 & D_4 & D_5 & D_6 & \mathbf{E}_1 & \mathbf{E}_2 & \text{FI} \\
\hline
U(1)_{M'} & 0 & 0 & 0 & 0 & 0 & 0 & -1 & 1 & r_0
\end{array}
\tag{6.41}
$$

The resulting IIA background consists of a resolved $A_1$ singularity fibered over $\mathbb{R}$, with three D6-branes wrapped over the exceptional $\mathbb{P}^1$, and with two D6-branes wrapped over non-compact

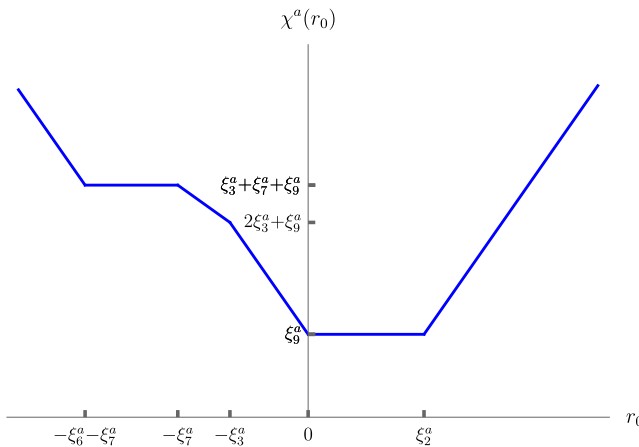

Figure 23: IIA profiles for the horizontal reduction of resolution (a), giving rise an an $SU(3)$ theory with two flavors.

divisors $\cong \mathbb{C}$, thus giving rise to an $SU(3)$ theory with two flavors. The profile $\chi(r_0)$ reads:

$$
\chi^a(r_0) = \begin{cases}
-2\xi_2 + \xi_9 + 2r_0 & \text{if} \quad r_0 \geq \xi_2, \\
\xi_9 & \text{if} \quad 0 \leq r_0 \leq \xi_2, \\
\xi_9 - 2r_0 & \text{if} \quad -\xi_3 \leq r_0 \leq 0, \\
\xi_3 + \xi_9 - r_0 & \text{if} \quad -\xi_7 \leq r_0 \leq -\xi_3, \\
\xi_3 + \xi_7 + \xi_9 & \text{if} \quad -\xi_6 - \xi_7 \leq r_0 \leq -\xi_7, \\
\xi_3 - 2\xi_6 - \xi_7 + \xi_9 - 2r_0 & \text{if} \quad r_0 \leq -\xi_6 - \xi_7.
\end{cases}
\tag{6.42}
$$

Recall that $\xi_i = \xi_i^a$. From the IIA profile (6.42), shown in Figure 23, we directly read off the effective CS level $k_{\text{eff}} = 0$, the masses of the $SU(3)$ W-bosons:

$$
M(W_1) = 2\widetilde{\varphi}_1 - \widetilde{\varphi}_2 = \xi_2 , \qquad M(W_2) = -\widetilde{\varphi}_1 + 2\widetilde{\varphi}_2 = \xi_6 + \xi_7 ,
\tag{6.43}
$$

and of the hypermultiplet modes:

$$
\begin{aligned}
M(\mathcal{H}_{1,1}) &= \widetilde{\varphi}_1 + \widetilde{m}_1 = \xi_2 + \xi_3 , & M(\mathcal{H}_{1,2}) &= \widetilde{\varphi}_1 + \widetilde{m}_2 = \xi_2 + \xi_7 , \\
M(\mathcal{H}_{2,1}) &= -\widetilde{\varphi}_1 + \widetilde{\varphi}_2 + \widetilde{m}_1 = \xi_3 , & M(\mathcal{H}_{2,2}) &= -\widetilde{\varphi}_1 + \widetilde{\varphi}_2 + \widetilde{m}_2 = \xi_7 , \\
M(\mathcal{H}_{3,1}) &= \widetilde{\varphi}_2 - \widetilde{m}_1 = -\xi_3 + \xi_6 + \xi_7 , & M(\mathcal{H}_{3,2}) &= \widetilde{\varphi}_2 - \widetilde{m}_2 = \xi_6 .
\end{aligned}
\tag{6.44}
$$

The instanton particle masses are given by:

$$
M(\mathcal{I}_1) = M(\mathcal{I}_2) = \widetilde{h}_0 + 2\widetilde{\varphi}_1 - \widetilde{m}_1 - \widetilde{m}_2 = \xi_9 , \qquad M(\mathcal{I}_3) = \widetilde{h}_0 + 2\widetilde{\varphi}_2 = \xi_3 + \xi_7 + \xi_9 . \tag{6.45}
$$

This establishes the map (6.39) between the geometry and the $SU(3)$ gauge-theory parameters. As usual, one can also match the magnetic string tensions.

From this analysis, we find that this resolution corresponds to the field-theory chamber:

$$
\widetilde{\varphi}_1 + \widetilde{m}_i > 0 , \qquad -\widetilde{\varphi}_1 + \widetilde{\varphi}_2 + \widetilde{m}_i > 0 , \qquad -\widetilde{\varphi}_2 + \widetilde{m}_i < 0 , \qquad i = 1, 2 .
\tag{6.46}
$$

By successive flops, it is straightforward to map the 16 different gauge-theory chambers to the 16 resolutions with an allowed horizontal reduction. For instance, by flopping the curve $\mathcal{C}_4^a \cong -\mathcal{C}_3^a + \mathcal{C}_6^a + \mathcal{C}_7^a$ we obtain resolution (g)—Figure 17(g). From (6.44), we see that this corresponds to changing the sign of the mass of the hypermultiplet mode $\mathcal{H}_{3,1}$. If, in addition, we also flop the curve $\mathcal{C}_6^a$, we obtain resolution (i) in Figure 17(i), corresponding to also changing

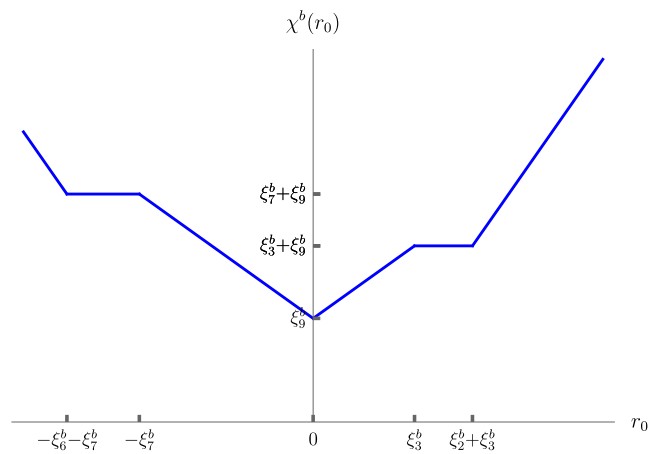

Figure 24: IIA profiles for the horizontal reduction of resolution (b).

the sign of $M(\mathcal{H}_{3,2})$. In that resolution, we have $\rho(\varphi) + m_1 > 0$ and $\rho(\varphi) + m_2 > 0$ for the fundamental hypermultiplets, and therefore we can consider the simultaneous limit $m_1 \to \infty$ and $m_2 \to \infty$. In the field theory, this gives us an $SU(3)_{-1}$ gauge theory without matter. The toric geometry in that limit is obtained by removing $D_4$ and $D_6$ from the toric diagram. This is in perfect agreement with our discussion of $SU(N)_k$ gauge theories in section 5.

**Resolution (b).** As another example, consider the horizontal reduction of resolution (b), which is obtained from resolution (a) by a single flop on $\mathcal{C}_3^a$—recall the relation (6.24) between the Kähler parameters. We find:

$$\chi^b(r_0) = \begin{cases} -2\xi_2^b - \xi_3^b + \xi_9^b + 2r_0 & \text{if } r_0 \geq \xi_2^b + \xi_3^b, \\ \xi_3^b + \xi_9^b & \text{if } \xi_3^b \leq r_0 \leq \xi_2^b + \xi_3^b, \\ \xi_9^b + r_0 & \text{if } 0 \leq r_0 \leq \xi_3^b, \\ \xi_9^b - r_0 & \text{if } -\xi_7^b \leq r_0 \leq 0, \\ \xi_7^b + \xi_9^b & \text{if } -\xi_6^b - \xi_7^b \leq r_0 \leq -\xi_7^b, \\ -2\xi_6^b - \xi_7^b + \xi_9^b - 2r_0 & \text{if } r_0 \leq -\xi_6^b - \xi_7^b. \end{cases} \tag{6.47}$$

This is shown in Figure 24. We have the perturbative particles:

$$\begin{aligned}
M(W_1) &= 2\widetilde{\varphi}_1 - \widetilde{\varphi}_2 = \xi_2^b + \xi_3^b, & M(W_2) &= -\widetilde{\varphi}_1 + 2\widetilde{\varphi}_2 = \xi_6^b + \xi_7^b, \\
M(\mathcal{H}_{1,1}) &= \widetilde{\varphi}_1 + \widetilde{m}_1 = \xi_2^b, & M(\mathcal{H}_{1,2}) &= \widetilde{\varphi}_1 + \widetilde{m}_2 = \xi_2^b + \xi_3^b + \xi_7^b, \\
M(\mathcal{H}_{2,1}) &= \widetilde{\varphi}_1 - \widetilde{\varphi}_2 - \widetilde{m}_1 = \xi_3^b, & M(\mathcal{H}_{2,2}) &= -\widetilde{\varphi}_1 + \widetilde{\varphi}_2 + \widetilde{m}_2 = \xi_7^b, \\
M(\mathcal{H}_{3,1}) &= \widetilde{\varphi}_2 - \widetilde{m}_1 = \xi_3^b + \xi_6^b + \xi_7^b, & M(\mathcal{H}_{3,2}) &= \widetilde{\varphi}_2 - \widetilde{m}_2 = \xi_6^b,
\end{aligned} \tag{6.48}$$

and the instanton particles:

$$\begin{aligned}
M(\mathcal{I}_1) &= \widetilde{h}_0 + 2\widetilde{\varphi}_1 - \widetilde{m}_1 - \widetilde{m}_2 = \xi_3^b + \xi_9^b, \\
M(\mathcal{I}_2) &= \widetilde{h}_0 + \widetilde{\varphi}_1 + \widetilde{\varphi}_2 - \widetilde{m}_2 = \xi_9^b, & M(\mathcal{I}_3) &= \widetilde{h}_0 + 2\widetilde{\varphi}_2 = \xi_7^b + \xi_9^b.
\end{aligned} \tag{6.49}$$

Of course, using (6.24), these relations also reproduce (6.39).

**Resolutions (d), (e) and (g)-(r).** All the other horizontal reductions can be performed similarly. The precise map between the resolutions and the 16 field-theory chambers is further discussed in Appendix C.

### 6.2.4 The S-duality map

By comparing the geometry-to-gauge-theory maps for the vertical and horizontal reductions, (6.12) and (6.39), respectively, we obtain an "S-duality map" between the $SU(3)_0, N_f = 2$ and the $SU(2) \times SU(2)$ gauge-theory parameters:

$$
\begin{aligned}
\widetilde{m}_1 &= \frac{1}{3}(h_1 - h_2) - m\,, & \widetilde{\varphi}_1 &= \varphi_1 + \frac{1}{3}(2h_1 + h_2) - m\,, \\
\widetilde{m}_2 &= \frac{1}{3}(h_1 - h_2) + m\,, & \widetilde{\varphi}_2 &= \varphi_2 + \frac{1}{3}(h_1 + 2h_2) - m\,, \qquad (6.50)\\
\widetilde{h}_0 &= -\frac{2}{3}(h_1 + 2h_2) + 2m\,.
\end{aligned}
$$

This same map was recently derived in [71] by studying Wilson loops. [31]

**S-duality for the resolution (a).** In resolution (a), we have the following dictionary between particle states in the two descriptions:

| Geometry: | $\mathcal{C}_1^a \cong \mathcal{C}_9^a$ | $\mathcal{C}_5^a$ | $\mathcal{C}_2^a \cong \mathcal{C}_8^a$ | $\mathcal{C}_6^a + \mathcal{C}_7^a$ | $\mathcal{C}_3^a$ | $\mathcal{C}_4^a$ | $\mathcal{C}_6^a$ | $\mathcal{C}_7^a$ |
|---|---|---|---|---|---|---|---|---|
| $SU(2) \times SU(2)$ | $W_{(1)}$ | $W_{(2)}$ | $\mathcal{I}_{(1),1}$ | $\mathcal{I}_{(2),1} \oplus \mathcal{H}_{-+}$ | $\mathcal{H}_{+-}$ | $\mathcal{I}_{(2),2}$ | $\mathcal{I}_{(2),1}$ | $\mathcal{H}_{-+}$ |
| $SU(3), N_f = 2$ | $\mathcal{I}_1$ | $\mathcal{I}_3$ | $W_1$ | $W_2$ | $\mathcal{H}_{2,1}$ | $\mathcal{H}_{3,1}$ | $\mathcal{H}_{3,2}$ | $\mathcal{H}_{2,2}$ |

The curves are as indicated in Fig. 18, with the relations (6.15). As expected from the $(p, q)$-web picture [5, 9], instanton particles are mapped to perturbatives particles in the S-dual description.

**S-duality for the resolution (b).** In resolution (b), we find:

| Geom: | $\mathcal{C}_1^b$ | $\mathcal{C}_5^b$ | $\mathcal{C}_8^b$ | $\mathcal{C}_4^b$ | $\mathcal{C}_2^b$ | $\mathcal{C}_3^b$ | $\mathcal{C}_6^b$ | $\mathcal{C}_7^b$ | $\mathcal{C}_9^b$ |
|---|---|---|---|---|---|---|---|---|---|
| $SU(2)^2$ | $W_{(1)}$ | $W_{(2)}$ | $\mathcal{I}_{(1),1} \oplus \mathcal{H}_{+-}$ | $\mathcal{I}_{(2),1} \oplus \mathcal{H}_{-+}$ | $\mathcal{I}_{(1),1}$ | $\mathcal{H}_{+-}$ | $\mathcal{I}_{(2),1}$ | $\mathcal{H}_{-+}$ | $\mathcal{H}_{--}$ |
| $SU(3)$ | $\mathcal{I}_1$ | $\mathcal{I}_3$ | $W_1$ | $W_2$ | $\mathcal{H}_{1,1}$ | $\mathcal{H}_{2,1}$ | $\mathcal{H}_{3,2}$ | $\mathcal{H}_{2,2}$ | $\mathcal{I}_2$ |

We obtain resolution (b) from resolution (a) by flopping $\mathcal{C}_3^a$ to $\mathcal{C}_3^b \cong -\mathcal{C}_3^a$. Since $\mathcal{C}_3^a$ corresponds to an hypermultiplet mode, $\mathcal{H}_{+-}$ or $\mathcal{H}_{2,1}$, in either gauge-theory theory description, we can cross this Kähler wall "perturbatively" in both descriptions, since this simply amounts to changing the sign of that hypermultiplet mass. This is in agreement with the fact that resolutions (a) and (b) both admit the two gauge-theory descriptions simultaneously.

**Some examples of phase transitions.** In general, more interesting phenomena can happen. Consider the flop of $\mathcal{C}_9^b$ in resolution (b), which gives rise to resolution (c). As discussed above, this simply corresponds to changing the sign of mass of $\mathcal{H}_{--}$ in the $SU(2) \times SU(2)$ description. On the other hand, in the $SU(3), N_f = 2$ description, this would correspond to changing the sign of $M(\mathcal{I}_2)$, the mass of an instanton particle, which cannot be done consistently in the low-energy $SU(3)$ gauge-theory language.

Moreover, in both resolutions (a) and (b), M2-branes wrapped over curves that cannot be flopped—namely, $\mathcal{C}_1^a$, $\mathcal{C}_2^a$ and $\mathcal{C}_5^a$ in resolution (a), and $\mathcal{C}_1^b$, $\mathcal{C}_4^b$, $\mathcal{C}_5^b$, $\mathcal{C}_8^b$ in resolution (b)—give rise to W-bosons in one of the two gauge-theory descriptions. When such a curve is blown down, we reach a non-traversable Kähler wall associated with the appearance of an $SU(2)$ non-abelian gauge symmetry [47]. Note that not all such walls are apparent from the point of

---

[31]Our result agrees with the ones of [71] up to some shifts of the gauge couplings by the real masses—for instance, our $\widetilde{h}_0$ is equal to $t - \frac{1}{2}(m_1 + m_2)$ in their notation (their $m_i$ is our $-\widetilde{m}_i$). These shifts arise from our different treatment of the parity anomaly.

view of a single low-energy gauge-theory description, since some of those walls also correspond to instanton particles becoming massless.

Interestingly, from the point of view of a given gauge theory, the walls corresponding to massless instanton particles may either be traversable or not; that determination requires to keep into account the string tensions. Let us consider resolution (a) in the $SU(2) \times SU(2)$ frame. In that chamber the string tensions are:

$$
\begin{aligned}
T_1^{(a)} &= 2\varphi_1(2\varphi_1 + h_1 - \varphi_2 - m) \,, \\
T_2^{(a)} &= -m^2 - \varphi_1^2 + \varphi_2(3\varphi_2 - 2m + 2h_2) \,.
\end{aligned}
\tag{6.51}
$$

The consistency of the effective field theory description requires that $T_1 > 0$ and $T_2 > 0$, which is an extra condition to be imposed on the Coulomb phase, and gives rise to hard walls, along $T_1 = 0$ or $T_2 = 0$. In this case, the masses of the various particles were computed in (6.21). We see that the instantons $\mathcal{I}_{(1),1}$ and $\mathcal{I}_{(1),2}$ have masses:

$$
M(\mathcal{I}_{(1),a}) = h_1 + 2\varphi_1 - \varphi_2 - m \,, \quad a = 1, 2 \,,
\tag{6.52}
$$

and become massless precisely along the hard wall $T_1 = 0$, hence the corresponding curves cannot be flopped. The instanton $\mathcal{I}_{(2),2}$ has mass $\varphi_2 + h_2$ and cannot become massless within this chamber by definition. Finally, the instanton $\mathcal{I}_{(2),1}$ has mass:

$$
M(\mathcal{I}_{(2),1}) = h_2 + \varphi_2 - 2m = \xi_6 \,.
\tag{6.53}
$$

We see that, while $T_1$ and $T_2$ remain positive throughout this chamber, we hit a wall at $\xi_6 = 0$, where:

$$
\varphi_2 = 2m - h_2 \,.
\tag{6.54}
$$

At this locus, the $SU(2) \times SU(2)$ description is no longer valid and the prepotential needs to be modified to account for the flop transition. Exploiting our map, we see that we are flopping curve $\mathcal{C}_6^a$ in figure 18. This maps to resolution (h), which has an $SU(3)$ description but no $SU(2) \times SU(2)$ interpretation. The particle that arise on the other side of the wall is an $SU(3)$ hypermultiplet with real mass $-\widetilde{\varphi}_2 + \widetilde{m}_2 > 0$.

Similarly, in the $SU(3), N_f = 2$ description of resolution (b), the instantons $\mathcal{I}_1$ and $\mathcal{I}_3$ are associated to non-traversable walls, while the instanton $\mathcal{I}_2$ is associated to a traversable wall, corresponding to flopping the curve $\mathcal{C}_9^b$. This is only apparent in the S-dual $SU(2) \times SU(2)$ description, where the two types of walls are associated to the W-bosons and to the hypermultiplet $\mathcal{H}_{--}$, respectively.

In summary, we see that distinct inequivalent gauge theory phases are interconnected in a non-trivial way along the Coulomb branch of the deformed beetle SCFT.

### 6.3 Probing the Coulomb branch of the 5d SCFT

So far in this section, we considered $\mu$ and $\nu$ generic. Then, we often have a low-energy gauge-theory interpretation at energies much lower than the mass scale set by $\mu$. In the rest of this section, we would like to explore the opposite limit, $\mu \to 0$ with $\nu$ finite. This corresponds to a "strong coupling limit" where the weakly-coupled gauge-theory approximation breaks down entirely. Nonetheless, it is instructive to use the gauge-theory language in order to gain some intuition about the SCFT Coulomb branch, $\mathcal{M}_C$, itself.

In rank-one cases, the $\mu = 0$ limit is somewhat uninteresting: the Coulomb branch has the form $\mathcal{M}_C \cong \mathbb{R}_+$—in the gauge-theory language, it is spanned by $\varphi > 0$, with $\varphi = 0$ the 5d fixed point.

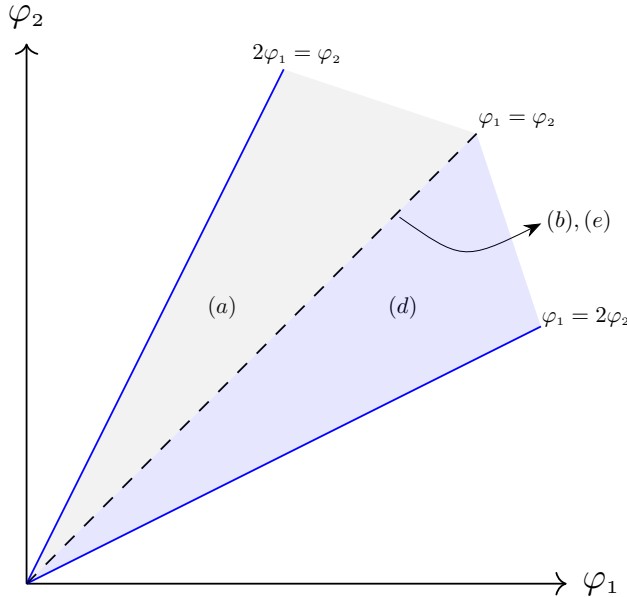

Figure 25: The Coulomb branch of the rank-two beetle SCFT is shown in the shaded area. It consists of two chambers, corresponding to resolutions (a) and (d). The middle wall corresponds to a degeneration of both resolutions (b) and (e).

Let us then consider the beetle geometry, which corresponds to a rank-two SCFT. Setting $\mu = 0$, we are left with two parameters:

$$\nu_1 = -\varphi_1 \,, \qquad \nu_2 = -\varphi_2 \,, \tag{6.55}$$

which can be thought of as living in the Cartan of either $SU(2) \times SU(2)$ or $SU(3)$. Indeed, in the limit when all the mass parameters vanish, the "S-duality map" (6.50) trivializes to $\varphi_1 = \widetilde{\varphi}_1$, $\varphi_2 = \widetilde{\varphi}_2$.

Moreover, out of the 24 Kähler chambers of Figure 17, only 4 survive in the massless limit. They are:

$$
\begin{aligned}
\text{chamber (a)} \quad &: \quad -\varphi_1 + \varphi_2 \geq 0 \,, \qquad 2\varphi_1 - \varphi_2 \geq 0 \,, \\
\text{chamber (b)} \quad &: \quad \varphi_1 = \varphi_2 \geq 0 \,, \\
\text{chamber (d)} \quad &: \quad \varphi_1 - \varphi_2 \geq 0 \,, \qquad -\varphi_1 + 2\varphi_2 \geq 0 \,, \\
\text{chamber (e)} \quad &: \quad \varphi_1 = \varphi_2 \geq 0 \,,
\end{aligned}
\tag{6.56}
$$

as depicted in Figure 25. Not coincidentally, these are the four resolutions that have both an $SU(2) \times SU(2)$ and an $SU(3)$ $N_f = 2$ field-theory interpretation. The prepotential in the $\mu = 0$ limit reads:

$$\mathcal{F}_a = \frac{4}{3}\varphi_1^3 - \varphi_1^2 \varphi_2 + \varphi_2^3 \,, \qquad \mathcal{F}_d = \frac{4}{3}\varphi_1^3 - \varphi_1^2 \varphi_2 + \varphi_2^3 \,, \qquad \mathcal{F}_b = \mathcal{F}_e = \frac{4}{3}\varphi_1^3 \,, \tag{6.57}$$

in the respective chambers. Note that the Kähler chambers (b) and (e) collapse to the line $\varphi_1 = \varphi_2$, at the interface between chamber (a) and chamber (d). That wall corresponds to a simultaneous flop of the curves $\mathcal{C}_3^a$ and $\mathcal{C}_7^a$ in Figure 18. In the gauge-theory language, the Kähler walls have the following interpretations:

$SU(2) \times SU(2)$ **interpretation.** The outer wall of chamber (a), at $2\varphi_1 - \varphi_2 = 0$ corresponds to an instanton particle for the first $SU(2)$ factor, $\mathcal{I}_{(1)}$, going massless. At the same time, a

magnetic string for that gauge group becomes tensionless at the wall, since:

$$T_1 = \partial_{\varphi_1} \mathcal{F}_a = 2\varphi_1(2\varphi_1 - \varphi_2) \,. \tag{6.58}$$

Similarly, at the outer wall of chamber (d), at $-\varphi_1 + 2\varphi_2 = 0$, the instanton particle $\mathcal{I}_{(2)}$ and the magnetic string of the second $SU(2)$ gauge group become massless. On the other hand, the middle wall at $\varphi_1 = \varphi_2$ corresponds to the bifundamental hypermultiplet of the $SU(2) \times SU(2)$ quiver going massless. (All magnetic strings are tensionfull there.)

Note that, naively, the $SU(2) \times SU(2)$ Coulomb branch spans the whole quadrant $\varphi_1 \geq 0$, $\varphi_2 \geq 0$. However, as pointed out in [22], one should exclude the regions beyond the walls at which some non-perturbative states (instantons or monopole string) become massless.

$SU(3)$ $N_f = 2$ **interpretation.** In the $SU(3)$ language, the outer walls are clearly hard walls, at which an $SU(3)$ W-boson goes massless. On the other hand, the middle wall at $\varphi_1 = \varphi_2$ simply corresponds to an hypermultiplet mode going massless.

# 7   Non-isolated toric singularities and the 5d $T_N$ theory

In our discussion so far, we restricted ourselves to the study of *isolated* toric CY threefold singularities. This corresponds to toric diagrams $\Gamma$ which are strictly convex: no external point $w \in \Gamma$ lies inside an external line. More general toric diagrams correspond to non-isolated singularities—see Figure 26 for some examples.

There are some well-known difficulties with non-isolated toric singularities, which are dual to $(p,q)$-webs with parallel external legs [5]; in particular, it is not clear that the low-energy theory at the singularity is really five-dimensional. In the $(p,q)$-web language, such difficulties can be alleviated by introducing 7-branes on which the external five-branes legs can end [27]. That approach, however, takes us outside of the realm of toric geometry on the M-theory side.

In this section, we limit ourselves to making some general comments about non-isolated toric singularities, including the toric realization of the five-dimensional $T_N$ theory [28], from the point of view of the M-theory/type IIA duality. We leave a more systematic study for future work.

## 7.1   Rank-one toric singularities and $T_2$ gauging

It is interesting to consider all the toric $CY_3$ singularities with a single exceptional divisor—that is, the toric diagrams with a single internal point. There are only 16 of them, up to an $SL(2, \mathbb{Z})$ transformation, corresponding to the 16 two-dimensional toric Fano varieties [93]. Out of those 16, only 5 are isolated singularties, corresponding to the smooth del Pezzo surfaces $dP_n$, with $n \leq 3$, which we studied in section 4—see Figure 6.

We display the toric diagrams for 10 of the 11 non-isolated singularities in Figure 26, in an $SL(2, \mathbb{Z})$ frame such that they all admit an obvious vertical reduction (for some of the possible triangulations). Conveniently, 9 of those 10 fit neatly in a family of singularities which we will denote by:

$$E_n^{(h_L, h_R)} \,, \qquad n = h_L + h_R + 1 \leq 5 \,. \tag{7.1}$$

The toric diagram contains the following $n + 4$ points:

$$E_n^{(h_L, h_R)} \; : \qquad \Gamma = \begin{cases} w_i^L = (-1, i) \,, & i = 0, \cdots, h_L \,, \\ w_j = (0, j) \,, & j = 0, 1, 2 \,, \\ w_k^R = (1, k) \,, & k = 0, \cdots, h_R \,, \end{cases} \tag{7.2}$$

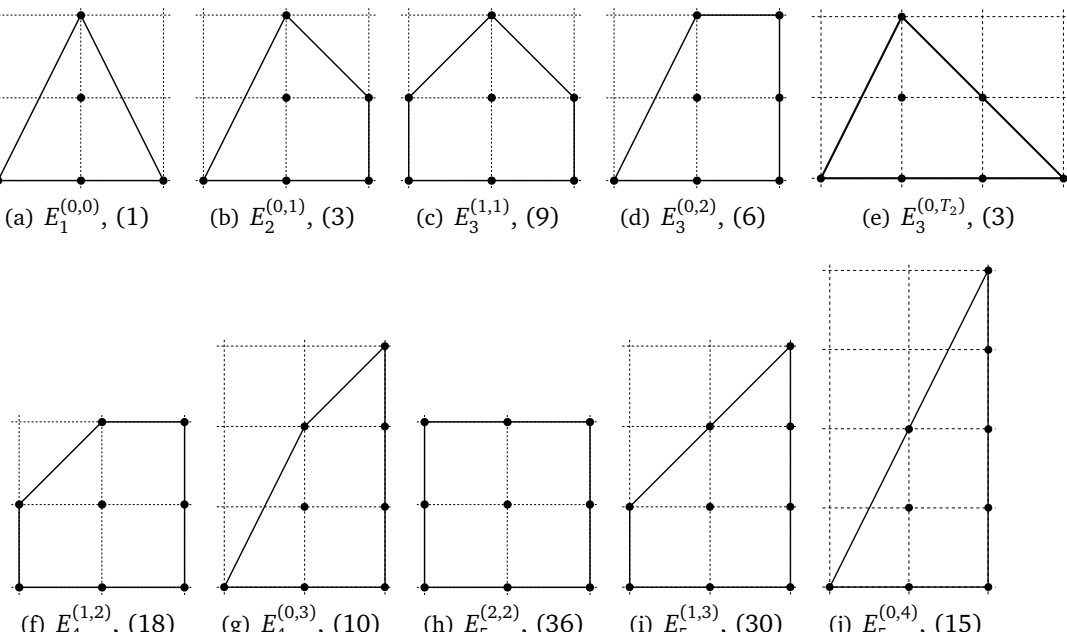

Figure 26: 10 of the 11 non-isolated "rank-one" toric singularities. The number in parenthesis is the number of triangulations with an allowed vertical reduction.

including the internal point $w_1 = (0, 1)$. Given a triangulation of $\Gamma$ with an allowed vertical reduction, we can easily read off the low-energy field theory one obtains in type IIA. We simply have an $SU(2)$ theory with $N_f = n - 1$ flavor, hence the name (7.1) for those non-isolated singularities (in a particular S-duality frame).

Amongst the toric diagrams in Figure 26, the singularity shown in Fig. 26(e) is more peculiar. We claim that it also corresponds to an $SU(2)$ theory with $N_f = 2$, but in a subtler way. As we will explain, one of the two flavors appears non-perturbatively in type IIA. We can think of the toric diagram of Fig. 26(e) as a "gauging" of the so-called $T_2$ toric singularity—a.k.a. the $\mathbb{C}^2/(\mathbb{Z}_2 \times \mathbb{Z}_2)$ orbifold—shown in Figure 27(a). The $T_2$ singularity realizes the 5d $T_2$ theory, the lowest member of the 5d $T_N$ family of 5d SCFTs. Moreover, an appropriate massive deformation of $T_2$ gives rise to free hypermultiplets filling two doublets of $SU(2)$ [29]. We will come back to the $T_N$ family at the end of this section. Since the singularity of Fig. 26(e) realizes an $SU(2)$ gauging of the $T_2$ SCFT "on the right" of the toric diagram, we denote it by $E_3^{(0,T_2)}$.

By including the $T_2$ gauging in our toolbox, we directly find several S-dual descriptions for the geometries in Figure 26, such as shown in Figure 28.

Finally, the last of the 16 "rank-one" toric diagrams is shown in Figure 27(b). It consists of a $T_2$ gauging on the left, while the right-hand-side of the toric diagram corresponds to three additional flavors of $SU(2)$ upon vertical reduction to type IIA. Therefore, this geometry realizes the $SU(2)$ gauge theory with $N_f = 5$ flavors. It is also known as the $T_3$ SCFT.

### 7.1.1 Allowed vertical reductions and missing field-theory chambers

It is interesting to count the triangulations of the toric diagrams $E_n^{(h_L, h_R)}$ (where $h_L, h_R$ could be $T_2$) in Figure 26, or of the S-dual diagrams in Figure 28. We focus on the number of

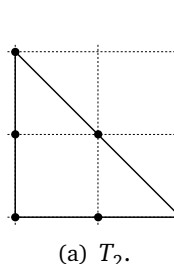

(a) $T_2$.

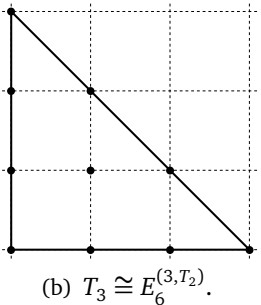

(b) $T_3 \cong E_6^{(3,T_2)}$.

Figure 27: The $T_2$ "rank-zero" toric geometry, and the $T_3$ toric geometry. They have 3 and 30 allowed vertical resolutions, respectively.

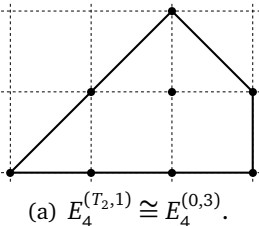

(a) $E_4^{(T_2,1)} \cong E_4^{(0,3)}$.

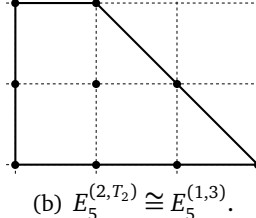

(b) $E_5^{(2,T_2)} \cong E_5^{(1,3)}$.

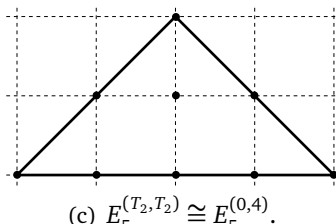

(c) $E_5^{(T_2,T_2)} \cong E_5^{(0,4)}$.

Figure 28: $SL(2,\mathbb{Z})$-transformed diagrams whose vertical reduction involves "$T_2$ gauging." They have 9, 18 and 9 allowed vertical reductions, respectively.

triangulations *with an allowed vertical reduction.* One finds:

$$N_{\text{allowed vertical reductions}} = n_{h_L} n_{h_R} , \quad \text{with:} \quad
\begin{array}{c|cccccc}
h & 0 & 1 & 2 & 3 & 4 & T_2 \\
\hline
n_h & 1 & 3 & 6 & 10 & 15 & 3
\end{array} . \tag{7.3}$$

We displayed these numbers in Figure 26. This should be compared to the number of distinct field-theory chambers of an $SU(2)$ gauge theory with $N_f$ flavors (varying both the Coulomb VEV and the mass parameters). We have:

$$N_{\text{FT chambers for SU(2)},N_f} = 3^{N_f} . \tag{7.4}$$

For the isolated toric singuarities, we found a one-to-one match between the number of allowed vertical reductions and the field-theory chambers. This is not true for the non-isolated singularities, since (7.3) is generally smaller than (7.4) (for $N_f = h_L + h_R$). For instance, the geometry $E_5^{(0,4)}$ of Fig. 26(j) has 15 allowed vertical reductions, which only span 15 out of the 81 field theory chambers of $SU(2)$ with $N_f = 4$.

The missing chambers are related to the presence of non-isolated singularities: one can check in examples that exploring those additional field theory chambers correspond to flopping curves that cannot be flopped in those particular toric geometries. In the IIA setup, that would correspond to D6-branes crossings which are disallowed because some segments of the profiles $\chi_s(r_0)$ would become negative.

Those pathologies are symptoms of the fact that non-isolated singularities, by themselves, do not define five-dimensional SCFTs. In the present case, of course, every known rank-one SCFT can be realized at an isolated singularity, the complex cone over a smooth del Pezzo surface, $dP_{N_f+1}$, which happens to be non-toric for $N_f > 2$. The non-isolated toric singularities considered here are complex cones over singular "pseudo-del Pezzo" toric varieties obtained

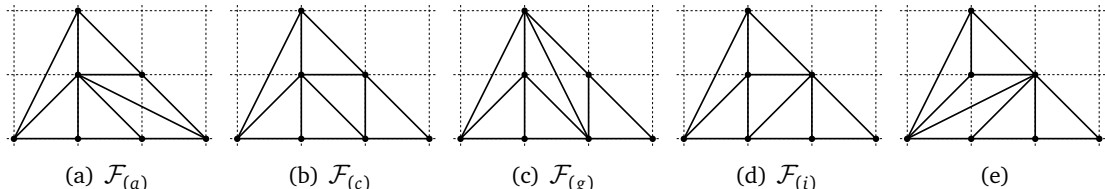

Figure 29: The 5 triangulations of the $E_3^{(0,T_2)}$ toric diagram. For the first 4 of them, we indicated the $SU(2)$ $N_f = 2$ field-theory prepotentials they correspond to, as we explain in the text.

by blowing up $\mathbb{P}^2$ at non-generic points [30]. They are still interesting, in particular because one can easily engineer 5d $\mathcal{N} = 1$ gauge theories by resolving them, but one has to keep the above caveats in mind.

### 7.1.2 The $E_3^{(0,T_2)}$ geometry and the $SU(2)$ $N_f = 2$ gauge theory

We would like to obtain a better understanding of the "$T_2$ gauging" alluded to above. For that purpose, it is sufficient to focus on the $E_3^{(0,T_2)}$ singularity of Figure 26(e). It has five toric resolutions, as shown in Figure 29. The three resolutions of Fig. 29(b), Fig. 29(c) and Fig. 29(d) obviously have an allowed vertical reduction, which gives a quiver of the form:

$$SU(2) \text{---} SU(1) \qquad \cong \qquad SU(2) \text{---} T_2 . \tag{7.5}$$

The naive "$SU(1)$" factor is realized by a single wrapped D6-brane in type IIA. We will show that it can also be understood as two fundamentals of $SU(2)$, one of which arises non-perturbatively. This is denoted by the $T_2$ factor in (7.5).

**Geometric prepotential and gauge theory interpretation.** Consider the toric divisors of $E_3^{(0,T_2)}$ as indicated in Figure 30(a). We may take:

$$S = \mu_3 D_3 + \mu_5 D_5 + \mu_6 D_6 + v\mathbf{E}_0 . \tag{7.6}$$

The geometric prepotential $\mathcal{F} = -\frac{1}{6}S^3$ for the first four resolutions in Figure 29 reads:

$$\begin{aligned}
\mathcal{F}_{(a)} &= -v^3 + \frac{1}{2}(\mu_3 + 3\mu_5 + 2\mu_6)v^2 + \frac{1}{2}(\mu_3^2 - \mu_5^2 - 2\mu_5\mu_6)v , \\
\mathcal{F}_{(c)} &= -\frac{7}{6}v^3 + \frac{1}{2}(3\mu_5 + 2\mu_6)v^2 - \frac{1}{2}(\mu_5^2 + 2\mu_5\mu_6)v , \\
\mathcal{F}_{(g)} &= -\frac{4}{3}v^3 + (2\mu_5 + \mu_6)v^2 - (\mu_5^2 + \mu_5\mu_6)v , \\
\mathcal{F}_{(i)} &= -\frac{4}{3}v^3 + \frac{1}{2}(3\mu_5 + 2\mu_6)v^2 - \frac{1}{2}(\mu_5^2 + 2\mu_5\mu_6)v ,
\end{aligned} \tag{7.7}$$

as one can check by direct computation of the intersection numbers. We claim that these four geometric resolutions corresponds to the field theory chambers denoted by (a), (c), (g) and (i) of the $SU(2)$ $N_f = 2$ gauge theory, as studied in section 4.4. The geometry-to-field-theory map is given by:

$$\mu_3 = m_1 - m_2 , \qquad \mu_5 = -2m_2 , \qquad \mu_6 = h_0 - m_1 , \qquad v = -\varphi - m_1 . \tag{7.8}$$

Indeed, plugging (7.8) into (7.7), one reproduces the corresponding field-theory prepotentials in (4.59)-(4.60), modulo the constant term. Interestingly, the resolution of Fig. 29(a) also has

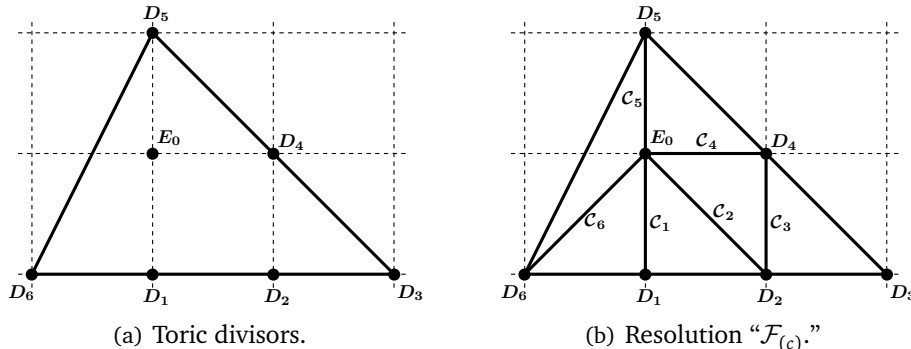

(a) Toric divisors.

(b) Resolution "$\mathcal{F}_{(c)}$."

Figure 30: The $E_3^{(0,T_2)}$ toric geometry, resolution "(c)" and its curves.

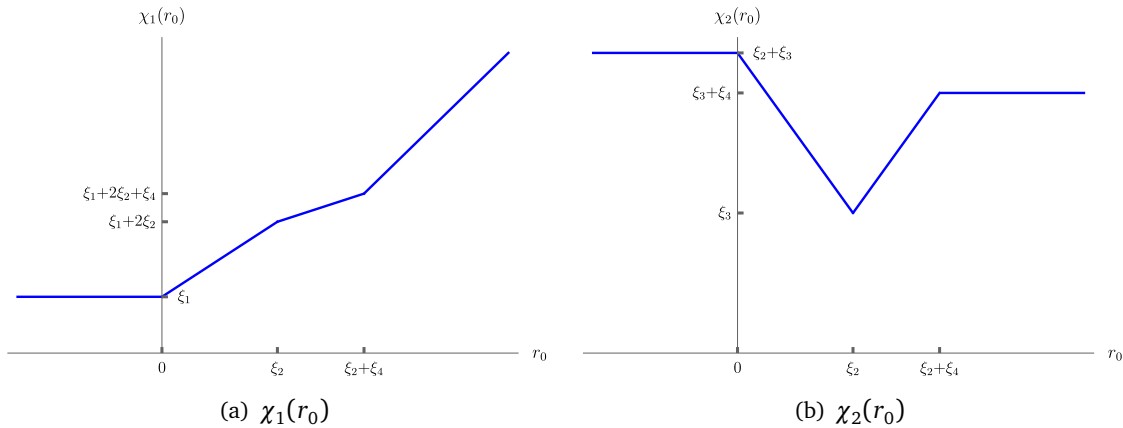

(a) $\chi_1(r_0)$

(b) $\chi_2(r_0)$

Figure 31: IIA profile for the resolution "(c)" of the $E_3^{(0,T_2)}$ geometry.

a gauge-theory interpretation, even though it doesn't have an allowed IIA reduction, strictly speaking. To understand this point better, let us study the resolution of Fig. 29(b) and its IIA reduction in more detail.

**Vertical resolution and field theory chamber (c).**  Consider the resolution of Figure 29(b), with its curves labelled as in Figure 30(b). We have the relations:

$$\mathcal{C}_5 \cong \mathcal{C}_1 + 2\mathcal{C}_2 + \mathcal{C}_4 , \qquad \mathcal{C}_6 \cong \mathcal{C}_2 + \mathcal{C}_4 . \tag{7.9}$$

The GLSM reads:

|  | $D_1$ | $D_2$ | $D_3$ | $D_4$ | $D_5$ | $D_6$ | $\mathbf{E}_0$ |  |
|---|---|---|---|---|---|---|---|---|
| $\mathcal{C}_1$ | $-2$ | $1$ | $0$ | $0$ | $0$ | $1$ | $0$ | $\xi_1$ |
| $\mathcal{C}_2$ | $1$ | $-1$ | $0$ | $1$ | $0$ | $0$ | $-1$ | $\xi_2$ |
| $\mathcal{C}_3$ | $0$ | $-1$ | $1$ | $-1$ | $0$ | $0$ | $1$ | $\xi_3$ |
| $\mathcal{C}_4$ | $0$ | $1$ | $0$ | $-1$ | $1$ | $0$ | $-1$ | $\xi_4$ |
| $U(1)_M$ | $1$ | $0$ | $0$ | $0$ | $0$ | $0$ | $-1$ | $r_0$ |

$$\tag{7.10}$$

The relation between the $\mu, \nu$ parameters in (7.6) and the FI parameters is:

$$\mu_3 = \xi_2 + \xi_3 , \qquad \mu_5 = -\xi_2 + \xi_4 , \qquad \mu_6 = \xi_1 , \qquad \nu = -\xi_2 . \tag{7.11}$$

A vertical reduction of this geometry gives an $A_2$ singularity in type IIA, with two D6-branes wrapped on $\mathbb{P}^1_1$, and a single D6-brane wrapped over $\mathbb{P}^1_2$. The IIA profiles for the two exceptional curves are:

$$
\chi_1(r_0) = \begin{cases} 3r_0 + \xi_1 - \xi_2 - 2\xi_4 & \text{if } r_0 \geq \xi_2 + \xi_4, \\ r_0 + \xi_1 + \xi_2 & \text{if } \xi_2 \leq r_0 \leq \xi_2 + \xi_4, \\ 2r_0 + \xi_1 & \text{if } 0 \leq r_0 \leq \xi_2, \\ \xi_1 & \text{if } r_0 \leq 0, \end{cases}
\tag{7.12}
$$

$$
\chi_2(r_0) = \begin{cases} \xi_3 + \xi_4 & \text{if } r_0 \geq \xi_2 + \xi_4, \\ r_0 - \xi_2 + \xi_3 & \text{if } \xi_2 \leq r_0 \leq \xi_2 + \xi_4, \\ -r_0 + \xi_2 + \xi_3 & \text{if } 0 \leq r_0 \leq \xi_2, \\ \xi_2 + \xi_3 & \text{if } r_0 \leq 0. \end{cases}
\tag{7.13}
$$

This is displayed in Figure 31. The profile for $\chi_1(r_0)$ shows the two gauge D6-branes wrapped over $\mathbb{P}^1_1$, realizing the $SU(2)$ gauge group, and one "flavor" brane between them, corresponding to the single D6-brane wrapped over $\mathbb{P}^1_2$. We then naively find a single flavor of $SU(2)$ realized by open strings, with

$$
M(\mathcal{H}_{1,1}) = \varphi + m_1 = \xi_2, \quad M(\mathcal{H}_{2,1}) = \varphi - m_1 = \xi_4, \quad M(W_\alpha) = 2\varphi = \xi_2 + \xi_4.
\tag{7.14}
$$

Indeed, in the limit $\xi_3 \to \infty$, one can decouple the divisor $D_3$ and the toric diagram in Fig. 30(b) goes over to the toric diagram of $E_2^{(0,1)}$ in Fig. 26(b), corresponding to $SU(2)$ with $N_f = 1$. Thus, we expect that, for finite values of $\xi_3$, M2-branes wrapped over $\mathcal{C}_3$ realize one fundamental hypermultiplet of the gauged $SU(2)$.

To understand this better, we look at the instanton particles. From the IIA setup, the naive unitary quiver is:

$$
U(2)_{-\frac{3}{2}} \text{——} U(1)_0,
\tag{7.15}
$$

where the subscripts are the effective CS levels, which are read off from the IIA profile. In our conventions, that means that the bare CS levels for the $U(2)$ and $U(1)$ gauge groups are $k_{U(2)} = -1$ and $k_{U(1)} = 1$, respectively. The corresponding prepotential (2.30) reads:

$$
\begin{aligned}
\mathcal{F}^{U(2)-U(1)} &= \sum_{i=1}^{2} \left( \frac{1}{2} h_0 \phi_i^2 - \frac{1}{6} \phi_i^3 - \frac{1}{6} \Theta(\phi_i + \phi_0 + m_1)(\phi_i + \phi_0 + m_1)^3 \right) \\
&+ \frac{1}{6}(\phi_1 - \phi_2)^3 + \frac{1}{2} \widetilde{h}_0 \phi_0^2 + \frac{1}{6} \phi_0^3.
\end{aligned}
\tag{7.16}
$$

Here, $\widetilde{h}_0$ is the $U(1)$ gauge coupling, $\phi_0$ is the $U(1)$ Coulomb branch parameter, and $m_1$ is the mass of the bifundamental hypermultiplet coupling the $U(2)$ and $U(1)$ gauge groups. Using (2.32), we find:

$$
M(\mathcal{I}_{1,SU(2)}) = h_0 - m_1 = \xi_1, \qquad M(\mathcal{I}_{1,SU(2)}) = h_0 + 3\varphi = \xi_1 + 2\xi_2 + \xi_4,
\tag{7.17}
$$

for the $SU(2)$ instanton masses, where the identification with the FI terms follows from the IIA profile in Fig. 31(a). We also have a more mysterious "$SU(1)$ instanton," with mass:

$$
M(\mathcal{I}_{SU(1)}) = \widetilde{h}_0 - m_1 - \varphi = \xi_3.
\tag{7.18}
$$

The identification with $\xi_3$ is clear from Fig. 31(b). The fact that there can be non-trivial contributions to the low-energy physics from "stringy instantons" at a quiver node with trivial gauge group—here, $SU(1)$—, due to wrapped D-branes, is familiar in string theory (see *e.g.* [94]).

We claim that this particular "stringy instanton" particle is equivalent to an hypermultiplet in the fundamental of $SU(2)$. Indeed, if we identify the "$SU(1)$ gauge coupling" with the mass difference:

$$\widetilde{h}_0 = m_1 - m_2 \,, \tag{7.19}$$

we obtain the relation:

$$\xi_3 = -\varphi - m_2 \,, \tag{7.20}$$

which is the positive mass of the second hypermultiplet mode in the correct field theory chamber. We can then identify:

$$M(\mathcal{I}_{SU(1)}) = M(\mathcal{H}_{2,1}) = \xi_3 = -\varphi - m_2 \,, \quad M(\mathcal{H}_{2,2}) = \xi_2 + \xi_3 + \xi_4 = \varphi - m_2 \,, \tag{7.21}$$

for the two modes of this "non-perturbative" hypermultiplet; it corresponds to M2-branes wrapped over $\mathcal{C}_3$ and $\mathcal{C}_3 + \mathcal{C}_6$, respectively. Some additional discussion of this "$SU(1)$" mode can be found in Appendix D.

In summary, we have the following relation between the FI terms in the GLSM (7.10) and the parameters of an $SU(2)$ $N_f = 2$ gauge theory:

$$\xi_1 = h_0 - m_1 \,, \qquad \xi_2 = \varphi + m_1 \,, \qquad \xi_4 = \varphi - m_1 \,, \qquad \xi_3 = -\varphi - m_2 \,. \tag{7.22}$$

Plugging this into (7.11), we obtain the relations (7.8), as anticipated.

**Flowing to the $T_2$ theory.** We can obtain the $T_2$ geometry of Fig. 27(a) as a decoupling limit from the geometry in Fig. 30(b), simply by taking the limit $\xi_1 \to \infty$ (and thus $\xi_5 \to \infty$). This has the effect of decoupling the toric divisor $D_6$. It corresponds to the limit of vanishing gauge coupling, $h_0 \to \infty$, so that the gauged $SU(2)$ becomes a global symmetry.

**Other resolutions of the $E_3^{(0,T_2)}$ singularity.** The other two resolutions that admit a vertical reduction, Figures 29(c) and 29(d), correspond to the field-theory chambers (g) and (i), respectively, of the $SU(2)$ $N_f = 2$ gauge theory in (4.61) (see the Figures 11(g) and 11(i) for the corresponding $E_3$ resolutions). The field theory chambers (c), (g), (i) are the only three chambers such that $\pm\varphi_2 + m_2 < 0$, as we can see from (4.61). Correspondingly, the second hypermultiplet does not contribute at all to the prepotential (due to the step function). For this reason, these are the only 3 field theory chambers (out of 9) in which the $SU(2)$ $N_f = 2$ prepotential is compatible with the prepotential (7.16), which corresponds to the unitary quiver seen by open strings in the type-IIA picture.

The vertical reduction can be performed for the field theory chambers (g) and (i) exactly as for chamber (c), and one confirms the map (7.22) between geometry and field theory. What is perhaps more surprising is that the resolution of Figure 29(a), which has no allowed vertical reduction, is nonetheless described by the field-theory chamber (a). This is necessary for consistency of the whole picture: one can go from "chamber (c)" to "chamber (a)" by flopping the curve $\mathcal{C}_3$ in Fig. 30(b); due to the indentification (7.21), this corresponds to flipping the sign of the mass of the hypermultiplet $\mathcal{H}_{2,1}$.

We note also that the toric singularity $E_3^{(0,T_2)}$ has only five Kähler chambers, compared to the 24 Kähler chambers for the $E_3$ singularity that we studied in section 4.4. Moreover, in the 3 chambers where both singularities have the same gauge-theory description, the spectrum of instanton operators is nonetheless different between the two geometries. We again interpret these observations as an indication that non-isolated singularities do not give rise to well-defined 5d SCFTs.

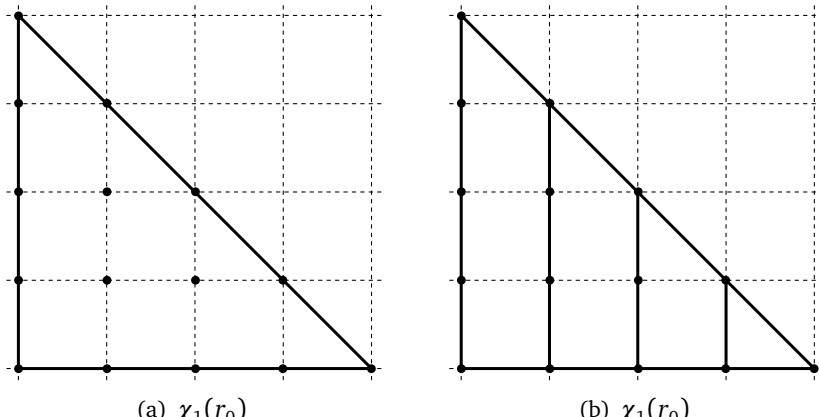

(a) $\chi_1(r_0)$          (b) $\chi_1(r_0)$

Figure 32: The $T_N$ toric diagram (a.k.a. $\mathbb{C}^3/(\mathbb{Z}_N \times \mathbb{Z}_N)$) and a partial resolution with an allowed vertical reduction, for $N = 4$.

## 7.2 General $T_N$ quiver

The four-dimensional 4d $\mathcal{N} = 2$ supersymmetric $T_N$ theory is a 4d SCFT with (at least) $SU(N)^3$ global symmetry, corresponding to $N$ M5-branes on a sphere with three full punctures [95]. Its five-dimensional uplift was proposed in [28], as the 5D SCFT arising at the intersection of $N$ D5-branes, $N$ NS5-branes and $N$ $(1, 1)$-fivebranes in type IIB. While all the 5-branes should end on 7-branes, it has been argued that one can send the 7-branes to infinity without changing the low-energy SCFT—possibly up to some "decoupled factors." Then, one has a simple $(p, q)$-web, dual to a $\mathbb{C}^3/(\mathbb{Z}_N \times \mathbb{Z}_N)$ toric orbifold singularity, as shown in Figure 32(a).

For any given partial resolution of the singularity with an allowed vertical reduction, as shown in Figure 32(b), one can directly read off the quiver description (3.48) from type IIA:

$$[U(N)] \longrightarrow SU(N-1) \longrightarrow \quad \cdots \quad \longrightarrow SU(3) \longrightarrow SU(2) \longrightarrow T_2 . \qquad (7.23)$$

Here, all the effective CS levels vanish, as one can check by direct computation. The "$T_2$ tail" corresponds to two flavors of $SU(2)$, as described above. In this way, our IIA perspective nicely reproduces the known quiver description of the mass-deformed $T_N$ theory [11, 29].

In the rank-one case, we know that the $T_3$ theory is better described by the isolated singularity $\mathbf{X} = E_6$, with resolution $\widehat{\mathbf{X}} = \text{Tot}(\mathcal{K} \to dP_6)$, which is non-toric. It is tempting to conjecture that, for any $N$, there exists some (non-toric) isolated singularity $\mathbf{X}$ whose crepant resolutions capture the full deformation space of the five-dimensional $\mathcal{T}_{\mathbf{X}} = T_N$ SCFT. We hope to come back to this question in future work.

## Acknowledgements

We would like to thank Fabio Apruzzi, Benjamin Assel, Francesco Benini, Sergio Benvenuti, Daniel Butter, Thomas Dumitrescu, Marco Fazzi, Amihay Hanany, Kenneth Intriligator, Heeyeon Kim, Martin Roček, Antonio Sciarappa and Brian Willett for interesting discussions and correspondence. CC is a Royal Society University Research Fellow and a Research Fellow at St John's College, Oxford. VS is supported in part by NSF grant PHY-1620628.

# A  The 5d $\mathcal{N} = 1$ gauge-theory prepotential, revisited

In this Appendix, we revisit the derivation of the well-known one-loop prepotential $\mathcal{F}(\varphi)$, which determines the low-energy theory on the Coulomb branch of a 5d $\mathcal{N} = 1$ gauge theory [1,4,65]. The prepotential can be derived in a slightly roundabout way, by compactifying the theory on a circle, thus obtaining a 4d $\mathcal{N} = 2$ KK theory, and then taking the 5d limit on the 4d $\mathcal{N} = 2$ prepotential [65].

Let us denote by $a$ the 4d Coulomb branch complex scalar. For a four-dimensional gauge theory, the one-loop 4d $\mathcal{N} = 2$ prepotential is given by:

$$
\begin{aligned}
\mathcal{F}_{\text{4d}}(a) \;\; &= \frac{\tau}{2} \operatorname{Tr}(a^2) - \frac{1}{8\pi i} \sum_{\alpha \in \Delta} \alpha(a)^2 \left[ \log\left( \frac{\alpha(a)^2}{\Lambda^2} \right) - 3 \right] \\
&\quad + \frac{1}{8\pi i} \sum_{\rho,\omega} (\rho(a) + \omega(\mu))^2 \left[ \log\left( \frac{(\rho(a) + \omega(\mu))^2}{\Lambda^2} \right) - 3 \right] + \cdots .
\end{aligned}
\tag{A.1}
$$

The ellipsis denotes the instanton corrections, which can be safely ignored because they will be suppressed in the 5d limit [65]. The prepotential of the 5d $\mathcal{N} = 1$ theory on a circle is obtained by resumming the contributions of the KK modes. In particular, an hypermultiplet of charge 1 under some $U(1)$ will contribute:

$$
\mathcal{F}_{\text{4d}}^{\mathcal{H}}(a) = \frac{1}{8\pi i} \sum_{n \in \mathbb{Z}} (a+n)^2 \left[ \log(a+n)^2 - 3 \right] ,
\tag{A.2}
$$

formally. Here, we choose $a$ a dimensionless parameter, soaking up dimensions with the $S^1$ radius $\beta$:

$$
a = a_0 - i\beta\varphi , \qquad a_0 = \frac{1}{2\pi} \int_{S^1} A ,
\tag{A.3}
$$

with $\varphi$ and $A_\mu$ the 5d scalar and gauge field, respectively. The diverging sum (A.2) can be regularized to: [32]

$$
\mathcal{F}_{\text{4d}}^{\mathcal{H}}(a) = -\frac{1}{(2\pi i)^3} \operatorname{Li}_3(e^{2\pi i a}) .
\tag{A.4}
$$

This, we claim, is the correct result for a single hypermultiplet in the $U(1)_{-\frac{1}{2}}$ quantization, as discussed in section 2.2. Indeed, the limit $\varphi \to -\infty$ gives us:

$$
\lim_{\operatorname{Im}(a) \to -\infty} \mathcal{F}_{\text{4d}}^{\mathcal{H}}(a) = 0 , \qquad \lim_{\operatorname{Im}(a) \to \infty} \mathcal{F}_{\text{4d}}^{\mathcal{H}}(a) = \frac{1}{6} a^3 + \frac{1}{4} a^2 + \frac{1}{2} a ,
\tag{A.5}
$$

in agreement with (2.15). The cubic polynomial on the right-hand-side corresponds to the 5d CS action at level $k = -1$.

**5d limit.**  The 5d prepotential can be obtained from the 4d prepotential of the theory on a circle in the large-$\beta$ limit, with:

$$
\mathcal{F} = \lim_{\beta \to \infty} \frac{i}{\beta^3} \mathcal{F}_{\text{4d}} .
\tag{A.6}
$$

---

[32]One quick way to obtain this result is to note that, formally, the fourth derivative of (A.2) is a convergent sum, which gives:

$$
\partial_a^4 \mathcal{F}_{\text{4d}}^{\mathcal{H}}(a) = \frac{\pi i}{2 \sin^2(\pi a)} .
$$

This determines $\mathcal{F}_{\text{4d}}^{\mathcal{H}}(a)$ up to a cubic polynomial in $a$, which we fix by requiring consistency with the decoupling limits $\varphi \to \pm\infty$.

The 4d gauge coupling is related to the 5d gauge coupling $g^2$ by:

$$\tau = \frac{\theta}{2\pi} + \frac{4\pi i}{g_{4d}^2} = \frac{\theta}{2\pi} + i\beta h_0 , \qquad h_0 \equiv \frac{8\pi^2}{g^2} . \tag{A.7}$$

We then obtain:

$$\mathcal{F}^{\text{classical}} = \frac{1}{2} h_0 \varphi^2 + \frac{k}{6} \varphi^3 \tag{A.8}$$

for the YM and CS terms, for $\mathbf{G} = U(1)$; the generalization to any $\mathbf{G}$ is straightforward. The one-loop contribution (A.4) from a single hypermultiplet gives us:

$$\mathcal{F}^{\mathcal{H}} = \begin{cases} 0 & \text{if } \varphi < 0 \\ -\frac{1}{6}\varphi^3 & \text{if } \varphi > 0 \end{cases} = -\frac{1}{6}\Theta(\varphi)\varphi^3 . \tag{A.9}$$

The analysis of the W-boson contribution is completely similar. We thus obtain the five-dimensional one-loop prepotential given in (2.19), namely:

$$\begin{aligned} \mathcal{F}(\varphi, \mu) =\ & \frac{1}{2} h_0 \operatorname{Tr}(\varphi^2) + \frac{k^{abc}}{6} \varphi_a \varphi_b \varphi_c + \frac{1}{6} \sum_{\alpha \in \Delta} \Theta(\alpha(\varphi))(\alpha(\varphi))^3 \\ & - \frac{1}{6} \sum_{\omega} \sum_{\rho \in \mathfrak{R}} \Theta(\rho(\varphi) + \omega(m))(\rho(\varphi) + \omega(m))^3 . \end{aligned} \tag{A.10}$$

**Comparing with IMS.** At first sight, the result (A.10) differs from the well-known IMS prepotential [4], which reads:

$$\mathcal{F}_{\text{IMS}}(\varphi, \mu) = \frac{1}{2} h_0 \operatorname{Tr}(\varphi^2) + \frac{k_{\text{eff}}^{abc}}{6} \varphi_a \varphi_b \varphi_c + \frac{1}{12} \sum_{\alpha \in \Delta} |\alpha(\sigma)|^3 - \frac{1}{12} \sum_{\rho, \omega} |\rho(\sigma) + \omega(m)|^3 . \tag{A.11}$$

Since the roots come in pairs, $\alpha$ and $-\alpha$, it is easy to see that the W-boson contribution in (A.10) is the same as in (A.11), namely:

$$\mathcal{F}^{\text{vec}}(\varphi) = \frac{1}{6} \sum_{\alpha \in \Delta} \Theta(\alpha(\varphi))(\alpha(\varphi))^3 = \frac{1}{12} \sum_{\alpha \in \Delta} |\alpha(\varphi)|^3 . \tag{A.12}$$

On the other hand, our choice of a gauge-invariant quantization for the hypermultiplets leads to a small discrepancy between the IMS prepotential (A.11) and (A.10).

Let us explain this point in more detail. At some level, it is only a matter of notation. For instance, consider a $U(1)$ theory with bare CS level $k \in \mathbb{Z}$ and one hypermultiplet of charge $Q$. We have:

$$\mathcal{F} = \frac{1}{2} h_0 \varphi^2 + \frac{k}{6} \varphi^3 - \frac{1}{6} \Theta(Q\varphi)(Q\varphi)^3 . \tag{A.13}$$

This is equal to the IMS prepotential:

$$\mathcal{F}_{\text{IMS}} = \frac{1}{2} h_0 \varphi^2 + \frac{k_{\text{eff}}}{6} \varphi^3 - \frac{1}{12} |Q\varphi|^3 , \tag{A.14}$$

with

$$k_{\text{eff}} = k - \frac{1}{2} Q^3 . \tag{A.15}$$

This theory would generally be called "$U(1)_{k_{\text{eff}}}$ coupled to an hypermultiplet;" we are being slightly pedantic in distinguishing between the integer-quantized bare CS level $k$ and the contribution $-\frac{1}{2}Q^3$ from the hypermultiplet itself. More generally, the cubic terms in $\varphi$ agree between (A.11) and (A.10) if we identify:

$$k_{\text{eff}}^{abc} = k^{abc} - \frac{1}{2} \sum_{\rho, \omega} \rho^a \rho^b \rho^c . \tag{A.16}$$

On the other hand, the lower-order terms (of order $\varphi^2$ and $\varphi$) are, in general, slightly different between the two expressions. In our conventions, all the IR contact terms $\kappa$, for both gauge and flavor symmetries (or mixed gauge-flavor), are integer-quantized at a generic point on the Coulomb branch.

# B  Intersection numbers of toric CY threefolds

In this appendix, we discuss the computation of triple-interesection numbers in smooth local Calabi-Yau threefolds. We consider many examples, collecting results that are useful in the main text.

## B.1  Computing the M-theory prepotential

Consider a toric $CY_3$ geometry $\widehat{\mathbf{X}}$, which we present as a GLSM (3.28). It will be convenient to write it as:

$$
\begin{array}{c|cc|c}
 & D_l & \mathbf{E}_a & \text{FI} \\
\hline
\mathcal{C}^{\mathbf{a}} & Q_l^{\mathbf{a}} & Q_a^{\mathbf{a}} & \xi^{\mathbf{a}}
\end{array} \, ,
\tag{B.1}
$$

where we distinguish between the non-compact toric divisors $D_i$, $l = 1, \cdots, n_E$, and the compact ones, $\mathbf{E}_a$, $a = 1, \cdots, r$. For definiteness, we can choose the rows in (B.1) to correspond to curves in $\widehat{\mathbf{X}}$. The Kähler class of the threefold is parameterized by:

$$
S = \mu^i D_i + \nu^a \mathbf{E}_a \, ,
\tag{B.2}
$$

where the $D_i$'s appearing here are a chosen subset of $n_E - 3$ elements amongst the $n_E$ non-compact toric divisors:

$$
\{D_i\}_{i=1}^{n_E - 3} \subset \{D_l\}_{l=1}^{n_E} \, .
\tag{B.3}
$$

We choose the $D_i$'s such that $(D_i, \mathbf{E}_a)$ form a dual basis to the curves $\mathcal{C}^{\mathbf{a}}$. In other words, we define the following square matrix of intersection numbers:

$$
(\mathbf{Q}^{\mathbf{a}}{}_{\mathbf{b}}) = \left( Q_i^{\mathbf{a}}, Q_b^{\mathbf{a}} \right) = \left( \mathcal{C}^{\mathbf{a}} \cdot D_i, \mathcal{C}^{\mathbf{a}} \cdot \mathbf{E}_b \right) \, , \qquad \mathbf{b} = (i, b) \, ,
\tag{B.4}
$$

and we choose the $D_i$'s such that $\det(\mathbf{Q}) \neq 0$. [33] Then, the parameters $\mu$, $\nu$ in (B.2) are related to the Kähler parameters $\xi^{\mathbf{a}}$ (the FI parameters) as:

$$
\xi^{\mathbf{a}} = Q_i^{\mathbf{a}} \mu^i + Q_b^{\mathbf{a}} \nu^b \qquad \Longleftrightarrow \qquad \begin{pmatrix} \mu \\ \nu \end{pmatrix}^{\mathbf{a}} = (\mathbf{Q}^{-1})^{\mathbf{a}}{}_{\mathbf{b}} \, \xi^{\mathbf{b}} \, .
\tag{B.5}
$$

**The prepotential from M-theory.**  The geometric prepotential of $\widehat{\mathbf{X}}$ is given by:

$$
\mathcal{F} = -\frac{1}{6} S \cdot S \cdot S \, .
\tag{B.6}
$$

It is convenient to expand it as:

$$
\mathcal{F} = -\frac{1}{6} \mathbf{E}_a \mathbf{E}_b \mathbf{E}_c \, \nu^a \nu^b \nu^c - \frac{1}{2} \mathbf{E}_a \mathbf{E}_b D_i \, \nu^a \nu^b \mu^i - \frac{1}{2} \mathbf{E}_a D_i D_j \, \nu^a \mu^i \mu^j - \frac{1}{6} D_i D_j D_k \mu^i \mu^j \mu^k \, .
\tag{B.7}
$$

For most purposes, we can discard the last term:

$$
\mathcal{F}_0 \equiv -\frac{1}{6} D_i D_j D_k \mu^i \mu^j \mu^k \, ,
\tag{B.8}
$$

---

[33] Whenever possible, we choose a basis such that $|\det(\mathbf{Q})| = 1$.

which is ill-defined on a non-compact Calabi-Yau. Whenever there is a gauge-theory interpretation, $\mathcal{F}_0$ corresponds to a $\varphi$-independent term—namely, a constant term that does not affect any flat-space observable of the five-dimensional theory. Nonetheless, it is sometimes possible to give a useful prescription for (B.8), as we shall discuss momentarily. Let us also note that $\mathcal{F}_0$ actually introduces local terms for background vector multiplet for the flavor symmetry, and therefore some ambiguity is indeed expected.

Neglecting $\mathcal{F}_0$ for now, we must simply compute the intersection numbers that involve at least one compact divisor:

$$\mathbf{E}_a \cdot \mathbf{E}_b \cdot \mathbf{E}_c \,, \qquad \mathbf{E}_a \cdot \mathbf{E}_b \cdot D_i \,, \qquad \mathbf{E}_a \cdot D_i \cdot D_j \,. \tag{B.9}$$

This can be done systematically using toric methods, as we reviewed in section 3.2.2.

### B.2 Triple intersection numbers and JK residue

One can also compute the intersection numbers using a residue formula, which allows us to consistently define the "non-compact" intersection numbers in (B.8), by introducing some equivariant parameters as regulators [96–98].

This computation is based on the topological A-model. Namely, even though we are considering the threefold $\widehat{\mathbf{X}}$ in M-theory, it is still useful to view the GLSM description of the toric geometry as 2d $\mathcal{N} = (2,2)$ gauge theory, and to consider the corresponding topological A-twist. Then, the intersection numbers are simply the zero-instanton contribution to the genus-zero correlators. A useful localization formula for the latter can be given [96, 97] in terms of the so-called Jeffrey-Kirwan (JK) residue [99].

**The residue formula.** Let us introduce the dummy variables $\sigma = (\sigma_{\mathbf{a}})$, and let us assign the following (formal) linear functions of $\sigma$ to the toric divisors:

$$D_i \equiv Q_i(\sigma) \equiv Q_i^{\mathbf{a}}\sigma_{\mathbf{a}} \,, \qquad \mathbf{E}_a \equiv Q_a(\sigma) = Q_a^{\mathbf{a}}\sigma_{\mathbf{a}} \,. \tag{B.10}$$

We define the "correlator" of any polynomial $P$ of the $\sigma$'s as:

$$\langle P(\sigma)\rangle_0 \equiv \oint_{\mathrm{JK}(\eta=\xi)} \frac{d\sigma_1}{2\pi i} \wedge \cdots \wedge \frac{d\sigma_{n-3}}{2\pi i} \frac{P(\sigma)}{\prod_{l=1}^{n_E}(Q_l(\sigma)+\lambda_l)\prod_{a=1}^{r}Q_a(\sigma)} \,. \tag{B.11}$$

Here, note that the denominator is a product over *all* the $n = n_E + r$ toric divisors from (B.1). The "equivariant parameters" $\lambda_l \in \mathbb{C}$ are regulators, which we choose to be generic. The JK residue is a simple operation on the integrand. For $\lambda_l$ generic, it is essentially a sum of iterated residues at so-called "regular singularities," where $n-3$ hyperplanes $\{Q_l(\sigma)+\lambda_l = 0\} \subset \mathbb{C}^{n-3}$ or $\{Q_a(\sigma) = 0\} \subset \mathbb{C}^{n-3}$ intersect at a point. Importantly, the singularities that contributes depend on the auxiliary vector:

$$\eta^{\mathbf{a}} = \xi^{\mathbf{a}} \,, \tag{B.12}$$

which coincides with the FI parameters of the GLSM. In this way, the JK residue gives a different answer for $\widehat{\mathbf{X}}$ in different Kähler chambers. We refer to [97] (and references therein) for a complete definition of the JK residue.

The triple intersection numbers (B.9) can be computed as "correlators" of the form:

$$\langle \mathbf{E}_a\mathbf{E}_b\mathbf{E}_c\rangle_0 \,, \qquad \langle \mathbf{E}_a\mathbf{E}_bD_i\rangle_0 \,, \qquad \langle \mathbf{E}_aD_iD_j\rangle_0 \,, \tag{B.13}$$

respectively. These numbers are independent of the regulators $\lambda_l$ (assuming $\lambda_l$ is generic), and only depend on the FI parameters $\xi$ through the Kähler chamber in which $\xi$ sits. (We assume that $\xi$ is not on a Kähler wall, so that the JK residue is well-defined.)

The advantage of the residue formula is that it allows us to *define* the triple intersection numbers of three non-compact divisors, as:

$$D_i \cdot D_j \cdot D_k \equiv \langle D_i D_j D_k \rangle_0 . \tag{B.14}$$

This approach was recently discussed in [98]. The result (B.14) does depend on the regulator $\lambda_l$ in a non-trivial way. However, in simple-enough cases at least, one can always take some convenient limit on the $\lambda_l$'s to simplify the final answer. We will see some examples in the next subsection, where we can find an answer for (B.14) that allows us to reproduce the constant term of the gauge-theory prepotential.

### B.3 Intersection numbers in examples

Let us now consider various toric geometries studied in this paper, and compute the intersection numbers in each case..

### B.3.1 Resolution of the $E_1$ singularity (local $\mathbb{F}_0$)

Consider the $E_1$ singularity, studied in section 4.1, with the GLSM (4.4). The toric divisors are shown in Figure 7(a). The intersection numbers involving the compact divisor $\mathbf{E}_0$ are easily computed:

$$\mathbf{E}_0^3 = 8 , \quad D_1 \mathbf{E}_0^2 = D_2 \mathbf{E}_0^2 = -2 , \quad D_1^2 \mathbf{E}_0 = D_2^2 \mathbf{E}_0 = 0 , \quad D_1 D_2 \mathbf{E}_0 = 1 . \tag{B.15}$$

One can also check this using the JK residue. In this case, we have a unique Kähler chamber, corresponding to $\eta = (1,1)$, say, in the JK residue (B.11). We thus have:

$$\langle P(\sigma) \rangle_0 = \oint_{\substack{JK \\ \eta=(1,1)}} \frac{d\sigma_1 d\sigma_2}{(2\pi i)^2} \frac{P(\sigma)}{(\sigma_1 + \lambda_1)(\sigma_2 + \lambda_2)(\sigma_1 + \lambda_3)(\sigma_2 + \lambda_4)(-2\sigma_1 + -2\sigma_2)} .$$

This reproduces the intersection numbers (B.15)—for instance:

$$\mathbf{E}_0^3 = \langle (-2\sigma_1 - 2\sigma_2)^3 \rangle_0 = 8 , \qquad D_1 \mathbf{E}_0^2 = \langle \sigma_1 (-2\sigma_1 - 2\sigma_2)^2 \rangle_0 = -2 , \quad \text{etc.} \tag{B.16}$$

We can also use the residue formula to define the "non-compact" intersection numbers, as explained above. Let us choose

$$D_i D_j D_k = \lim_{(\lambda_1, \lambda_2, \lambda_3, \lambda_4) \to (-3,1,-3,1)\lambda} \langle D_i(\sigma) D_j(\sigma) D_j(\sigma) \rangle_0 . \tag{B.17}$$

This gives

$$D_1^3 = 0 , \quad D_1^2 D_2 = 0 , \quad D_1 D_2^2 = -\frac{1}{2} , \quad D_2^3 = \frac{1}{2} . \tag{B.18}$$

This is the result we quoted in (4.9); of course, upon using the linear relation $\mathbf{E}_0 \cong -2D_1 - 2D_2$, these interesection numbers also reproduce (B.15). We chose an *ad-hoc* limit (B.17) on the $\lambda$ regulator in order to reproduce the field-theory result for pure $SU(2)$ upon vertical reduction.

### B.3.2 Resolutions of the $\widetilde{E}_1$ singularity (local $dP_1$)

Consider the $\widetilde{E}_1$ singularity, studied in section 4.2, with the GLSM (4.17). It has two resolutions, shown in Figure 8.

**Resolution (a).** Consider the resolution shown in Figure 8(a). One can easily compute the intersection numbers:

$$\mathbf{E}_0^3 = 8 , \qquad D_1\mathbf{E}_0^2 = -2 , \qquad D_2\mathbf{E}_0^2 = -3 ,$$
$$D_1^2\mathbf{E}_0 = 0 , \qquad D_2^2\mathbf{E}_0 = 1 , \qquad D_1D_2\mathbf{E}_0 = 1 . \tag{B.19}$$

Using the residue formula, one can also define the "non-compact" intersection numbers. We again choose a convenient limit:

$$D_iD_jD_k = \lim_{(\lambda_1,\lambda_2,\lambda_3,\lambda_4)\to(0,1,0,1)\lambda} \langle D_i(\sigma)D_j(\sigma)D_j(\sigma)\rangle_0 . \tag{B.20}$$

Here, the JK residue is taken with $\eta = (1,1)$, since the Kähler chamber is such that $\xi_2 > 0$, $\xi_3 > 0$. This gives:

$$D_1^3 = 0 , \quad D_1^2D_2 = 0 , \quad D_1D_2^2 = -\frac{1}{2} , \quad D_2^3 = -\frac{1}{4} . \tag{B.21}$$

**Resolution (b).** For completeness, let us give the intersection numbers in the second resolution, shown in Figure 8(c). One finds:

$$\mathbf{E}_0^3 = 9 , \qquad D_1\mathbf{E}_0^2 = -3 , \qquad D_2\mathbf{E}_0^2 = -3 ,$$
$$D_1^2\mathbf{E}_0 = 1 , \qquad D_2^2\mathbf{E}_0 = 1 , \qquad D_1D_2\mathbf{E}_0 = 1 . \tag{B.22}$$

Using the same limit (B.20) with $\eta = (2,-1)$, since the Kähler chamber is such that $\xi_2+\xi_3 > 0$, $\xi_3 < 0$. We find:

$$D_1^3 = -1 , \quad D_1^2D_2 = 0 , \quad D_1D_2^2 = -\frac{1}{2} , \quad D_2^3 = -\frac{1}{4} . \tag{B.23}$$

### B.3.3 Resolutions of the $E_2$ singularity (local dP$_2$)

Consider the $E_2$ singularity, studied in section 4.3. It admits 5 distinct resolutions, shown in Figure 9. The toric divisors $D_1, \cdots, D_5$ and $\mathbf{E}_0$ are as shown in Figure 10, with the linear equivalences:

$$D_1 \sim D_3 + D_5, \quad D_2 \sim D_4 + D_5 , \quad \mathbf{E}_0 \sim -2D_1 - 2D_2 + D_5 . \tag{B.24}$$

Let us focus on the three resolutions (a), (b), (c) with an allowed vertical reduction—one can similarly consider the resolutions (d) and (e). The redundant GLSM, showing the intersections between toric divisors and curves, are as follows:

$$(\mathbf{a}) \quad : \quad \left\{ \begin{array}{c|cccccc|c} & D_1 & D_2 & D_3 & D_4 & D_5 & \mathbf{E}_0 & \\ \hline \mathcal{C}_1 & 1 & 0 & 1 & 0 & 0 & -2 & \xi_1 \\ \mathcal{C}_2 & 0 & 1 & 0 & 1 & 0 & -2 & \xi_2 \\ \mathcal{C}_3 & 1 & 0 & 0 & -1 & 1 & -1 & \xi_3 \\ \mathcal{C}_4 & 0 & 1 & -1 & 0 & 1 & -1 & \xi_4 \\ \mathcal{C}_5 & 0 & 0 & 1 & 1 & -1 & -1 & \xi_5 \end{array} \right. \tag{B.25}$$

$$(\mathbf{b}) \quad : \quad \left\{ \begin{array}{c|cccccc|c} & D_1 & D_2 & D_3 & D_4 & D_5 & \mathbf{E}_0 & \\ \hline \mathcal{C}_1 & 1 & 0 & 1 & 0 & 0 & -2 & \xi_1 \\ \mathcal{C}_2 & 0 & 1 & 0 & 1 & 0 & -2 & \xi_2 \\ \mathcal{C}_3' & 1 & 0 & 1 & 0 & 0 & -2 & \xi_3+\xi_5 \\ \mathcal{C}_4' & 0 & 1 & 0 & 1 & 0 & -2 & \xi_4+\xi_5 \\ \bar{\mathcal{C}}_5 & 0 & 0 & -1 & -1 & 1 & 1 & -\xi_5 \end{array} \right. \tag{B.26}$$

$$(\mathbf{c}) \quad : \quad \left\{ \begin{array}{c|cccccc|c} & D_1 & D_2 & D_3 & D_4 & D_5 & \mathbf{E}_0 & \\ \hline \mathcal{C}'_1 & 1 & 1 & 0 & 0 & 1 & -3 & \xi_1 + \xi_4 \\ \mathcal{C}_2 & 0 & 1 & 0 & 1 & 0 & -2 & \xi_2 \\ \mathcal{C}_3 & 1 & 0 & 0 & -1 & 1 & -1 & \xi_3 \\ \bar{\mathcal{C}}_4 & 0 & -1 & 1 & 0 & -1 & 1 & -\xi_4 \\ \mathcal{C}'_5 & 0 & 1 & 0 & 1 & 0 & -2 & \xi_4 + \xi_5 \end{array} \right. \tag{B.27}$$

**Intersection numbers.** To compute the geometric prepotential using $S$ in (4.34), the relevant intersection numbers involving $\mathbf{E}_0$ are:

$$\begin{array}{c|c|c|c|c|c} & (a) & (b) & (c) & (d) & (e) \\ \hline \mathbf{E}_0^3 & 7 & 8 & 8 & 8 & 9 \\ \hline \mathbf{E}_0^2 D_1 & -2 & -2 & -2 & -3 & -3 \\ \hline \mathbf{E}_0^2 D_5 & -1 & 0 & -2 & -2 & -3 \\ \hline \mathbf{E}_0 D_1^2 & 0 & 0 & 0 & 1 & 1 \\ \hline \mathbf{E}_0 D_5^2 & -1 & 0 & 0 & 0 & 1 \\ \hline \mathbf{E}_0 D_1 D_5 & 0 & 0 & 0 & 1 & 1 \end{array}, \tag{B.28}$$

in the five resolutions. Moreover, one can again define the "non-compact" intersection numbers using the JK residue. We choose: [34]

$$D_i D_j D_k = \lim_{(\lambda_1,\lambda_2,\lambda_3,\lambda_4,\lambda_5) \to (0,1,0,0,0)\lambda} \langle D_i(\sigma) D_j(\sigma) D_j(\sigma) \rangle_0 . \tag{B.29}$$

For the relevant intersections amongst $D_1$ and $D_5$, this gives:

$$\begin{array}{c|c|c|c|c|c} & (a) & (b) & (c) & (d) & (e) \\ \hline D_1^3 & 0 & 0 & 0 & -1 & -1 \\ \hline D_1^2 D_5 & 0 & 0 & 0 & -1 & -1 \\ \hline D_1 D_5^2 & 0 & 0 & 0 & -1 & -1 \\ \hline D_5^3 & 1 & 2 & 0 & 0 & -1 \end{array} \tag{B.30}$$

For the resolutions (a), (b) and (c), which admit a vertical reduction, plugging the intersection numbers (B.28) and (B.30) into the M-theory prepotential reproduces precisely the field-theory prepotential, including the constant term.

### B.3.4 Resolutions of the $E_3$ singularity (local dP$_3$)

Consider the $E_3$ singularity, studied in section 4.4. It admits 18 distinct resolutions, shown in Figure 11. In the following, we list the triple intersection numbers that involve at least one compact divisor, in all 9 resolutions with an allowed vertical reduction. The toric divisors $D_1, \cdots, D_6$ and $\mathbf{E}_0$ are as indicated in Figure 12, with the linear equivalences:

$$D_4 \sim D_1 + D_2 - D_5 , \quad D_6 \sim D_2 + D_3 - D_5 , \quad \mathbf{E}_0 \sim -2D_1 - 3D_2 - 2D_3 + D_5 . \tag{B.31}$$

We find:

$$(\mathbf{a}) \quad : \quad \left\{ \begin{array}{l} \mathbf{E}_0^3 = 6 \; , \quad D_1 \mathbf{E}_0^2 = -1 \; , \quad D_2 \mathbf{E}_0^2 = -1 \; , \quad D_3 \mathbf{E}_0^2 = -1 \; , \\ D_4 \mathbf{E}_0^2 = -1 \; , \quad D_5 \mathbf{E}_0^2 = -1 \; , \quad D_6 \mathbf{E}_0^2 = -1 \; , \quad D_1 D_2 \mathbf{E}_0 = 1 \; , \\ D_1 D_3 \mathbf{E}_0 = 0 \; , \quad D_1 D_4 \mathbf{E}_0 = 0 \; , \quad D_1 D_5 \mathbf{E}_0 = 0 \; , \quad D_1 D_6 \mathbf{E}_0 = 1 \; , \\ D_2 D_3 \mathbf{E}_0 = 1 \; , \quad D_2 D_4 \mathbf{E}_0 = 0 \; , \quad D_2 D_5 \mathbf{E}_0 = 0 \; , \quad D_2 D_6 \mathbf{E}_0 = 0 \; , \\ D_3 D_4 \mathbf{E}_0 = 1 \; , \quad D_3 D_5 \mathbf{E}_0 = 0 \; , \quad D_3 D_6 \mathbf{E}_0 = 0 \; , \quad D_4 D_5 \mathbf{E}_0 = 1 \; , \\ D_4 D_6 \mathbf{E}_0 = 0 \; , \quad D_5 D_6 \mathbf{E}_0 = 1 \; , \quad D_1^2 \mathbf{E}_0 = -1 \; , \quad D_2^2 \mathbf{E}_0 = -1 \; , \\ D_3^2 \mathbf{E}_0 = -1 \; , \quad D_4^2 \mathbf{E}_0 = -1 \; , \quad D_5^2 \mathbf{E}_0 = -1 \; , \quad D_6^2 \mathbf{E}_0 = -1 \; , \end{array} \right. \tag{B.32}$$

---

[34]One can choose $\eta$ in the JK residue to be $\eta_a = (2, 2, 1)$, $\eta_b = (1, 1, -1)$, $\eta_c = (3, 1, 2)$, $\eta_d = (1, 3, 2)$, and $\eta_e = (2, 2, 3)$, respectively, for the five resolutions.

$$\textbf{(b)} \quad : \quad \begin{cases}
\mathbf{E}_0^3 = 7 \;, & D_1\mathbf{E}_0^2 = 0 \;, & D_2\mathbf{E}_0^2 = -2 \;, & D_3\mathbf{E}_0^2 = -1 \;, \\
D_4\mathbf{E}_0^2 = -1 \;, & D_5\mathbf{E}_0^2 = -1 \;, & D_6\mathbf{E}_0^2 = -2 \;, & D_1D_2\mathbf{E}_0 = 0 \;, \\
D_1D_3\mathbf{E}_0 = 0 \;, & D_1D_4\mathbf{E}_0 = 0 \;, & D_1D_5\mathbf{E}_0 = 0 \;, & D_1D_6\mathbf{E}_0 = 0 \;, \\
D_2D_3\mathbf{E}_0 = 1 \;, & D_2D_4\mathbf{E}_0 = 0 \;, & D_2D_5\mathbf{E}_0 = 0 \;, & D_2D_6\mathbf{E}_0 = 1 \;, \\
D_3D_4\mathbf{E}_0 = 1 \;, & D_3D_5\mathbf{E}_0 = 0 \;, & D_3D_6\mathbf{E}_0 = 0 \;, & D_4D_5\mathbf{E}_0 = 1 \;, \\
D_4D_6\mathbf{E}_0 = 0 \;, & D_5D_6\mathbf{E}_0 = 1 \;, & D_1^2\mathbf{E}_0 = 0 \;, & D_2^2\mathbf{E}_0 = 0 \;, \\
D_3^2\mathbf{E}_0 = -1 \;, & D_4^2\mathbf{E}_0 = -1 \;, & D_5^2\mathbf{E}_0 = -1 \;, & D_6^2\mathbf{E}_0 = 0 \;,
\end{cases} \tag{B.33}$$

$$\textbf{(c)} \quad : \quad \begin{cases}
\mathbf{E}_0^3 = 7 \;, & D_1\mathbf{E}_0^2 = -2 \;, & D_2\mathbf{E}_0^2 = 0 \;, & D_3\mathbf{E}_0^2 = -2 \;, \\
D_4\mathbf{E}_0^2 = -1 \;, & D_5\mathbf{E}_0^2 = -1 \;, & D_6\mathbf{E}_0^2 = -1 \;, & D_1D_2\mathbf{E}_0 = 0 \;, \\
D_1D_3\mathbf{E}_0 = 1 \;, & D_1D_4\mathbf{E}_0 = 0 \;, & D_1D_5\mathbf{E}_0 = 0 \;, & D_1D_6\mathbf{E}_0 = 1 \;, \\
D_2D_3\mathbf{E}_0 = 0 \;, & D_2D_4\mathbf{E}_0 = 0 \;, & D_2D_5\mathbf{E}_0 = 0 \;, & D_2D_6\mathbf{E}_0 = 0 \;, \\
D_3D_4\mathbf{E}_0 = 1 \;, & D_3D_5\mathbf{E}_0 = 0 \;, & D_3D_6\mathbf{E}_0 = 0 \;, & D_4D_5\mathbf{E}_0 = 1 \;, \\
D_4D_6\mathbf{E}_0 = 0 \;, & D_5D_6\mathbf{E}_0 = 1 \;, & D_1^2\mathbf{E}_0 = 0 \;, & D_2^2\mathbf{E}_0 = 0 \;, \\
D_3^2\mathbf{E}_0 = 0 \;, & D_4^2\mathbf{E}_0 = -1 \;, & D_5^2\mathbf{E}_0 = -1 \;, & D_6^2\mathbf{E}_0 = -1 \;,
\end{cases} \tag{B.34}$$

$$\textbf{(d)} \quad : \quad \begin{cases}
\mathbf{E}_0^3 = 7 \;, & D_1\mathbf{E}_0^2 = -1 \;, & D_2\mathbf{E}_0^2 = -1 \;, & D_3\mathbf{E}_0^2 = -2 \;, \\
D_4\mathbf{E}_0^2 = 0 \;, & D_5\mathbf{E}_0^2 = -2 \;, & D_6\mathbf{E}_0^2 = -1 \;, & D_1D_2\mathbf{E}_0 = 1 \;, \\
D_1D_3\mathbf{E}_0 = 0 \;, & D_1D_4\mathbf{E}_0 = 0 \;, & D_1D_5\mathbf{E}_0 = 0 \;, & D_1D_6\mathbf{E}_0 = 1 \;, \\
D_2D_3\mathbf{E}_0 = 1 \;, & D_2D_4\mathbf{E}_0 = 0 \;, & D_2D_5\mathbf{E}_0 = 0 \;, & D_2D_6\mathbf{E}_0 = 0 \;, \\
D_3D_4\mathbf{E}_0 = 0 \;, & D_3D_5\mathbf{E}_0 = 1 \;, & D_3D_6\mathbf{E}_0 = 0 \;, & D_4D_5\mathbf{E}_0 = 0 \;, \\
D_4D_6\mathbf{E}_0 = 0 \;, & D_5D_6\mathbf{E}_0 = 1 \;, & D_1^2\mathbf{E}_0 = -1 \;, & D_2^2\mathbf{E}_0 = -1 \;, \\
D_3^2\mathbf{E}_0 = 0 \;, & D_4^2\mathbf{E}_0 = 0 \;, & D_5^2\mathbf{E}_0 = 0 \;, & D_6^2\mathbf{E}_0 = -1 \;,
\end{cases} \tag{B.35}$$

$$\textbf{(e)} \quad : \quad \begin{cases}
\mathbf{E}_0^3 = 7 \;, & D_1\mathbf{E}_0^2 = -1 \;, & D_2\mathbf{E}_0^2 = -1 \;, & D_3\mathbf{E}_0^2 = -1 \;, \\
D_4\mathbf{E}_0^2 = -2 \;, & D_5\mathbf{E}_0^2 = 0 \;, & D_6\mathbf{E}_0^2 = -2 \;, & D_1D_2\mathbf{E}_0 = 1 \;, \\
D_1D_3\mathbf{E}_0 = 0 \;, & D_1D_4\mathbf{E}_0 = 0 \;, & D_1D_5\mathbf{E}_0 = 0 \;, & D_1D_6\mathbf{E}_0 = 1 \;, \\
D_2D_3\mathbf{E}_0 = 1 \;, & D_2D_4\mathbf{E}_0 = 0 \;, & D_2D_5\mathbf{E}_0 = 0 \;, & D_2D_6\mathbf{E}_0 = 0 \;, \\
D_3D_4\mathbf{E}_0 = 1 \;, & D_3D_5\mathbf{E}_0 = 0 \;, & D_3D_6\mathbf{E}_0 = 0 \;, & D_4D_5\mathbf{E}_0 = 0 \;, \\
D_4D_6\mathbf{E}_0 = 1 \;, & D_5D_6\mathbf{E}_0 = 0 \;, & D_1^2\mathbf{E}_0 = -1 \;, & D_2^2\mathbf{E}_0 = -1 \;, \\
D_3^2\mathbf{E}_0 = -1 \;, & D_4^2\mathbf{E}_0 = 0 \;, & D_5^2\mathbf{E}_0 = 0 \;, & D_6^2\mathbf{E}_0 = 0 \;,
\end{cases} \tag{B.36}$$

$$\textbf{(f)} \quad : \quad \begin{cases}
\mathbf{E}_0^3 = 8 \;, & D_1\mathbf{E}_0^2 = 0 \;, & D_2\mathbf{E}_0^2 = -2 \;, & D_3\mathbf{E}_0^2 = -2 \;, \\
D_4\mathbf{E}_0^2 = 0 \;, & D_5\mathbf{E}_0^2 = -2 \;, & D_6\mathbf{E}_0^2 = -2 \;, & D_1D_2\mathbf{E}_0 = 0 \;, \\
D_1D_3\mathbf{E}_0 = 0 \;, & D_1D_4\mathbf{E}_0 = 0 \;, & D_1D_5\mathbf{E}_0 = 0 \;, & D_1D_6\mathbf{E}_0 = 0 \;, \\
D_2D_3\mathbf{E}_0 = 1 \;, & D_2D_4\mathbf{E}_0 = 0 \;, & D_2D_5\mathbf{E}_0 = 0 \;, & D_2D_6\mathbf{E}_0 = 1 \;, \\
D_3D_4\mathbf{E}_0 = 0 \;, & D_3D_5\mathbf{E}_0 = 1 \;, & D_3D_6\mathbf{E}_0 = 0 \;, & D_4D_5\mathbf{E}_0 = 0 \;, \\
D_4D_6\mathbf{E}_0 = 0 \;, & D_5D_6\mathbf{E}_0 = 1 \;, & D_1^2\mathbf{E}_0 = 0 \;, & D_2^2\mathbf{E}_0 = 0 \;, \\
D_3^2\mathbf{E}_0 = 0 \;, & D_4^2\mathbf{E}_0 = 0 \;, & D_5^2\mathbf{E}_0 = 0 \;, & D_6^2\mathbf{E}_0 = 0 \;,
\end{cases} \tag{B.37}$$

$$\textbf{(g)} \quad : \quad \begin{cases}
\mathbf{E}_0^3 = 8 \;, & D_1\mathbf{E}_0^2 = -2 \;, & D_2\mathbf{E}_0^2 = 0 \;, & D_3\mathbf{E}_0^2 = -2 \;, \\
D_4\mathbf{E}_0^2 = -2 \;, & D_5\mathbf{E}_0^2 = 0 \;, & D_6\mathbf{E}_0^2 = -2 \;, & D_1D_2\mathbf{E}_0 = 0 \;, \\
D_1D_3\mathbf{E}_0 = 1 \;, & D_1D_4\mathbf{E}_0 = 0 \;, & D_1D_5\mathbf{E}_0 = 0 \;, & D_1D_6\mathbf{E}_0 = 1 \;, \\
D_2D_3\mathbf{E}_0 = 0 \;, & D_2D_4\mathbf{E}_0 = 0 \;, & D_2D_5\mathbf{E}_0 = 0 \;, & D_2D_6\mathbf{E}_0 = 0 \;, \\
D_3D_4\mathbf{E}_0 = 1 \;, & D_3D_5\mathbf{E}_0 = 0 \;, & D_3D_6\mathbf{E}_0 = 0 \;, & D_4D_5\mathbf{E}_0 = 0 \;, \\
D_4D_6\mathbf{E}_0 = 1 \;, & D_5D_6\mathbf{E}_0 = 0 \;, & D_1^2\mathbf{E}_0 = 0 \;, & D_2^2\mathbf{E}_0 = 0 \;, \\
D_3^2\mathbf{E}_0 = 0 \;, & D_4^2\mathbf{E}_0 = 0 \;, & D_5^2\mathbf{E}_0 = 0 \;, & D_6^2\mathbf{E}_0 = 0 \;,
\end{cases} \tag{B.38}$$

$$\textbf{(h)} \quad : \quad \begin{cases}
\mathbf{E}_0^3 = 8 \;, & D_1\mathbf{E}_0^2 = 0 \;, & D_2\mathbf{E}_0^2 = -2 \;, & D_3\mathbf{E}_0^2 = -1 \;, \\
D_4\mathbf{E}_0^2 = -2 \;, & D_5\mathbf{E}_0^2 = 0 \;, & D_6\mathbf{E}_0^2 = -3 \;, & D_1D_2\mathbf{E}_0 = 0 \;, \\
D_1D_3\mathbf{E}_0 = 0 \;, & D_1D_4\mathbf{E}_0 = 0 \;, & D_1D_5\mathbf{E}_0 = 0 \;, & D_1D_6\mathbf{E}_0 = 0 \;, \\
D_2D_3\mathbf{E}_0 = 1 \;, & D_2D_4\mathbf{E}_0 = 0 \;, & D_2D_5\mathbf{E}_0 = 0 \;, & D_2D_6\mathbf{E}_0 = 1 \;, \\
D_3D_4\mathbf{E}_0 = 1 \;, & D_3D_5\mathbf{E}_0 = 0 \;, & D_3D_6\mathbf{E}_0 = 0 \;, & D_4D_5\mathbf{E}_0 = 0 \;, \\
D_4D_6\mathbf{E}_0 = 1 \;, & D_5D_6\mathbf{E}_0 = 0 \;, & D_1^2\mathbf{E}_0 = 0 \;, & D_2^2\mathbf{E}_0 = 0 \;, \\
D_3^2\mathbf{E}_0 = -1 \;, & D_4^2\mathbf{E}_0 = 0 \;, & D_5^2\mathbf{E}_0 = 0 \;, & D_6^2\mathbf{E}_0 = 1 \;,
\end{cases} \tag{B.39}$$

$$\textbf{(i)} \quad : \quad \begin{cases}
\mathbf{E}_0^3 = 8 \;, & D_1\mathbf{E}_0^2 = -2 \;, & D_2\mathbf{E}_0^2 = 0 \;, & D_3\mathbf{E}_0^2 = -3 \;, \\
D_4\mathbf{E}_0^2 = 0 \;, & D_5\mathbf{E}_0^2 = -2 \;, & D_6\mathbf{E}_0^2 = -1 \;, & D_1D_2\mathbf{E}_0 = 0 \;, \\
D_1D_3\mathbf{E}_0 = 1 \;, & D_1D_4\mathbf{E}_0 = 0 \;, & D_1D_5\mathbf{E}_0 = 0 \;, & D_1D_6\mathbf{E}_0 = 1 \;, \\
D_2D_3\mathbf{E}_0 = 0 \;, & D_2D_4\mathbf{E}_0 = 0 \;, & D_2D_5\mathbf{E}_0 = 0 \;, & D_2D_6\mathbf{E}_0 = 0 \;, \\
D_3D_4\mathbf{E}_0 = 0 \;, & D_3D_5\mathbf{E}_0 = 1 \;, & D_3D_6\mathbf{E}_0 = 0 \;, & D_4D_5\mathbf{E}_0 = 0 \;, \\
D_4D_6\mathbf{E}_0 = 0 \;, & D_5D_6\mathbf{E}_0 = 1 \;, & D_1^2\mathbf{E}_0 = 0 \;, & D_2^2\mathbf{E}_0 = 0 \;, \\
D_3^2\mathbf{E}_0 = 1 \;, & D_4^2\mathbf{E}_0 = 0 \;, & D_5^2\mathbf{E}_0 = 0 \;, & D_6^2\mathbf{E}_0 = -1 \;.
\end{cases} \tag{B.40}$$

### B.3.5 Resolutions of the beetle geometry

Consider the "beetle singularity" that we studied in section 6.2. It admits 24 distinct resolutions, shown in Figure 17. The toric divisors $D_1, \cdots, D_6$ and $\mathbf{E}_1, \mathbf{E}_2$ are as shown in Figure 16, with the linear equivalences:

$$
\begin{aligned}
D_4 &\sim D_1 - D_3 + D_6 \,, &\qquad \mathbf{E}_1 &\sim -2D_1 - 2D_2 + D_4 + D_5 - D_6 \,, \\
\mathbf{E}_2 &\sim D_2 - D_4 - 2D_5 - D_6 \,.
\end{aligned}
\tag{B.41}
$$

In the following, we compute the intersection numbers for every resolution, and we verify that the geometric prepotential matches precisely with the $SU(2) \times SU(2)$ and/or with the $SU(3)$, $N_f = 2$ prepotential, whenever a gauge-theory interpretation exists.

**Resolutions with a vertical reduction.** The resolutions (a) to (f) have an allowed vertical reduction. Their intersection numbers are:

$$
\textbf{(a)}: \left\{
\begin{array}{lllll}
E_1^3 = 8 \,, & E_2^3 = 6 \,, & D_1^2 E_2 = -1 \,, & D_1 D_2 E_2 = 0 \,, & D_2^2 E_2 = 0 \,, \\
D_1 D_6 E_2 = 1 \,, & D_2 D_6 E_2 = 0 \,, & D_6^2 E_2 = -1 \,, & D_1 E_1^2 = -2 \,, & D_2 E_1^2 = -2 \,, \\
D_6 E_1^2 = 0 \,, & E_1^2 E_2 = -2 \,, & D_1 E_2^2 = -1 \,, & D_2 E_2^2 = 0 \,, & D_6 E_2^2 = -1 \,, \\
D_1^2 E_1 = 0 \,, & D_1 D_2 E_1 = 1 \,, & D_2^2 E_1 = 0 \,, & D_1 D_6 E_1 = 0 \,, & D_2 D_6 E_1 = 0 \,, \\
D_6^2 E_1 = 0 \,, & E_1 E_2^2 = 0 \,, & D_1 E_1 E_2 = 1 \,, & D_2 E_1 E_2 = 0 \,, & D_6 E_1 E_2 = 0 \,,
\end{array}
\right.
\tag{B.42}
$$

$$
\textbf{(b)}: \left\{
\begin{array}{lllll}
E_1^3 = 7 \,, & E_2^3 = 7 \,, & D_1^2 E_2 = -1 \,, & D_1 D_2 E_2 = 0 \,, & D_2^2 E_2 = 0 \,, \\
D_1 D_6 E_2 = 1 \,, & D_2 D_6 E_2 = 0 \,, & D_6^2 E_2 = -1 \,, & D_1 E_1^2 = -2 \,, & D_2 E_1^2 = -2 \,, \\
D_6 E_1^2 = 0 \,, & E_1^2 E_2 = -1 \,, & D_1 E_2^2 = -1 \,, & D_2 E_2^2 = 0 \,, & D_6 E_2^2 = -1 \,, \\
D_1^2 E_1 = 0 \,, & D_1 D_2 E_1 = 1 \,, & D_2^2 E_1 = 0 \,, & D_1 D_6 E_1 = 0 \,, & D_2 D_6 E_1 = 0 \,, \\
D_6^2 E_1 = 0 \,, & E_1 E_2^2 = -1 \,, & D_1 E_1 E_2 = 1 \,, & D_2 E_1 E_2 = 0 \,, & D_6 E_1 E_2 = 0 \,,
\end{array}
\right.
\tag{B.43}
$$

$$
\textbf{(c)}: \left\{
\begin{array}{lllll}
E_1^3 = 8 \,, & E_2^3 = 8 \,, & D_1^2 E_2 = 0 \,, & D_1 D_2 E_2 = 0 \,, & D_2^2 E_2 = 0 \,, \\
D_1 D_6 E_2 = 1 \,, & D_2 D_6 E_2 = 0 \,, & D_6^2 E_2 = -1 \,, & D_1 E_1^2 = -3 \,, & D_2 E_1^2 = -2 \,, \\
D_6 E_1^2 = 0 \,, & E_1^2 E_2 = 0 \,, & D_1 E_2^2 = -2 \,, & D_2 E_2^2 = 0 \,, & D_6 E_2^2 = -1 \,, \\
D_1^2 E_1 = 1 \,, & D_1 D_2 E_1 = 1 \,, & D_2^2 E_1 = 0 \,, & D_1 D_6 E_1 = 0 \,, & D_2 D_6 E_1 = 0 \,, \\
D_6^2 E_1 = 0 \,, & E_1 E_2^2 = 0 \,, & D_1 E_1 E_2 = 0 \,, & D_2 E_1 E_2 = 0 \,, & D_6 E_1 E_2 = 0 \,,
\end{array}
\right.
\tag{B.44}
$$

$$
\textbf{(d)}: \left\{
\begin{array}{lllll}
E_1^3 = 6 \,, & E_2^3 = 8 \,, & D_1^2 E_2 = 0 \,, & D_1 D_2 E_2 = 0 \,, & D_2^2 E_2 = 0 \,, \\
D_1 D_6 E_2 = 0 \,, & D_2 D_6 E_2 = 0 \,, & D_6^2 E_2 = 0 \,, & D_1 E_1^2 = -1 \,, & D_2 E_1^2 = -2 \,, \\
D_6 E_1^2 = -1 \,, & E_1^2 E_2 = 0 \,, & D_1 E_2^2 = 0 \,, & D_2 E_2^2 = 0 \,, & D_6 E_2^2 = -2 \,, \\
D_1^2 E_1 = -1 \,, & D_1 D_2 E_1 = 1 \,, & D_2^2 E_1 = 0 \,, & D_1 D_6 E_1 = 1 \,, & D_2 D_6 E_1 = 0 \,, \\
D_6^2 E_1 = -1 \,, & E_1 E_2^2 = -2 \,, & D_1 E_1 E_2 = 0 \,, & D_2 E_1 E_2 = 0 \,, & D_6 E_1 E_2 = 1 \,,
\end{array}
\right.
\tag{B.45}
$$

$$
\textbf{(e)}: \left\{
\begin{array}{lllll}
E_1^3 = 7 \,, & E_2^3 = 7 \,, & D_1^2 E_2 = 0 \,, & D_1 D_2 E_2 = 0 \,, & D_2^2 E_2 = 0 \,, \\
D_1 D_6 E_2 = 0 \,, & D_2 D_6 E_2 = 0 \,, & D_6^2 E_2 = 0 \,, & D_1 E_1^2 = -1 \,, & D_2 E_1^2 = -2 \,, \\
D_6 E_1^2 = -1 \,, & E_1^2 E_2 = -1 \,, & D_1 E_2^2 = 0 \,, & D_2 E_2^2 = 0 \,, & D_6 E_2^2 = -2 \,, \\
D_1^2 E_1 = -1 \,, & D_1 D_2 E_1 = 1 \,, & D_2^2 E_1 = 0 \,, & D_1 D_6 E_1 = 1 \,, & D_2 D_6 E_1 = 0 \,, \\
D_6^2 E_1 = -1 \,, & E_1 E_2^2 = -1 \,, & D_1 E_1 E_2 = 0 \,, & D_2 E_1 E_2 = 0 \,, & D_6 E_1 E_2 = 1 \,,
\end{array}
\right.
\tag{B.46}
$$

$$
\textbf{(f)}: \left\{
\begin{array}{lllll}
E_1^3 = 8 \,, & E_2^3 = 8 \,, & D_1^2 E_2 = 0 \,, & D_1 D_2 E_2 = 0 \,, & D_2^2 E_2 = 0 \,, \\
D_1 D_6 E_2 = 0 \,, & D_2 D_6 E_2 = 0 \,, & D_6^2 E_2 = 1 \,, & D_1 E_1^2 = -1 \,, & D_2 E_1^2 = -2 \,, \\
D_6 E_1^2 = -2 \,, & E_1^2 E_2 = 0 \,, & D_1 E_2^2 = 0 \,, & D_2 E_2^2 = 0 \,, & D_6 E_2^2 = -3 \,, \\
D_1^2 E_1 = -1 \,, & D_1 D_2 E_1 = 1 \,, & D_2^2 E_1 = 0 \,, & D_1 D_6 E_1 = 1 \,, & D_2 D_6 E_1 = 0 \,, \\
D_6^2 E_1 = 0 \,, & E_1 E_2^2 = 0 \,, & D_1 E_1 E_2 = 0 \,, & D_2 E_1 E_2 = 0 \,, & D_6 E_1 E_2 = 0 \,.
\end{array}
\right.
\tag{B.47}
$$

**Resolutions with an horizontal reduction.** The resolutions (g) to (r) have an horizontal reduction. So do the resolutions (a), (b), (d), (e) above. The intersection numbers are:

$$
\textbf{(g)}: \left\{
\begin{array}{lllll}
E_1^3 = 8 \,, & E_2^3 = 7 \,, & D_1^2 E_2 = -1 \,, & D_1 D_2 E_2 = 0 \,, & D_2^2 E_2 = 0 \,, \\
D_1 D_6 E_2 = 1 \,, & D_2 D_6 E_2 = 0 \,, & D_6^2 E_2 = -1 \,, & D_1 E_1^2 = -2 \,, & D_2 E_1^2 = -2 \,, \\
D_6 E_1^2 = 0 \,, & E_1^2 E_2 = -2 \,, & D_1 E_2^2 = -1 \,, & D_2 E_2^2 = 0 \,, & D_6 E_2^2 = -1 \,, \\
D_1^2 E_1 = 0 \,, & D_1 D_2 E_1 = 1 \,, & D_2^2 E_1 = 0 \,, & D_1 D_6 E_1 = 0 \,, & D_2 D_6 E_1 = 0 \,, \\
D_6^2 E_1 = 0 \,, & E_1 E_2^2 = 0 \,, & D_1 E_1 E_2 = 1 \,, & D_2 E_1 E_2 = 0 \,, & D_6 E_1 E_2 = 0 \,,
\end{array}
\right.
\tag{B.48}
$$

$$
\textbf{(h)}: \begin{cases}
E_1^3 = 8 \quad, & E_2^3 = 7 \quad, & D_1^2 E_2 = 0 \quad, & D_1 D_2 E_2 = 0 \,, & D_2^2 E_2 = 0 \quad, \\
D_1 D_6 E_2 = 0 \,, & D_2 D_6 E_2 = 0 \,, & D_6^2 E_2 = 0 \quad, & D_1 E_1^2 = -2 \,, & D_2 E_1^2 = -2 \quad, \\
D_6 E_1^2 = 0 \quad, & E_1^2 E_2 = -2 \,, & D_1 E_2^2 = -2 \,, & D_2 E_2^2 = 0 \quad, & D_6 E_2^2 = 0 \quad, \\
D_1^2 E_1 = 0 \quad, & D_1 D_2 E_1 = 1 \,, & D_2^2 E_1 = 0 \quad, & D_1 D_6 E_1 = 0 \,, & D_2 D_6 E_1 = 0 \,, \\
D_6^2 E_1 = 0 \quad, & E_1 E_2^2 = 0 \quad, & D_1 E_1 E_2 = 1 \,, & D_2 E_1 E_2 = 0 \,, & D_6 E_1 E_2 = 0 \,,
\end{cases}
\tag{B.49}
$$

$$
\textbf{(i)}: \begin{cases}
E_1^3 = 8 \quad, & E_2^3 = 8 \quad, & D_1^2 E_2 = 0 \quad, & D_1 D_2 E_2 = 0 \,, & D_2^2 E_2 = 0 \quad, \\
D_1 D_6 E_2 = 0 \,, & D_2 D_6 E_2 = 0 \,, & D_6^2 E_2 = 0 \quad, & D_1 E_1^2 = -2 \,, & D_2 E_1^2 = -2 \quad, \\
D_6 E_1^2 = 0 \quad, & E_1^2 E_2 = -2 \,, & D_1 E_2^2 = -2 \,, & D_2 E_2^2 = 0 \quad, & D_6 E_2^2 = 0 \quad, \\
D_1^2 E_1 = 0 \quad, & D_1 D_2 E_1 = 1 \,, & D_2^2 E_1 = 0 \quad, & D_1 D_6 E_1 = 0 \,, & D_2 D_6 E_1 = 0 \,, \\
D_6^2 E_1 = 0 \quad, & E_1 E_2^2 = 0 \quad, & D_1 E_1 E_2 = 1 \,, & D_2 E_1 E_2 = 0 \,, & D_6 E_1 E_2 = 0 \,,
\end{cases}
\tag{B.50}
$$

$$
\textbf{(j)}: \begin{cases}
E_1^3 = 8 \quad, & E_2^3 = 7 \quad, & D_1^2 E_2 = -1 \,, & D_1 D_2 E_2 = 0 \,, & D_2^2 E_2 = 0 \quad, \\
D_1 D_6 E_2 = 1 \,, & D_2 D_6 E_2 = 0 \,, & D_6^2 E_2 = -1 \,, & D_1 E_1^2 = -2 \,, & D_2 E_1^2 = -3 \quad, \\
D_6 E_1^2 = 0 \quad, & E_1^2 E_2 = -1 \,, & D_1 E_2^2 = -1 \,, & D_2 E_2^2 = 0 \quad, & D_6 E_2^2 = -1 \,, \\
D_1^2 E_1 = 0 \quad, & D_1 D_2 E_1 = 1 \,, & D_2^2 E_1 = 1 \quad, & D_1 D_6 E_1 = 0 \,, & D_2 D_6 E_1 = 0 \,, \\
D_6^2 E_1 = 0 \quad, & E_1 E_2^2 = -1 \,, & D_1 E_1 E_2 = 1 \,, & D_2 E_1 E_2 = 0 \,, & D_6 E_1 E_2 = 0 \,,
\end{cases}
\tag{B.51}
$$

$$
\textbf{(k)}: \begin{cases}
E_1^3 = 7 \quad, & E_2^3 = 8 \quad, & D_1^2 E_2 = 0 \quad, & D_1 D_2 E_2 = 0 \,, & D_2^2 E_2 = 0 \quad, \\
D_1 D_6 E_2 = 0 \,, & D_2 D_6 E_2 = 0 \,, & D_6^2 E_2 = 0 \quad, & D_1 E_1^2 = -2 \,, & D_2 E_1^2 = -2 \quad, \\
D_6 E_1^2 = 0 \quad, & E_1^2 E_2 = -1 \,, & D_1 E_2^2 = -2 \,, & D_2 E_2^2 = 0 \quad, & D_6 E_2^2 = 0 \quad, \\
D_1^2 E_1 = 0 \quad, & D_1 D_2 E_1 = 1 \,, & D_2^2 E_1 = 0 \quad, & D_1 D_6 E_1 = 0 \,, & D_2 D_6 E_1 = 0 \,, \\
D_6^2 E_1 = 0 \quad, & E_1 E_2^2 = -1 \,, & D_1 E_1 E_2 = 1 \,, & D_2 E_1 E_2 = 0 \,, & D_6 E_1 E_2 = 0 \,,
\end{cases}
\tag{B.52}
$$

$$
\textbf{(l)}: \begin{cases}
E_1^3 = 8 \quad, & E_2^3 = 8 \quad, & D_1^2 E_2 = 0 \quad, & D_1 D_2 E_2 = 0 \,, & D_2^2 E_2 = 0 \quad, \\
D_1 D_6 E_2 = 0 \,, & D_2 D_6 E_2 = 0 \,, & D_6^2 E_2 = 0 \quad, & D_1 E_1^2 = -2 \,, & D_2 E_1^2 = -3 \quad, \\
D_6 E_1^2 = 0 \quad, & E_1^2 E_2 = -1 \,, & D_1 E_2^2 = -2 \,, & D_2 E_2^2 = 0 \quad, & D_6 E_2^2 = 0 \quad, \\
D_1^2 E_1 = 0 \quad, & D_1 D_2 E_1 = 1 \,, & D_2^2 E_1 = 1 \quad, & D_1 D_6 E_1 = 0 \,, & D_2 D_6 E_1 = 0 \,, \\
D_6^2 E_1 = 0 \quad, & E_1 E_2^2 = -1 \,, & D_1 E_1 E_2 = 1 \,, & D_2 E_1 E_2 = 0 \,, & D_6 E_1 E_2 = 0 \,,
\end{cases}
\tag{B.53}
$$

$$
\textbf{(m)}: \begin{cases}
E_1^3 = 7 \quad, & E_2^3 = 8 \quad, & D_1^2 E_2 = 0 \quad, & D_1 D_2 E_2 = 0 \,, & D_2^2 E_2 = 0 \quad, \\
D_1 D_6 E_2 = 0 \,, & D_2 D_6 E_2 = 0 \,, & D_6^2 E_2 = 0 \quad, & D_1 E_1^2 = -1 \,, & D_2 E_1^2 = -3 \quad, \\
D_6 E_1^2 = -1 \,, & E_1^2 E_2 = 0 \quad, & D_1 E_2^2 = 0 \quad, & D_2 E_2^2 = 0 \quad, & D_6 E_2^2 = -2 \,, \\
D_1^2 E_1 = -1 \,, & D_1 D_2 E_1 = 1 \,, & D_2^2 E_1 = 1 \quad, & D_1 D_6 E_1 = 1 \,, & D_2 D_6 E_1 = 0 \,, \\
D_6^2 E_1 = -1 \,, & E_1 E_2^2 = -2 \,, & D_1 E_1 E_2 = 0 \,, & D_2 E_1 E_2 = 0 \,, & D_6 E_1 E_2 = 1 \,,
\end{cases}
\tag{B.54}
$$

$$
\textbf{(n)}: \begin{cases}
E_1^3 = 7 \quad, & E_2^3 = 8 \quad, & D_1^2 E_2 = 0 \quad, & D_1 D_2 E_2 = 0 \,, & D_2^2 E_2 = 0 \quad, \\
D_1 D_6 E_2 = 0 \,, & D_2 D_6 E_2 = 0 \,, & D_6^2 E_2 = 0 \quad, & D_1 E_1^2 = 0 \quad, & D_2 E_1^2 = -3 \quad, \\
D_6 E_1^2 = -2 \,, & E_1^2 E_2 = 0 \quad, & D_1 E_2^2 = 0 \quad, & D_2 E_2^2 = 0 \quad, & D_6 E_2^2 = -2 \,, \\
D_1^2 E_1 = 0 \quad, & D_1 D_2 E_1 = 0 \,, & D_2^2 E_1 = 1 \quad, & D_1 D_6 E_1 = 0 \,, & D_2 D_6 E_1 = 1 \,, \\
D_6^2 E_1 = 0 \quad, & E_1 E_2^2 = -2 \,, & D_1 E_1 E_2 = 0 \,, & D_2 E_1 E_2 = 0 \,, & D_6 E_1 E_2 = 1 \,,
\end{cases}
\tag{B.55}
$$

$$
\textbf{(o)}: \begin{cases}
E_1^3 = 8 \quad, & E_2^3 = 8 \quad, & D_1^2 E_2 = 0 \quad, & D_1 D_2 E_2 = 0 \,, & D_2^2 E_2 = 0 \quad, \\
D_1 D_6 E_2 = 0 \,, & D_2 D_6 E_2 = 0 \,, & D_6^2 E_2 = 0 \quad, & D_1 E_1^2 = 0 \quad, & D_2 E_1^2 = -4 \quad, \\
D_6 E_1^2 = -2 \,, & E_1^2 E_2 = 0 \quad, & D_1 E_2^2 = 0 \quad, & D_2 E_2^2 = 0 \quad, & D_6 E_2^2 = -2 \,, \\
D_1^2 E_1 = 0 \quad, & D_1 D_2 E_1 = 0 \,, & D_2^2 E_1 = 2 \quad, & D_1 D_6 E_1 = 0 \,, & D_2 D_6 E_1 = 1 \,, \\
D_6^2 E_1 = 0 \quad, & E_1 E_2^2 = -2 \,, & D_1 E_1 E_2 = 0 \,, & D_2 E_1 E_2 = 0 \,, & D_6 E_1 E_2 = 1 \,,
\end{cases}
\tag{B.56}
$$

$$
\textbf{(p)}: \begin{cases}
E_1^3 = 7 \quad, & E_2^3 = 8 \quad, & D_1^2 E_2 = 0 \quad, & D_1 D_2 E_2 = 0 \,, & D_2^2 E_2 = 0 \quad, \\
D_1 D_6 E_2 = 0 \,, & D_2 D_6 E_2 = 0 \,, & D_6^2 E_2 = 0 \quad, & D_1 E_1^2 = -1 \,, & D_2 E_1^2 = -2 \quad, \\
D_6 E_1^2 = -1 \,, & E_1^2 E_2 = -1 \,, & D_1 E_2^2 = 0 \quad, & D_2 E_2^2 = 0 \quad, & D_6 E_2^2 = -2 \,, \\
D_1^2 E_1 = -1 \,, & D_1 D_2 E_1 = 1 \,, & D_2^2 E_1 = 0 \quad, & D_1 D_6 E_1 = 1 \,, & D_2 D_6 E_1 = 0 \,, \\
D_6^2 E_1 = -1 \,, & E_1 E_2^2 = -1 \,, & D_1 E_1 E_2 = 0 \,, & D_2 E_1 E_2 = 0 \,, & D_6 E_1 E_2 = 1 \,,
\end{cases}
\tag{B.57}
$$

$$
\textbf{(q)}: \begin{cases}
E_1^3 = 8 \quad, & E_2^3 = 7 \quad, & D_1^2 E_2 = 0 \quad, & D_1 D_2 E_2 = 0 \,, & D_2^2 E_2 = 0 \quad, \\
D_1 D_6 E_2 = 0 \,, & D_2 D_6 E_2 = 0 \,, & D_6^2 E_2 = 0 \quad, & D_1 E_1^2 = 0 \quad, & D_2 E_1^2 = -3 \quad, \\
D_6 E_1^2 = -2 \,, & E_1^2 E_2 = -1 \,, & D_1 E_2^2 = 0 \quad, & D_2 E_2^2 = 0 \quad, & D_6 E_2^2 = -2 \,, \\
D_1^2 E_1 = 0 \quad, & D_1 D_2 E_1 = 0 \,, & D_2^2 E_1 = 1 \quad, & D_1 D_6 E_1 = 0 \,, & D_2 D_6 E_1 = 1 \,, \\
D_6^2 E_1 = 0 \quad, & E_1 E_2^2 = -1 \,, & D_1 E_1 E_2 = 0 \,, & D_2 E_1 E_2 = 0 \,, & D_6 E_1 E_2 = 1 \,,
\end{cases}
\tag{B.58}
$$

$$(\mathbf{r}): \begin{cases} E_1^3 = 8 \quad, \quad E_2^3 = 8 \quad, \quad D_1^2 E_2 = 0 \quad, \quad D_1 D_2 E_2 = 0, \quad D_2^2 E_2 = 0 \,, \\ D_1 D_6 E_2 = 0, \quad D_2 D_6 E_2 = 0, \quad D_6^2 E_2 = 0, \quad D_1 E_1^2 = 0, \quad D_2 E_1^2 = -3 \,, \\ D_6 E_1^2 = -2, \quad E_1^2 E_2 = -1, \quad D_1 E_2^2 = 0, \quad D_2 E_2^2 = 0, \quad D_6 E_2^2 = -2 \,, \\ D_1^2 E_1 = 0 \quad, \quad D_1 D_2 E_1 = 0, \quad D_2^2 E_1 = 1, \quad D_1 D_6 E_1 = 0, \quad D_2 D_6 E_1 = 1 \,, \\ D_6^2 E_1 = 0 \quad, \quad E_1 E_2^2 = -1, \quad D_1 E_1 E_2 = 0, \quad D_2 E_1 E_2 = 0, \quad D_6 E_1 E_2 = 1 \,. \end{cases} \tag{B.59}$$

**Resolutions without a gauge-theory phase.** For completeness, let us also give the intersection numbers for the resolutions (s)-(x), which do not admit a gauge-theory description.

$$(\mathbf{s}): \begin{cases} E_1^3 = 9 \quad, \quad E_2^3 = 8 \quad, \quad D_1^2 E_2 = 0 \quad, \quad D_1 D_2 E_2 = 0, \quad D_2^2 E_2 = 0 \,, \\ D_1 D_6 E_2 = 1, \quad D_2 D_6 E_2 = 0, \quad D_6^2 E_2 = -1, \quad D_1 E_1^2 = -3, \quad D_2 E_1^2 = -3 \,, \\ D_6 E_1^2 = 0, \quad E_1^2 E_2 = 0, \quad D_1 E_2^2 = -2, \quad D_2 E_2^2 = 0, \quad D_6 E_2^2 = -1 \,, \\ D_1^2 E_1 = 1, \quad D_1 D_2 E_1 = 1, \quad D_2^2 E_1 = 1, \quad D_1 D_6 E_1 = 0, \quad D_2 D_6 E_1 = 0 \,, \\ D_6^2 E_1 = 0 \quad, \quad E_1 E_2^2 = 0, \quad D_1 E_1 E_2 = 0, \quad D_2 E_1 E_2 = 0, \quad D_6 E_1 E_2 = 0 \,, \end{cases} \tag{B.60}$$

$$(\mathbf{t}): \begin{cases} E_1^3 = 8 \quad, \quad E_2^3 = 9 \quad, \quad D_1^2 E_2 = 1 \quad, \quad D_1 D_2 E_2 = 0, \quad D_2^2 E_2 = 0 \,, \\ D_1 D_6 E_2 = 0, \quad D_2 D_6 E_2 = 0, \quad D_6^2 E_2 = 0, \quad D_1 E_1^2 = -3, \quad D_2 E_1^2 = -2 \,, \\ D_6 E_1^2 = 0, \quad E_1^2 E_2 = 0, \quad D_1 E_2^2 = -3, \quad D_2 E_2^2 = 0, \quad D_6 E_2^2 = 0 \,, \\ D_1^2 E_1 = 1, \quad D_1 D_2 E_1 = 1, \quad D_2^2 E_1 = 0, \quad D_1 D_6 E_1 = 0, \quad D_2 D_6 E_1 = 0 \,, \\ D_6^2 E_1 = 0 \quad, \quad E_1 E_2^2 = 0, \quad D_1 E_1 E_2 = 0, \quad D_2 E_1 E_2 = 0, \quad D_6 E_1 E_2 = 0 \,, \end{cases} \tag{B.61}$$

$$(\mathbf{u}): \begin{cases} E_1^3 = 9 \quad, \quad E_2^3 = 9 \quad, \quad D_1^2 E_2 = 1 \quad, \quad D_1 D_2 E_2 = 0, \quad D_2^2 E_2 = 0 \,, \\ D_1 D_6 E_2 = 0, \quad D_2 D_6 E_2 = 0, \quad D_6^2 E_2 = 0, \quad D_1 E_1^2 = -3, \quad D_2 E_1^2 = -3 \,, \\ D_6 E_1^2 = 0, \quad E_1^2 E_2 = 0, \quad D_1 E_2^2 = -3, \quad D_2 E_2^2 = 0, \quad D_6 E_2^2 = 0 \,, \\ D_1^2 E_1 = 1, \quad D_1 D_2 E_1 = 1, \quad D_2^2 E_1 = 1, \quad D_1 D_6 E_1 = 0, \quad D_2 D_6 E_1 = 0 \,, \\ D_6^2 E_1 = 0 \quad, \quad E_1 E_2^2 = 0, \quad D_1 E_1 E_2 = 0, \quad D_2 E_1 E_2 = 0, \quad D_6 E_1 E_2 = 0 \,, \end{cases} \tag{B.62}$$

$$(\mathbf{v}): \begin{cases} E_1^3 = 8 \quad, \quad E_2^3 = 9 \quad, \quad D_1^2 E_2 = 0 \quad, \quad D_1 D_2 E_2 = 0, \quad D_2^2 E_2 = 0 \,, \\ D_1 D_6 E_2 = 0, \quad D_2 D_6 E_2 = 0, \quad D_6^2 E_2 = 1, \quad D_1 E_1^2 = -1, \quad D_2 E_1^2 = -2 \,, \\ D_6 E_1^2 = -2, \quad E_1^2 E_2 = 0, \quad D_1 E_2^2 = 0, \quad D_2 E_2^2 = 0, \quad D_6 E_2^2 = -3 \,, \\ D_1^2 E_1 = -1, \quad D_1 D_2 E_1 = 1, \quad D_2^2 E_1 = 0, \quad D_1 D_6 E_1 = 1, \quad D_2 D_6 E_1 = 0 \,, \\ D_6^2 E_1 = 0 \quad, \quad E_1 E_2^2 = 0, \quad D_1 E_1 E_2 = 0, \quad D_2 E_1 E_2 = 0, \quad D_6 E_1 E_2 = 0 \,, \end{cases} \tag{B.63}$$

$$(\mathbf{w}): \begin{cases} E_1^3 = 9 \quad, \quad E_2^3 = 8 \quad, \quad D_1^2 E_2 = 0 \quad, \quad D_1 D_2 E_2 = 0, \quad D_2^2 E_2 = 0 \,, \\ D_1 D_6 E_2 = 0, \quad D_2 D_6 E_2 = 0, \quad D_6^2 E_2 = 1, \quad D_1 E_1^2 = 0, \quad D_2 E_1^2 = -3 \,, \\ D_6 E_1^2 = -3, \quad E_1^2 E_2 = 0, \quad D_1 E_2^2 = 0, \quad D_2 E_2^2 = 0, \quad D_6 E_2^2 = -3 \,, \\ D_1^2 E_1 = 0, \quad D_1 D_2 E_1 = 0, \quad D_2^2 E_1 = 1, \quad D_1 D_6 E_1 = 0, \quad D_2 D_6 E_1 = 1 \,, \\ D_6^2 E_1 = 1 \quad, \quad E_1 E_2^2 = 0, \quad D_1 E_1 E_2 = 0, \quad D_2 E_1 E_2 = 0, \quad D_6 E_1 E_2 = 0 \,, \end{cases} \tag{B.64}$$

$$(\mathbf{x}): \begin{cases} E_1^3 = 9 \quad, \quad E_2^3 = 9 \quad, \quad D_1^2 E_2 = 0 \quad, \quad D_1 D_2 E_2 = 0, \quad D_2^2 E_2 = 0 \,, \\ D_1 D_6 E_2 = 0, \quad D_2 D_6 E_2 = 0, \quad D_6^2 E_2 = 1, \quad D_1 E_1^2 = 0, \quad D_2 E_1^2 = -3 \,, \\ D_6 E_1^2 = -3, \quad E_1^2 E_2 = 0, \quad D_1 E_2^2 = 0, \quad D_2 E_2^2 = 0, \quad D_6 E_2^2 = -3 \,, \\ D_1^2 E_1 = 0, \quad D_1 D_2 E_1 = 0, \quad D_2^2 E_1 = 1, \quad D_1 D_6 E_1 = 0, \quad D_2 D_6 E_1 = 1 \,, \\ D_6^2 E_1 = 1 \quad, \quad E_1 E_2^2 = 0, \quad D_1 E_1 E_2 = 0, \quad D_2 E_1 E_2 = 0, \quad D_6 E_1 E_2 = 0 \,. \end{cases} \tag{B.65}$$

# C  Beetle geometry prepotential and gauge-theory phases

In this appendix, we consider more in detail the M-theory prepotential for the beetle geometry of section 6.2, and we verify that it matches precisely the prepotential of the two S-dual field theory descriptions, in the appropriate Kähler chambers.

## C.1 M-theory prepotential

We parameterize the Kähler class as in equation (6.8), namely:

$$S = \mu_1 D_1 + \mu_2 D_2 + \mu_6 D_6 + \nu_1 \mathbf{E}_1 + \nu_2 \mathbf{E}_2 . \tag{C.1}$$

The M-theory prepotential is then:

$$\mathcal{F} = -\frac{1}{6} \mathbf{E}_a \mathbf{E}_b \mathbf{E}_c \, \nu_a \nu_b \nu_c - \frac{1}{2} D_i \mathbf{E}_a \mathbf{E}_b \, \mu_i \nu_a \nu_b - \frac{1}{2} D_i D_j \mathbf{E}_a \, \mu_i \mu_j \nu_a , \tag{C.2}$$

where $a, b, c = 1, 2$ and $i, j = 1, 2, 6$, and repeated indices are summed over. Here and in the following, we neglect the constant ($\nu$-independent) term, for simplicity. Plugging the intersection numbers listed in Appendix B.3.5 into (C.2), we find the geometric prepotential in every chamber.

**Resolutions with a vertical reduction:** For the first 6 resolutions of Figure 17, we have:

$$\mathcal{F}_a = -\frac{4\nu_1^3}{3} + (\mu_1 + \mu_2)\,\nu_1^2 + \nu_2\,\nu_1^2 - \mu_1\mu_2\,\nu_1 - \mu_1\nu_2\,\nu_1 - \nu_2^3 + \left(\frac{\mu_1}{2} + \frac{\mu_6}{2}\right)\nu_2^2 + \left(\frac{\mu_1^2}{2} - \mu_6\mu_1 + \frac{\mu_6^2}{2}\right)\nu_2 ,$$

$$\mathcal{F}_b = -\frac{7\nu_1^3}{6} + (\mu_1 + \mu_2)\,\nu_1^2 + \frac{1}{2}\nu_2\,\nu_1^2 + \frac{1}{2}\nu_2^2\,\nu_1 - \mu_1\mu_2\,\nu_1 - \mu_1\nu_2\,\nu_1 - \frac{7\nu_2^3}{6} + \left(\frac{\mu_1}{2} + \frac{\mu_6}{2}\right)\nu_2^2 + \left(\frac{\mu_1^2}{2} - \mu_6\mu_1 + \frac{\mu_6^2}{2}\right)\nu_2 ,$$

$$\mathcal{F}_c = -\frac{4\nu_1^3}{3} + \left(\frac{3\mu_1}{2} + \mu_2\right)\nu_1^2 + \left(-\frac{\mu_1^2}{2} - \mu_2\mu_1\right)\nu_1 - \frac{4\nu_2^3}{3} + \left(\mu_1 + \frac{\mu_6}{2}\right)\nu_2^2 + \left(\frac{\mu_6^2}{2} - \mu_1\mu_6\right)\nu_2 ,$$

$$\mathcal{F}_d = -\nu_1^3 + \left(\frac{\mu_1}{2} + \mu_2 + \frac{\mu_6}{2}\right)\nu_1^2 + \nu_2^2\,\nu_1 + \left(\frac{\mu_1^2}{2} - \mu_2\mu_1 - \mu_6\mu_1 + \frac{\mu_6^2}{2}\right)\nu_1 - \mu_6\nu_2\,\nu_1 - \frac{4\nu_2^3}{3} + \mu_6\nu_2^2 ,$$

$$\mathcal{F}_e = -\frac{7\nu_1^3}{6} + \left(\frac{\mu_1}{2} + \mu_2 + \frac{\mu_6}{2}\right)\nu_1^2 + \frac{1}{2}\nu_2\,\nu_1^2 + \frac{1}{2}\nu_2^2\,\nu_1 + \left(\frac{\mu_1^2}{2} - \mu_2\mu_1 - \mu_6\mu_1 + \frac{\mu_6^2}{2}\right)\nu_1 - \mu_6\nu_2\,\nu_1 - \frac{7\nu_2^3}{6} + \mu_6\nu_2^2 ,$$

$$\mathcal{F}_f = -\frac{4\nu_1^3}{3} + \left(\frac{\mu_1}{2} + \mu_2 + \mu_6\right)\nu_1^2 + \left(\frac{\mu_1^2}{2} - \mu_2\mu_1 - \mu_6\mu_1\right)\nu_1 - \frac{4\nu_2^3}{3} + \frac{3}{2}\mu_6\nu_2^2 - \frac{1}{2}\mu_6^2\nu_2 .$$

**Resolutions (g) to (r):**

$$\mathcal{F}_g = -\frac{4\nu_1^3}{3} + (\mu_1 + \mu_2)\,\nu_1^2 + \nu_2\,\nu_1^2 - \mu_1\mu_2\,\nu_1 - \mu_1\nu_2\,\nu_1 - \frac{7\nu_2^3}{6} + \left(\frac{\mu_1}{2} + \frac{\mu_6}{2}\right)\nu_2^2 + \left(\frac{\mu_1^2}{2} - \mu_6\mu_1 + \frac{\mu_6^2}{2}\right)\nu_2 ,$$

$$\mathcal{F}_h = -\frac{4\nu_1^3}{3} + (\mu_1 + \mu_2)\,\nu_1^2 + \nu_2\,\nu_1^2 - \mu_1\mu_2\,\nu_1 - \mu_1\nu_2\,\nu_1 - \frac{7\nu_2^3}{6} + \mu_1\nu_2^2 ,$$

$$\mathcal{F}_i = -\frac{4\nu_1^3}{3} + (\mu_1 + \mu_2)\,\nu_1^2 + \nu_2\,\nu_1^2 - \mu_1\mu_2\,\nu_1 - \mu_1\nu_2\,\nu_1 - \frac{4\nu_2^3}{3} + \mu_1\nu_2^2 ,$$

$$\mathcal{F}_j = -\frac{4\nu_1^3}{3} + \left(\mu_1 + \frac{3\mu_2}{2}\right)\nu_1^2 + \frac{1}{2}\nu_2\,\nu_1^2 + \frac{1}{2}\nu_2^2\,\nu_1 + \left(-\frac{\mu_2^2}{2} - \mu_1\mu_2\right)\nu_1 - \mu_1\nu_2\,\nu_1 - \frac{7\nu_2^3}{6} + \left(\frac{\mu_1}{2} + \frac{\mu_6}{2}\right)\nu_2^2$$
$$+ \left(\frac{\mu_1^2}{2} - \mu_6\mu_1 + \frac{\mu_6^2}{2}\right)\nu_2 ,$$

$$\mathcal{F}_k = -\frac{7\nu_1^3}{6} + (\mu_1 + \mu_2)\,\nu_1^2 + \frac{1}{2}\nu_2\,\nu_1^2 + \frac{1}{2}\nu_2^2\,\nu_1 - \mu_1\mu_2\,\nu_1 - \mu_1\nu_2\,\nu_1 - \frac{4\nu_2^3}{3} + \mu_1\nu_2^2 ,$$

$$\mathcal{F}_l = -\frac{4\nu_1^3}{3} + \left(\mu_1 + \frac{3\mu_2}{2}\right)\nu_1^2 + \frac{1}{2}\nu_2\,\nu_1^2 + \frac{1}{2}\nu_2^2\,\nu_1 + \left(-\frac{\mu_2^2}{2} - \mu_1\mu_2\right)\nu_1 - \mu_1\nu_2\,\nu_1 - \frac{4\nu_2^3}{3} + \mu_1\nu_2^2 ,$$

$$\mathcal{F}_m = -\frac{7\nu_1^3}{6} + \left(\frac{\mu_1}{2} + \frac{3\mu_2}{2} + \frac{\mu_6}{2}\right)\nu_1^2 + \nu_2^2\nu_1 + \left(\frac{\mu_1^2}{2} - \mu_2\mu_1 - \mu_6\mu_1 - \frac{\mu_2^2}{2} + \frac{\mu_6^2}{2}\right)\nu_1 - \mu_6\nu_2\nu_1 - \frac{4\nu_2^3}{3} + \mu_6\nu_2^2 \,,$$

$$\mathcal{F}_n = -\frac{7\nu_1^3}{6} + \left(\frac{3\mu_2}{2} + \mu_6\right)\nu_1^2 + \nu_2^2\nu_1 + \left(-\frac{\mu_2^2}{2} - \mu_6\mu_2\right)\nu_1 - \mu_6\nu_2\nu_1 - \frac{4\nu_2^3}{3} + \mu_6\nu_2^2 \,,$$

$$\mathcal{F}_o = -\frac{4\nu_1^3}{3} + (2\mu_2 + \mu_6)\nu_1^2 + \nu_2^2\nu_1 + \left(-\mu_2^2 - \mu_6\mu_2\right)\nu_1 - \mu_6\nu_2\nu_1 - \frac{4\nu_2^3}{3} + \mu_6\nu_2^2 \,,$$

$$\mathcal{F}_p = -\frac{7\nu_1^3}{6} + \left(\frac{\mu_1}{2} + \mu_2 + \frac{\mu_6}{2}\right)\nu_1^2 + \frac{1}{2}\nu_2\nu_1^2 + \frac{1}{2}\nu_2^2\nu_1 + \left(\frac{\mu_1^2}{2} - \mu_2\mu_1 - \mu_6\mu_1 + \frac{\mu_6^2}{2}\right)\nu_1 - \mu_6\nu_2\nu_1 - \frac{4\nu_2^3}{3} + \mu_6\nu_2^2 \,,$$

$$\mathcal{F}_q = -\frac{4\nu_1^3}{3} + \left(\frac{3\mu_2}{2} + \mu_6\right)\nu_1^2 + \frac{1}{2}\nu_2\nu_1^2 + \frac{1}{2}\nu_2^2\nu_1 + \left(-\frac{\mu_2^2}{2} - \mu_6\mu_2\right)\nu_1 - \mu_6\nu_2\nu_1 - \frac{7\nu_2^3}{6} + \mu_6\nu_2^2 \,,$$

$$\mathcal{F}_r = -\frac{4\nu_1^3}{3} + \left(\frac{3\mu_2}{2} + \mu_6\right)\nu_1^2 + \frac{1}{2}\nu_2\nu_1^2 + \frac{1}{2}\nu_2^2\nu_1 + \left(-\frac{\mu_2^2}{2} - \mu_6\mu_2\right)\nu_1 - \mu_6\nu_2\nu_1 - \frac{4\nu_2^3}{3} + \mu_6\nu_2^2 \,.$$

**Resolutions (s) to (x):**

$$\mathcal{F}_s = -\frac{3\nu_1^3}{2} + \left(\frac{3\mu_1}{2} + \frac{3\mu_2}{2}\right)\nu_1^2 + \left(-\frac{\mu_1^2}{2} - \mu_2\mu_1 - \frac{\mu_2^2}{2}\right)\nu_1 - \frac{4\nu_2^3}{3} + \left(\mu_1 + \frac{\mu_6}{2}\right)\nu_2^2 + \left(\frac{\mu_6^2}{2} - \mu_1\mu_6\right)\nu_2 \,,$$

$$\mathcal{F}_t = -\frac{4\nu_1^3}{3} + \left(\frac{3\mu_1}{2} + \mu_2\right)\nu_1^2 + \left(-\frac{\mu_1^2}{2} - \mu_2\mu_1\right)\nu_1 - \frac{3\nu_2^3}{2} + \frac{3}{2}\mu_1\nu_2^2 - \frac{1}{2}\mu_1^2\nu_2 \,,$$

$$\mathcal{F}_u = -\frac{3\nu_1^3}{2} + \left(\frac{3\mu_1}{2} + \frac{3\mu_2}{2}\right)\nu_1^2 + \left(-\frac{\mu_1^2}{2} - \mu_2\mu_1 - \frac{\mu_2^2}{2}\right)\nu_1 - \frac{3\nu_2^3}{2} + \frac{3}{2}\mu_1\nu_2^2 - \frac{1}{2}\mu_1^2\nu_2 \,,$$

$$\mathcal{F}_v = -\frac{4\nu_1^3}{3} + \left(\frac{\mu_1}{2} + \mu_2 + \mu_6\right)\nu_1^2 + \left(\frac{\mu_1^2}{2} - \mu_2\mu_1 - \mu_6\mu_1\right)\nu_1 - \frac{3\nu_2^3}{2} + \frac{3}{2}\mu_6\nu_2^2 - \frac{1}{2}\mu_6^2\nu_2 \,,$$

$$\mathcal{F}_w = -\frac{3\nu_1^3}{2} + \left(\frac{3\mu_2}{2} + \frac{3\mu_6}{2}\right)\nu_1^2 + \left(-\frac{\mu_2^2}{2} - \mu_6\mu_2 - \frac{\mu_6^2}{2}\right)\nu_1 - \frac{4\nu_2^3}{3} + \frac{3}{2}\mu_6\nu_2^2 - \frac{1}{2}\mu_6^2\nu_2 \,,$$

$$\mathcal{F}_x = -\frac{3\nu_1^3}{2} + \left(\frac{3\mu_2}{2} + \frac{3\mu_6}{2}\right)\nu_1^2 + \left(-\frac{\mu_2^2}{2} - \mu_6\mu_2 - \frac{\mu_6^2}{2}\right)\nu_1 - \frac{3\nu_2^3}{2} + \frac{3}{2}\mu_6\nu_2^2 - \frac{1}{2}\mu_6^2\nu_2 \,.$$

## C.2 Matching to the $SU(2) \times SU(2)$ chambers

The resolutions (a) to (f) can be matched to the six field theory chambers of the $SU(2) \times SU(2)$ quiver. The chambers are shown in equation 6.14. It is convenient to introduce the notation:

$$\begin{aligned}
\vartheta_a &= (1,0,1,0)\,, & \vartheta_d &= (1,1,0,0)\,, \\
\vartheta_b &= (1,1,1,0)\,, & \vartheta_e &= (1,0,0,0)\,, \\
\vartheta_c &= (1,1,1,1)\,, & \vartheta_f &= (0,0,0,0)\,.
\end{aligned} \tag{C.3}$$

Here, the vectors $\vartheta$ denote the field-theory chambers (6.14) in the obvious way; the entries in the vector are 1 or 0 depending on whether the hypermultiplet real masses:

$$\mathcal{M} \equiv (\varphi_1 + \varphi_2 + m \,,\; \varphi_1 - \varphi_2 + m \,,\; -\varphi_1 + \varphi_2 + m \,,\; -\varphi_1 - \varphi_2 + m)\,, \tag{C.4}$$

are positive or negative, respectively. In this notation, the $SU(2) \times SU(2)$ prepotential (6.11) takes the simple form:

$$\mathcal{F}_{\hat{x}}^{SU(2) \times SU(2)} = h_1\varphi_1^2 + h_2\varphi_2^2 + \frac{4}{3}\varphi_1^3 + \frac{4}{3}\varphi_2^3 - \frac{1}{6}\sum_{\alpha=1}^{4}\theta_{\hat{x}}{}^{\alpha}(\mathcal{M}_{\alpha})^3\,, \tag{C.5}$$

where $\hat{x} = (a, \cdots, f)$ runs over the 6 chambers, and $\alpha$ runs over the components of the vectors (C.3) and (C.4). Plugging in the relations (6.13) into the M-theory prepotentials given above, we find perfect agreement with the field theory in all 6 chambers (modulo the constant terms, which we did not keep track of).

## C.3   Matching to the $SU(3)$, $N_f = 2$ chambers

The 16 resolutions (a), (b), (d), (e) and (g) to (r) can be matched to the field theory chambers of the $SU(3)$, $N_f = 2$ field theory. Using the same notation as above, we find:

$$
\begin{aligned}
\vartheta_a &= (1,1,0,1,1,0), & \vartheta_k &= (1,0,0,1,1,1)\,, \\
\vartheta_b &= (1,0,0,1,1,0), & \vartheta_l &= (0,0,0,1,1,1)\,, \\
\vartheta_d &= (1,0,0,1,0,0), & \vartheta_m &= (0,0,0,1,0,0)\,, \\
\vartheta_e &= (1,1,0,1,0,0), & \vartheta_n &= (1,0,0,0,0,0)\,, \\
\vartheta_g &= (1,1,1,1,1,0), & \vartheta_o &= (0,0,0,0,0,0)\,, \\
\vartheta_h &= (1,1,0,1,1,1), & \vartheta_p &= (1,1,1,1,0,0)\,, \\
\vartheta_i &= (1,1,1,1,1,1), & \vartheta_q &= (1,1,0,0,0,0)\,, \\
\vartheta_j &= (0,0,0,1,1,0), & \vartheta_r &= (1,1,1,0,0,0)\,.
\end{aligned}
\tag{C.6}
$$

Here, the 6 hypermultiplet modes are:

$$
\mathcal{M} \equiv (\widetilde{\varphi}_1 + \widetilde{m}_1\,,\, -\widetilde{\varphi}_1 + \widetilde{\varphi}_2 + \widetilde{m}_1\,,\, -\widetilde{\varphi}_2 + \widetilde{m}_1\,,\, \widetilde{\varphi}_1 + \widetilde{m}_2\,,\, -\widetilde{\varphi}_1 + \widetilde{\varphi}_2 + \widetilde{m}_2\,,\, -\widetilde{\varphi}_2 + \widetilde{m}_2)\,. \tag{C.7}
$$

The gauge-theory prepotential (C.8) in chamber $\hat{x}$ then reads:

$$
\begin{aligned}
\mathcal{F}_{\hat{x}}^{SU(3),N_f=2} &= \widetilde{h}_0(\widetilde{\varphi}_1^2 + \widetilde{\varphi}_2^2 - \widetilde{\varphi}_1\widetilde{\varphi}_2) + \frac{1}{2}(\widetilde{\varphi}_1^2\widetilde{\varphi}_2 - \widetilde{\varphi}_1\widetilde{\varphi}_2^2) \\
&\quad + \frac{4}{3}(\widetilde{\varphi}_1^3 + \widetilde{\varphi}_2^3) - \frac{1}{2}(\widetilde{\varphi}_1^2\widetilde{\varphi}_2 + \widetilde{\varphi}_1\widetilde{\varphi}_2^2) - \frac{1}{6}\sum_{\alpha=1}^{6}\theta_{\hat{x}}{}^{\alpha}(\mathcal{M}_{\alpha})^3\,.
\end{aligned}
\tag{C.8}
$$

Plugging in the relations (6.40) into the M-theory prepotentials given above, we again find perfect agreement with the field theory description, in all 16 chambers (C.6).

## C.4   Selected chambers of the deformed beetle SCFT

To conclude this appendix, we provide some representative examples of the phases of the beetle geometry, which illustrate the essential features of the phase structure captured by geometry—see Figures 33 and 34. The phase diagram of the beetle geometry, parametrized by $(\boldsymbol{\nu}; \boldsymbol{\mu}) \equiv (\nu_1, \nu_2; \mu_1, \mu_2, \mu_6)$, is a five dimensional region, given by the (disjoint) union of the regions described by the defining inequalities of 24 Kähler chambers. We visualize it by taking slices at fixed values of $(\mu_1, \mu_2, \mu_6)$, revealing different chambers. Certain chambers vanish altogether for some ranges of $(\boldsymbol{\nu}; \boldsymbol{\mu})$, whereas other ones collapse to real codimension-one boundaries in this parameter space (along which flops may occur). On the various plots, the origin $(\nu_1, \nu_2) = (0,0)$ is denoted by a red dot. For the sake of clarity, in Figures 33 and 34, we only indicate chambers that have finite area in parameter space in the slices that we consider. Note also that, when $\boldsymbol{\mu} \neq 0$, the origin $\nu_1 = \nu_2 = 0$ is not generally the origin of the gauge-theory Coulomb branch, when such a description is available, since the map (6.13) between $\nu_1, \nu_2$ and $\varphi_1, \varphi_2$ for $SU(2) \times SU(2)$ is non-trivial, and similarly for the $SU(3)$ phases.

Flops can occur when two chambers are separated by interior walls. This is consistent with the allowed flop transitions between resolutions, as one can infer from Figure 17. For instance, consider the plot (iii) in Figure 33. As we can see from the triangulations in Figure 17, one can indeed start from resolution (w), then go to resolution (q) by flopping a single curve, then go to either resolution (n) or resolution (e) by a single flop, and so on and so forth.

(i): phases: (a), (d)
$\boldsymbol{\mu} = (0, 0, 0)$

(ii): phases: (a), (d), (e), (q), (n)
$\boldsymbol{\mu} = (0.35, 0, 0)$

(iii): phases: (a), (d), (e), (q), (n), (w)
$\boldsymbol{\mu} = (0.15, 0.15, -0.30)$

(iv): phases: (a), (b), (d), (i), (k), (q)
$\boldsymbol{\mu} = (0.28, 0.66, 0.46)$

Figure 33: Sample slices of the moduli space of the beetle geometry. In (i) we see the Coulomb Branch of the beetle SCFT, where chambers (b) and (e) degenerate along the wall. Turning on distinct mass deformations can open up distinct gauge theory phases, some of which may be connected to the origin as in (ii) and (iv), but also non-gauge theoretic phase such as (w) in (iii).

# D   The conifold and the hypermultiplet

In this appendix, we consider a somewhat degenerate case: "rank-zero" isolated toric $CY_3$ singularities, whose toric diagrams have no internal points. The only such singularity is the conifold, whose toric diagram is shown in Figure 35(a). It defines the simplest 5d SCFT, a free massless hypermultiplet.



(v): phases: (a), (d), (e), (n), (q), (r)
$\mu = (0.35, 0.11, 0.13)$

(vi): phases: (a), (d), (e), (n), (o), (q)
$\mu = (0.09, -0.38, -0.05)$

(vii): phases: (h), (k), (l), (u)
$\mu = (-0.99, -0.66, 0.66)$

(viii): phases: (a), (b), (h), (k), (t)
$\mu = (-0.66, -0.22, 0.22)$

Figure 34: Sample slices of the moduli space of the beetle geometry, (contd). Turning on different mass deformations reveals more $SU(3)$ $N_f = 2$ phases such as (q), (n), (o) and (r) as in (i) and (ii), but also non-gauge theoretic phases such as (u) in (vii) and (t) in (viii).

## D.1 Two ways of slicing a conifold

The GLSM of the conifold reads:

|  | $D_1$ | $D_2$ | $D_3$ | $D_4$ |  |
|---|---|---|---|---|---|
| $\mathcal{C}_1$ | 1 | −1 | 1 | −1 | $\xi_1$ |
| $U(1)_M$ | 1 | −1 | 0 | 0 | $r_0$ |
| $U(1)_{M'}$ | 0 | −1 | 0 | 1 | $r_0$ |

(D.1)

Here we also introduced two distinct M-theory circles, to be discussed momentarily. The small resolution of the conifold has a single exceptional curve, $\mathcal{C}_1 \cong \mathbb{P}^1$, with volume $\xi_1 > 0$. The

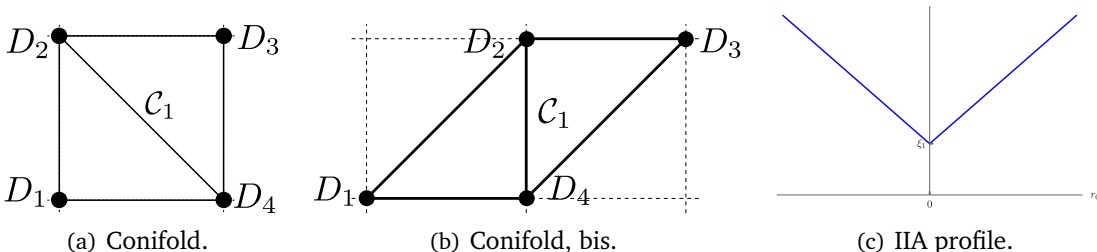

(a) Conifold.       (b) Conifold, bis.       (c) IIA profile.

Figure 35: The two toric diagrams are related by an $SL(2,\mathbb{Z})$ transformation. The vertical reduction of the second toric diagram gives the IIA profile displayed on the right.

toric divisors are as indicated on Fig. 35(a).

**Conifold as hypermultiplet.** If we perform the vertical reduction on the toric diagram of Fig. 35(a), it is clear that we obtain a IIA background $\mathcal{M}_5 \cong \mathbb{C}^2 \times \mathbb{R}$, with two non-compact D6-branes along the two $\mathbb{C}$ factors in $\mathbb{C}^2$:

$$
\begin{array}{c|ccccc|cc cc|c}
x^\mu & x^0 & x^1 & x^2 & x^3 & x^4 & x^5 & x^6 & x^7 & x^8 & x^9 = r_0 \\
\hline
\text{D6}_1 & \times & \times & \times & \times & \times & \times & \times & & & r_0 = 0 \\
\text{D6}_2 & \times & \times & \times & \times & \times & & & \times & \times & r_0 = \xi_1
\end{array}
\tag{D.2}
$$

These D6-branes are located at $r_0 = 0$ and $r_0 = \xi_1$, respectively. There is a single five-dimensional mode, the open string stretched between the two D6-branes, which gives rise to a single 5d $\mathcal{N} = 1$ hypermultiplet $\mathcal{H}$, with a real mass $m = \xi_1$.

**Conifold as "$SU(1)$ gauge theory."** The conifold admits another, inequivalent IIA reduction, as indicated by the $U(1)_{M'}$ charges in (35), corresponding to a vertical reduction of the $SL(2,\mathbb{Z})$-transformed toric diagram shown in Figure 35(b). The IIA description is in terms of a single D6-brane wrapping the exceptional $\mathbb{P}^1$ in a resolved $A_1$ singularity. This gives us a naive "$SU(1)$ gauge theory", which has no Coulomb branch parameter but still has an "$SU(1)$ gauge coupling," $h_0$. The corresponding IIA profile reads:

$$
\chi(r_0) = \begin{cases} r_0 + \xi_1 & \text{if } r_0 > 0 \,, \\ -r_0 + \xi_1 & \text{if } r_0 < 0 \,, \end{cases}
\tag{D.3}
$$

as shown in Figure 35(c). We thus find an effective CS level $k_{\text{eff}} = 0$ and the identification:

$$
\xi_1 = h_0 \,,
\tag{D.4}
$$

between the Kähler parameter and the gauge coupling. The M2-brane wrapped on $\mathcal{C}_1$ is an "$SU(1)_0$ instanton," in this description, while it was giving rise to a single hypermultiplet in the previous description. This is the same "duality" that we invoked in our discussion of the (toric) $T_2$ theory in section 7.1.2.

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
