# Peer review of "Five-dimensional SCFTs and gauge theory phases: an M-theory/type IIA perspective"

_SciPost Physics, doi:SciPost Phys. 6, 052 (2019)_

## Round 2 · Referee Report · Anonymous · 2019-2-22

Strengths
1-Provides clear understanding and systematic method to understand supersymmetric gauge theory from geometry.
2-Resolved several confusions in the past literatures.
3-Big impact due to the modification to the famous well known formula for the effective prepotential.
4-Self contained with various reviews.
5-Very detailed analysis.
Weaknesses
1-Too long.
Report
This is an interesting paper, which provides clear understanding about the relation between toric Calabi-Yau 3-folds and five dimensional supersymmetric gauge theories. Although such relation had been studied in various past literatures, there were still some unclear, subtle or confusing points, which potentially lead to misunderstanding. This paper resolved some of such problems.
One of the important results of this paper is to demonstrate the agreement between the gauge theory effective prepotential obtained from the perturbative 1-loop computation and the geometric potential computed from the triple intersection number with all the mass parameters turned on. During this process, this paper clarifies the geometric phase denoted as "gauge theory phases", where gauge theory description is possible, among all the possible resolution from the Calabi-Yau singularity. This paper gives systematic method from the point of view of M-theory/type IIA duality. This consideration turns out to be very useful also for better understanding of "UV dualities".
In this paper, the authors proposed the modification to the famous IMS (Intriligator-Morrison-Seiberg) prepotential based on the careful consideration of the quantization of the Chern-Simons level including the background gauge field for global symmetry. Also, the new prepotential is shown to be derived from the limit of the prepotential for the four dimensional gauge theory with Kaluza-Klein modes. This would be a very big impact to this field of study because so many papers had used the IMS prepotential until recently. This modification can be regarded as the redefinition of parameters for the SU(N) gauge theory. As stated in the paper, the extra (vev)(vev)(vev) term looks like the half integer shift of Chern-Simons level. Also, the difference in the form of (mass)(vev)(vev) can be absorbed by shifting the inverse bare coupling square by the mass parameters with half integer coefficients. The term (mass)(mass)(vev) will vanish due to traceless condition of SU(N). Therefore, from the point of view of this paper, the literatures in the past were identifying Chern-Simons level and the bare coupling in a confusing way.
Based on the new parametrization, this paper also clarified how we should parametrize Kahler parameters in terms of the parameters in the corresponding gauge theory including Chern-Simons level and bare coupling. Many of the past literatures, including the ones studying (p,q) 5-brane webs, used different parametrization than this paper in order to make agreement with the IMS prepotential. Although the problem of the parametrization of this kind had been often confusing and controversial when we try to see the consistency between the gauge theory and the geometry or the 5-brane web, the logic in this paper looks consistent and plausible to me.
This paper includes a lot of conceptual improvement about understanding five dimensional supersymmetric gauge theories from Calabi-Yau geometry as well as detailed analysis on the gauge theory phases. Therefore, I recommend this paper for publication after minor revision.
Requested changes
1- It would be better to mention that the correction to the IMS prepotential made by this paper can be absorbed into the redefinition of the bare coupling as well as Chern-Simons level even for massive case if the gauge group is SU(N):
Since the authors proposed the correction to the IMS prepotential, the readers may wonder what would happen to all the past papers written based on the IMS prepotential. For example, in some of the past literatures (e.g. Aharony-Hanany-Kol) indeed used the IMS prepotential to show the agreement between its first derivative and the area of the face in the corresponding (p,q) 5-brane web diagram even for the massive case. They seem to use different parametrization than this paper, which is expected to be the difference of the parametrization related to the modification of the IMS prepotential. Therefore, agreement will still hold after modifying the parametrization.

---

## Round 2 · Referee Report · Anonymous · 2019-3-29

Strengths
1- A careful re-analysis of both 5d SCFTs, and the M-theory geometry realizations and type IIA duality.
2-Finds interesting new results, and points out some problems with past analysis, and their corrections.
3-Very thorough and self-contained presentation.
4-Clearly and beautifully written.
Weaknesses
None.
Report
The paper is very clear and well-written, and I think that it merits publication. I generally agree with the comments and suggestions of the other referee. The paper provides a novel re-analysis of the M-theory realizations of 5d SCFTs, and their deformations, by assuming that the geometry is toric and using the connection to type IIA. The paper also makes important, new observations about the prepotential of 5d theories and the parity anomaly and CS contact terms. They find an expression for the prepotential that differs a bit from the one that was obtained by IMS in 1997. They explain the difference / relation in footnote 11, and the other referee also remarked about that. I agree with these comments, and I agree that the prepotential found here is an improvement vs the earlier literature. The paper clarifies these issues, and also the connection to type IIA, by considering many illuminating examples. The paper also explores and clarifies the S-duality map between dual 5d SCFTs, for example SU(3) with two flavors and SU(2) x SU(2). The paper also considers the transition boundaries of the Coulomb branch, including those where there are tensionless strings, and the string theory + geometry perspective. The paper is very thorough, and is full of interesting and important new results. I would also like to apologize to the editor, and the authors, that my report was delayed past the deadline. I have been eager to study the paper in detail, and this referee assignment gave good impetus for that, but the paper is rather long and it took me some time to get through. About the length, I think that it is OK for publication, and I appreciate that the authors wrote such a beautifully clear and relatively self-contained presentation.
Requested changes
None.

---

## Editorial Decision

published